# Mean-field analysis for heavy ball methods: Dropout-stability, connectivity, and global convergence

**Diyuan Wu**                                                                     *diyuan.wu@ista.ac.at*
*Institute of Science and Technology Austria (ISTA)*

**Vyacheslav Kungurtsev**                                                         *kunguvya@fel.cvut.cz*
*Czech Technical University*

**Marco Mondelli**                                                               *marco.mondelli@ista.ac.at*
*Institute of Science and Technology Austria (ISTA)*

**Reviewed on OpenReview:** *https://openreview.net/forum?id=gZna3IiGfl*

## Abstract

The stochastic heavy ball method (SHB), also known as stochastic gradient descent (SGD) with Polyak's momentum, is widely used in training neural networks. However, despite the remarkable success of such algorithm in practice, its theoretical characterization remains limited. In this paper, we focus on neural networks with two and three layers and provide a rigorous understanding of the properties of the solutions found by SHB: *(i)* stability after dropping out part of the neurons, *(ii)* connectivity along a low-loss path, and *(iii)* convergence to the global optimum. To achieve this goal, we take a mean-field view and relate the SHB dynamics to a certain partial differential equation in the limit of large network widths. This mean-field perspective has inspired a recent line of work focusing on SGD while, in contrast, our paper considers an algorithm with momentum. More specifically, after proving existence and uniqueness of the limit differential equations, we show convergence to the global optimum and give a quantitative bound between the mean-field limit and the SHB dynamics of a finite-width network. Armed with this last bound, we are able to establish the dropout-stability and connectivity of SHB solutions.

## 1 Introduction

Neural networks are one of the most popular modeling tools in machine learning tasks. In practice, they can be effectively trained by gradient-based methods, and are usually overparameterized. However, despite the empirical success of various training algorithms, it is still not well understood why such algorithms have good convergence properties, given that the optimization landscape is known to be highly non-convex and to contain spurious local minima (Auer et al., 1996; Safran & Shamir, 2018; Yun et al., 2019). A popular line of work starting from (Mei et al., 2018; Chizat & Bach, 2018; Rotskoff & Vanden-Eijnden, 2018; Sirignano & Spiliopoulos, 2020a) has proposed a new regime to analyze the behavior of stochastic gradient descent (SGD), namely, the *mean-field* regime. The idea is that, as the number of neurons of the network grows, the SGD training dynamics converges to the solution of a certain Wasserstein gradient flow. This perspective has facilitated the study of architectures with multiple layers (Araújo et al., 2019; Lu et al., 2020; Nguyen & Pham, 2020; Fang et al., 2021), and it has provided a path to rigorously understand a number of properties, including convergence towards a global optimum (Mei et al., 2018; Chizat & Bach, 2018; Javanmard et al., 2020; Pham & Nguyen, 2021a), dropout-stability and connectivity (Shevchenko & Mondelli, 2020), and implicit bias (Williams et al., 2019; Chizat & Bach, 2020; Shevchenko et al., 2022).

Optimization with momentum, e.g., the heavy ball method (Polyak, 1964) or Adam (Kingma & Ba, 2015), is widely used in practice (Sutskever et al., 2013). However, all the aforementioned works consider the vanilla

SGD algorithm and, in general, the theoretical understanding of algorithms with momentum has lagged behind. To address this gap, the recent paper by Krichene et al. (2020) defines a mean-field limit for the stochastic heavy ball (SHB) method – also known as SGD with Polyak's momentum – in a two-layer setup. In particular, the convergence to the mean-field limit is proved, as well as that the solution of the mean-field equation approaches a global optimum in the large time limit. However, Krichene et al. (2020) leave as an open problem finding a *quantitative* bound between the infinite-width limit and the finite-width neural network, and the analysis is restricted to two-layer networks.

In this paper, we define a mean-field limit for the heavy ball method applied to two-layer and three-layer networks. We show global convergence in the three-layer setting, and give quantitative bounds for networks with finite widths. This last result opens the way to providing a rigorous understanding of effects commonly observed in practice, such as the connectivity of solutions via low-loss paths (Goodfellow & Vinyals, 2015; Garipov et al., 2018; Draxler et al., 2018; Entezari et al., 2022). More specifically, our main contributions can be summarized as follows:

1. We show existence and uniqueness of the mean-field differential equations capturing the SHB training for two-layer and three-layer networks (Theorem 4.1).

2. We give *non-asymptotic* convergence results of the SHB dynamics of a finite-width neural network to the corresponding mean-field limit (Theorem 5.1 for two layers, and Theorem 5.2 for three layers). Our bounds are *dimension-free* in the sense that the layer widths are not required to scale with the input dimension.

3. We discuss how SHB solutions can be connected via a simple piece-wise linear path, along which the increase in loss vanishes as the width of the network grows (Section 6). This is a consequence of the stability against dropout displayed by the parameters found by SHB, which in turn follows from Theorems 5.1-5.2.

4. Finally, by exploiting a universal approximation property enjoyed by the activation function, we prove a global convergence result for three-layer networks, under certain assumptions on the mode of convergence of the dynamics (Theorem 7.2).

**Organization of the paper.** After discussing the related work in Section 2, the details concerning the network architecture and SHB training are presented in Section 3. In Section 4, we define the mean-field limit for both two-layer and three-layer networks, and we show that such limits exist and are unique. Next, in Section 5, we prove our quantitative convergence bounds between the discrete SHB dynamics and the mean-field limit. As an application of these bounds, we discuss the dropout-stability and connectivity displayed by the SHB solutions in Section 6. In Section 7, we prove global convergence for the three-layer mean-field differential equation. In Section 8, we show the results of some numerical experiments that illustrate the dropout stability of the SHB solution. Finally, we discuss future directions in Section 9.

## 2 Related work

**Mean-field analysis for two-layer networks.** A quantitative convergence result of the SGD dynamics towards a mean-field limit was presented in (Mei et al., 2018). This bound was refined to be independent of the input dimension in (Mei et al., 2019), which also considers a setting where both layers are trained. Our approach extends this line of work to the heavy-ball method. Because of the presence of momentum, in this case the mean-field limit is described by a second-order differential equation (instead of the first-order one capturing the SGD dynamics). The mean-field limit for heavy-ball methods was first considered in (Krichene et al., 2020), which deals with a setting regularized by noise and does not provide quantitative bounds. The recent work by Schuh (2022) gives non-asymptotic guarantees, but the argument crucially relies on the presence of additive Gaussian noise. Global optimality of SGD was proved by Chizat & Bach (2018) in the noiseless setting, using the homogeneity of the activation and under an additional convergence assumption. This assumption is not needed in the noisy case, as the mean-field dynamics is a Wasserstein gradient flow for a strongly convex free-energy functional (Mei et al., 2019). Convergence rates have been

obtained by exploiting displacement convexity (Javanmard et al., 2020) or via the log-Sobolev constant of the stationary measure (Chizat, 2022; Nitanda et al., 2022). For heavy ball methods, although the limiting dynamics does not yield a gradient flow structure, the rate of convergence has been studied in the noisy case (Hu et al., 2019; Kazeykina et al., 2020; Schuh, 2022). For a noiseless dynamics, the explicit characterization of the convergence rate is an open problem.

**Mean-field analysis for multi-layers networks.** The case of neural networks with more than two layers presents additional challenges in regards to defining and characterizing a mean-field limit. Here, we follow the "neuronal embedding" framework (Nguyen & Pham, 2020; Pham & Nguyen, 2021a) to define a mean-field limit for the heavy ball method in a three-layer setting. The key idea is to introduce a certain *fixed* product probability space at initialization, and regard the weights as *deterministic* functions which evolve during training over this fixed probability space. Other approaches have been proposed as well: Sirignano & Spiliopoulos (2020a) give asymptotic results for three-layer networks, which require taking the large-width limits in a specific sequential order; Fang et al. (2021) focus on the dynamics of the features, rather than on that of the weights; Araújo et al. (2019) consider neural networks with more than four layers, with un-trained first and last layer, and their result depends on the degeneracy phenomenon of middle layers under i.i.d. initialization. This degeneracy consists in the fact that all the weights in the middle layers (except the second and second-to-last) remain i.i.d., hence the corresponding neurons compute the same function. Pham & Nguyen (2021a) show global optimality of the mean-field dynamics in the noiseless case in the large time limit, under a convergence assumption in the same spirit of Chizat & Bach (2018). While the global convergence in Chizat & Bach (2018) crucially relies on the homogeneity of the activation, Pham & Nguyen (2021a) exploit a universal approximation property, similarly to Lu et al. (2020). For deeper networks, a key technical hurdle is due to the degeneracy phenomenon mentioned above, which prevents universal approximation. To address this issue, Nguyen & Pham (2020) employ a different initialization, and Lu et al. (2020); Fang et al. (2021) use skip connections.

For multi-layer networks, it is also possible to define a mean-field limit beyond the parameterization considered in the aforementioned works, see Hajjar et al. (2021); Chen et al. (2022). In particular, Hajjar et al. (2021) consider the regime of so called *integrable parameterizations*, which is a modification of the *maximum-update parameterization* proposed by Yang & Hu (2021). Furthermore, Chen et al. (2022) provide a convergence result for three-layer networks, where the first layer is random and fixed.

**Additional related work on optimization with momentum.** A recent line of work has considered training neural networks in the neural tangent kernel (NTK) regime (or lazy regime) (Jacot et al., 2018; Chizat et al., 2019). Here, the weights of the neural networks stay close to their initialization, so that the network is well approximated by its linearization and the convergence is related to the smallest eigenvalue of the NTK, see e.g. (Du et al., 2019; Allen-Zhu et al., 2019; Montanari & Zhong, 2020; Bombari et al., 2022) and references therein. This type of analysis has been adapted to the heavy ball method by Wang et al. (2021) and to other adaptive methods by Wu et al. (2019). However, we remark that, unlike in the mean-field regime, neural networks are unable to perform feature learning in the NTK regime (Yang & Hu, 2021). Beyond the training of neural networks, momentum-based stochastic gradient descent algorithms and their continuous variants have been widely studied in optimization: the continuous limit of these methods is studied in (Su et al., 2014; Wibisono et al., 2016; Shi et al., 2021), and such dynamics are known to be closely related to sampling methods such as MCMC (Ma et al., 2021).

## 3 Problem setup

### 3.1 Network architecture

We consider neural networks with two and three layers. In the two-layer case, the network has $n$ neurons and input $\boldsymbol{x} \in \mathbb{R}^D$:

$$H_1(\boldsymbol{x}, j; \boldsymbol{W}) = \boldsymbol{w}_1(j)^T \boldsymbol{x}, \quad j \in [n],$$
$$f(\boldsymbol{x}; \boldsymbol{W}) = \frac{1}{n} \sum_{j=1}^{n} w_2(j) \sigma(H_1(\boldsymbol{x}, j; \boldsymbol{W})). \tag{1}$$

Here, we use the short-hand $[n] := \{1, \ldots, n\}$ and, for $j \in [n]$, the parameters of the $j$-th neuron are denoted by $\boldsymbol{\theta}(j) = (\boldsymbol{w}_1(j), w_2(j))$, with $\boldsymbol{w}_1(j) \in \mathbb{R}^D$ and $w_2(j) \in \mathbb{R}$. In the three-layer case, the network has $n_1$ and $n_2$ neurons in the first and second hidden layer, respectively:

$$
\begin{aligned}
H_1(\boldsymbol{x}, j_1; \boldsymbol{W}) &= \boldsymbol{w}_1(j_1)^T \boldsymbol{x}, \quad j_1 \in [n_1], \\
H_2(\boldsymbol{x}, j_2; \boldsymbol{W}) &= \frac{1}{n_1} \sum_{j_1=1}^{n_1} w_2(j_1, j_2) \sigma_1(H_1(\boldsymbol{x}, j_1; \boldsymbol{W})), \quad j_1 \in [n_1], j_2 \in [n_2], \\
f(\boldsymbol{x}; \boldsymbol{W}) &= \frac{1}{n_2} \sum_{j_2=1}^{n_2} w_3(j_2) \sigma_2(H_2(\boldsymbol{x}, j_2; \boldsymbol{W})).
\end{aligned}
\tag{2}
$$

Here, $\boldsymbol{x} \in \mathbb{R}^D, \boldsymbol{w}_1(j_1) \in \mathbb{R}^D, w_2(j_1, j_2) \in \mathbb{R}$ and $w_3(j_2) \in \mathbb{R}$. In both cases, we use $\boldsymbol{W}$ to denote the collection of all parameters: in two-layer case $\boldsymbol{W} \in \mathbb{R}^{n(D+1)}$, and in three-layer case $\boldsymbol{W} \in \mathbb{R}^{n_1 D + n_2 n_1 + n_2}$.

### 3.2 Training algorithm

Our training data $\boldsymbol{z} = (\boldsymbol{x}, y)$ is generated i.i.d. from a distribution $\mathcal{D}$. The neural network is trained to minimize the population risk function $R(\boldsymbol{W}) = \mathbb{E}_{\boldsymbol{z}}[R(y, f(\boldsymbol{x}; \boldsymbol{W}))]$ via the following one-pass stochastic heavy ball (SHB) method:

$$
\boldsymbol{W}(k+1) = \boldsymbol{W}(k) + \beta(\boldsymbol{W}(k) - \boldsymbol{W}(k-1)) - \eta \widehat{\nabla}_{\boldsymbol{W}} R(y(k), f(\boldsymbol{x}(k); \boldsymbol{W}(k))),
\tag{3}
$$

where we use $\widehat{\nabla}_{\boldsymbol{W}} R(y(k), f(\boldsymbol{x}(k); \boldsymbol{W}))$ to denote the scaled gradient, and the scaling factors for each parameter are specified below. This is a *one-pass* method in the sense that, at each step, we sample a new data point $\boldsymbol{z}(k)$ independent from the previous ones. The requirements on the loss function $R(y, \hat{y})$ are contained in (A1) (see Assumption 3.1) and (B1) (see Assumption 3.3) for networks with two and three layers, respectively. In particular, we require $R(y, \hat{y})$ to be differentiable with respect to its second argument and to have a bounded derivative. We also remark that, for the convergence to the mean field limit (Theorem 5.1 and 5.2), the convexity of the loss is not required. In contrast, to obtain the global convergence result of Theorem 7.2, we need to additionally assume that $R(y, \hat{y})$ is convex in the second argument, as mentioned in the statement of the theorem.

Let $\gamma$ be a constant which does not depend on the network width or on the step size of gradient descent. Then, in order to define a continuous-time ODE for the heavy ball method, we pick $\beta = (1 - \gamma\varepsilon)$ and $\eta = \varepsilon^2$, so the one-pass SHB method can be equivalently written as follows:

$$
\begin{aligned}
\boldsymbol{W}(k+1) &= \boldsymbol{W}(k) + \boldsymbol{r}(k), \\
\boldsymbol{r}(k) &= (1 - \gamma\varepsilon)(\boldsymbol{W}(k) - \boldsymbol{W}(k-1)) - \varepsilon^2 \widehat{\nabla}_{\boldsymbol{W}} R(y(k), f(\boldsymbol{x}(k); \boldsymbol{W}(k))).
\end{aligned}
\tag{4}
$$

A similar formulation is common in the literature, see for example (Shi et al., 2021, Eq. 1.2)[1]. The corresponding continuous ODE, also studied in (Krichene et al., 2020, equation 6), is given by

$$
\partial_t \boldsymbol{W}(t) = \boldsymbol{r}(t), \qquad \partial_t \boldsymbol{r}(t) = -\gamma \boldsymbol{r}(t) - \widehat{\nabla}_{\boldsymbol{W}} R(\boldsymbol{W}(t)),
\tag{5}
$$

where we recall that $\widehat{\nabla}_{\boldsymbol{W}} R(\boldsymbol{W}(t))$ denotes the scaled gradient. We remark that there are different ways to derive a continuous dynamics from (3), and (4) is obtained by applying the Euler scheme based on the second-order Taylor expansion. The corresponding ODE (5) is denoted as the low-resolution ODE by Shi et al. (2021). It is an interesting and challenging task to analyze other types of ODEs associated to the SHB method, for example the high-resolution ODE proposed by Shi et al. (2021). We leave this to future works. We also remark that a similar formulation of the continuous counterpart of heavy ball methods with fixed momentum is considered in (Kovachki & Stuart, 2021; Kunin et al., 2021).

---

[1]Note that in (Shi et al., 2021, Eq. 1.2), $\beta = \frac{1-\gamma\epsilon}{1+\gamma\epsilon}$, while here we let $\beta = 1 - \gamma\varepsilon$. The two choices are basically the same when $\varepsilon$ is small.

We conclude this part by discussing the scaling factors for the gradient in (4). In the two-layer case, we have

$$\widehat{\nabla}_{\boldsymbol{W}} R(y, f(\boldsymbol{x}; \boldsymbol{W})) = \left( (\Delta_1^W(\boldsymbol{x}, j; \boldsymbol{W}))_{j \in [n]}, (\Delta_2^W(\boldsymbol{x}, j; \boldsymbol{W}))_{j \in [n]} \right), \tag{6}$$

where

$$\begin{aligned}
\Delta_2^W(\boldsymbol{x}, j; \boldsymbol{W}) &:= n \partial_{w_2(j)} R(y, f(\boldsymbol{x}; \boldsymbol{W})) = \partial_2 R(y, f(\boldsymbol{x}; \boldsymbol{W})) \sigma(H_1(\boldsymbol{x}, j; \boldsymbol{W})), \\
\Delta_1^W(\boldsymbol{x}, j; \boldsymbol{W}) &:= n \nabla_{\boldsymbol{w}_1(j)} R(y, f(\boldsymbol{x}; \boldsymbol{W})) = \partial_2 R(y, f(\boldsymbol{x}; \boldsymbol{W})) w_2(j) \sigma'(H_1(\boldsymbol{x}, j; \boldsymbol{W})) \boldsymbol{x}.
\end{aligned} \tag{7}$$

In the three-layer case, we have

$$\widehat{\nabla}_{\boldsymbol{W}} R(y, f(\boldsymbol{x}; \boldsymbol{W})) = \left( (\Delta_1^W(\boldsymbol{x}, j_1; \boldsymbol{W}))_{j_1 \in [n_1]}, (\Delta_2^W(\boldsymbol{x}, j_1, j_2; \boldsymbol{W}))_{j_1 \in [n_1], j_2 \in [n_2]}, (\Delta_3^W(\boldsymbol{x}, j_2; \boldsymbol{W}))_{j_2 \in [n_2]} \right), \tag{8}$$

where

$$\begin{aligned}
\Delta_3^W(\boldsymbol{x}, j_2; \boldsymbol{W}) &:= n_2 \partial_{w_3(j_2)} R(y, f(\boldsymbol{x}; \boldsymbol{W})) = \partial_2 R(y, f(\boldsymbol{x}; \boldsymbol{W})) \sigma_2(H_2(\boldsymbol{x}, j_2; \boldsymbol{W})), \\
\Delta_2^H(\boldsymbol{x}, j_2; \boldsymbol{W}) &:= n_2 \partial_{H_2(\boldsymbol{x}, j_2; \boldsymbol{W})} R(y, f(\boldsymbol{x}; \boldsymbol{W})) = \partial_2 R(y, f(\boldsymbol{x}; \boldsymbol{W})) w_3(j_2) \sigma_2'(H_2(\boldsymbol{x}, j_2; \boldsymbol{W})), \\
\Delta_2^W(\boldsymbol{x}, j_1, j_2; \boldsymbol{W}) &:= n_1 n_2 \partial_{w_2(j_1, j_2)} R(y, f(\boldsymbol{x}; \boldsymbol{W})) = \Delta_2^H(\boldsymbol{x}, j_2; \boldsymbol{W}) \sigma_1(H_1(\boldsymbol{x}, j_1; \boldsymbol{W})), \\
\Delta_1^H(\boldsymbol{x}, j_1; \boldsymbol{W}) &:= n_1 \partial_{H_1(\boldsymbol{x}, j_1; \boldsymbol{W})} R(y, f(\boldsymbol{x}; \boldsymbol{W})) = \frac{1}{n_2} \sum_{j_2=1}^{n_2} \Delta_2^H(\boldsymbol{x}, j_2; \boldsymbol{W}) w_2(j_1, j_2) \sigma_1'(H_1(\boldsymbol{x}, j_1; \boldsymbol{W})), \\
\Delta_1^W(\boldsymbol{x}, j_1; \boldsymbol{W}) &:= n_1 \nabla_{\boldsymbol{w}_1(j_1)} R(y, f(\boldsymbol{x}; \boldsymbol{W})) = \Delta_1^H(\boldsymbol{x}, j_1; \boldsymbol{W}) \boldsymbol{x}.
\end{aligned} \tag{9}$$

One can interpret this as indicating that in the two-layer case, the scaling factor is $n$. In the three-layer case, the scaling factor is $n_2$ for the third layer, $n_1 \times n_2$ for the second layer, and $n_1$ for the first layer. This choice of the scaling factors ensures that each component of the gradients is of order 1 (i.e., independent of the layer widths $n, n_1, n_2$). In the following sections, we will use $\boldsymbol{w}_1(t, j)$ or $\boldsymbol{w}_1(k, j)$ to represent the weights at time $t$ or time step $k$. The same notation also applies to $w_2, w_3$.

### 3.3 Assumptions

Our assumptions are rather standard in the mean-field literature and appear e.g. in (Mei et al., 2018; 2019; Nguyen & Pham, 2020; Pham & Nguyen, 2021a). We write such assumptions separately for networks with two and three layers.

**Assumption 3.1.** *We make the following assumptions for the training of a two-layer network:*

(A1) *(Boundedness) There exists a universal constant $K > 0$ such that $\|\sigma\|_\infty, \|\sigma'\|_\infty, \|\sigma''\|_\infty \le K$. The data distribution $\mathcal{D}$ is such that, almost surely, $|y|, \|\boldsymbol{x}\|_2 \le K$. Furthermore, $|\partial_2 R(y, f(\boldsymbol{x}; \boldsymbol{W}))|$ is $K$-Lipschitz continuous in $f(\boldsymbol{x}; \boldsymbol{W})$ and $K$-bounded for any $\boldsymbol{W}$.*

(A2) *(Initialization of weights) At initialization, $\boldsymbol{w}_1(0, j), w_2(0, j) \overset{i.i.d.}{\sim} \rho_0$, where $\rho_0$ is such that $\boldsymbol{w}_1(0, j)$ is $K^2$-sub-Gaussian, and $|w_2(0, j)| \le K$ almost surely.*

(A3) *(Initialization of momentum) We assume that $\boldsymbol{r}(0) = \boldsymbol{0}$.*

In order to state the assumption for a three-layer neural network, we first define the notion of i.i.d. initialization.

**Definition 3.2.** *We say that the three-layer neural network (2) has i.i.d. initialization from $\rho_0^1 \times \rho_0^2 \times \rho_0^3$, if:*

$$w_1(0, j_1) \overset{i.i.d.}{\sim} \rho_0^1, \quad w_2(0, j_1, j_2) \overset{i.i.d.}{\sim} \rho_0^2, \quad w_3(0, j_2) \overset{i.i.d.}{\sim} \rho_0^3, \quad j_1 \in [n_1], \quad j_2 \in [n_2]. \tag{10}$$

In short, i.i.d. initialization means both cross-layer and in-layer independence.

**Assumption 3.3.** *We make the following assumptions for the training of a three-layer network:*

*(B1) (Boundedness) There exists a universal constant $K > 0$ such that $\|\sigma_1\|_\infty$, $\|\sigma_1'\|_\infty$, $\|\sigma_1''\|_\infty$, $\|\sigma_2\|_\infty$, $\|\sigma_2'\|_\infty$, $\|\sigma_2''\|_\infty \leq K$. The data distribution $\mathcal{D}$ is such that $|y|, \|\boldsymbol{x}\|_2 \leq K$ almost surely. Furthermore, $\sigma_2'(x) \neq 0$ for all $x$, $|\partial_2 R(y, f(\boldsymbol{x}; \boldsymbol{W}))|$ is $K$-Lipschitz continuous with respect to the second argument and $K$-bounded for any $\boldsymbol{W}$.*

*(B2) (Initialization of weights) $\boldsymbol{w}_1(0, j_1), w_2(0, j_1, j_2), w_3(0, j_2)$ have an i.i.d. initialization from $\rho_0^1 \times \rho_0^2 \times \rho_0^3$. Furthermore, $\boldsymbol{w}_1(0, j_1)$ is $K^2$-sub-Gaussian, and $|w_2(0, j_1, j_2)|, |w_3(0, j_2)| \leq K$ almost surely.*

*(B3) (Initialization of momentum) We assume that $\boldsymbol{r}(0) = \boldsymbol{0}$.*

In the initialization of two-layer networks, $\rho_0$ is not required to be a product measure, that is, $\boldsymbol{w}_1(0, j), w_2(0, j)$ are not required to be independent from each other. In other words, we do not assume cross-layer independence, but only in-layer independence. However, in the three-layer case, we need to assume both cross-layer independence and in-layer independence, and it turns out later that the cross-layer independence is critical for proving our global convergence result.

The requirements above hold e.g. for tanh or sigmoid activation function, and logistic or Huber loss. The assumption that $\partial_2 R(y, f(\boldsymbol{x}; \boldsymbol{W}))$ is $K$-bounded does not hold for the square loss. However, we expect the same results proved in this paper to hold also for the square loss, provided that the assumptions are modified as in Mei et al. (2019) (for the two-layer case) and in Nguyen & Pham (2020) (for the three-layer case). In fact, it suffices that $\partial_2 R(y, f(\boldsymbol{x}; \boldsymbol{W}))$ is bounded with high probability, and then the arguments are similar.

Finally, we remark that the boundedness of the initialization $w_2(0, j)$ and $w_2(0, j_1, j_2), w_3(0, j_2)$ is purely to simplify the proof, which can be generalized to a $K^2$-sub-Gaussian initialization. In particular, one can bound the sub-Gaussian norm of the weights, and then the absolute value of the weights will also be bounded with high probability.

## 4 Derivation of the mean-field limit

**Two-layer networks.** The idea is that the output of the network can be viewed as an expectation over the empirical distribution of the weights, that is:

$$f(\boldsymbol{x}; \boldsymbol{W}) = \frac{1}{n} \sum_{j=1}^n w_2(j) \sigma_1(\boldsymbol{w}_1(j)^T \boldsymbol{x}) = \mathbb{E}_{\boldsymbol{\theta} \sim \hat{\rho}_{\boldsymbol{\theta}}} \sigma^\star(\boldsymbol{x}; \boldsymbol{\theta}),$$

where $\boldsymbol{W} = \{\boldsymbol{\theta}(j) : j \in [n]\}$ and $\boldsymbol{\theta}(j) = (\boldsymbol{w}_1(j), w_2(j))$. Furthermore, we define $\sigma^\star(\boldsymbol{x}; \boldsymbol{\theta}(j)) = w_2(j)\sigma(\boldsymbol{w}_1(j)^T \boldsymbol{x})$ and $\widehat{\rho}_{\boldsymbol{\theta}} = \frac{1}{n} \sum_{j=1}^n \delta_{\boldsymbol{\theta}(j)}$. Thus, the evolution of the parameters $\boldsymbol{\theta}(t)$ according to (5) can be viewed as the evolution of $\widehat{\rho}_{\boldsymbol{\theta}}(t)$ according to a certain distributional dynamics induced by (5).

Since we assume i.i.d. initialization, as the number of neurons $n \to \infty$, we expect that $\widehat{\rho}_{\boldsymbol{\theta}}(0) \to \rho_0$. In this limit, the distributional dynamics induced by (5), can be described by a certain ODE, with initial condition $\rho_0$. Let

$$f(\boldsymbol{x}; \rho) := \mathbb{E}_{\boldsymbol{\theta} \sim \rho} \sigma^\star(\boldsymbol{x}; \boldsymbol{\theta}), \qquad R(\boldsymbol{z}; \rho) := R(y, f(\boldsymbol{x}; \rho)), \qquad R(\rho) := \mathbb{E}_{\boldsymbol{z}} R(y, f(\boldsymbol{x}; \rho)),$$
$$\widehat{\Psi}(\boldsymbol{z}, \boldsymbol{\theta}; \rho) := \frac{\delta R(\boldsymbol{z}, \rho)}{\delta \rho}(\boldsymbol{\theta}), \qquad \Psi(\boldsymbol{\theta}; \rho) := \frac{\delta R(\rho)}{\delta \rho}(\boldsymbol{\theta}) = \mathbb{E}_{\boldsymbol{z}} \widehat{\Psi}(\boldsymbol{z}, \boldsymbol{\theta}; \rho). \tag{11}$$

Here, $f(\cdot; \rho) : \mathbb{R}^D \to \mathbb{R}$; $R(\cdot; \rho) : \mathbb{R}^{D+1} \to \mathbb{R}$, $R(\cdot) : \mathcal{P}_2(\mathbb{R}^{D+1}) \to \mathbb{R}$, where $\mathcal{P}_2(\mathbb{R}^{D+1})$ denotes the space of probability measures on $\mathbb{R}^{D+1}$ with finite second moment; $\hat{\Psi}(\cdot, \cdot; \rho) : \mathbb{R}^{D+1} \times \mathbb{R}^{D+1} \to \mathbb{R}$; $\Psi(\cdot; \rho) : \mathbb{R}^{D+1} \to \mathbb{R}$.

Then, we define the mean-field ODE associated to the heavy ball method as

$$d\boldsymbol{\theta}(t) = \boldsymbol{r}(t)dt, \qquad d\boldsymbol{r}(t) = \left(-\gamma \boldsymbol{r}(t) - \nabla_{\boldsymbol{\theta}} \Psi(\boldsymbol{\theta}(t); \rho^{\boldsymbol{\theta}}(t))\right) dt. \tag{12}$$

**Three-layer networks.** Among the various approaches to define a mean-field limit for multi-layer networks, we follow the "neuronal embedding" framework proposed in (Nguyen & Pham, 2020; Pham & Nguyen, 2021a)

to capture the dynamics of SGD training. We briefly summarize the key ideas, and then discuss how to obtain the mean-field limit for the stochastic heavy ball method.

Consider a three-layer neural network of the form (2) with weights $\boldsymbol{w}_1(0, j_1), w_2(0, j_1, j_2), w_3(0, j_2)$ obtained via an i.i.d. initialization from $\rho_1 \times \rho_2 \times \rho_3$, according to Definition 3.2. Then, in (Pham & Nguyen, 2021a, Proposition 7), it is proved that there exists a product probability space $(\Omega_1 \times \Omega_2, \mathcal{F}_1 \times \mathcal{F}_2, \mathbb{P}_1 \times \mathbb{P}_2)$ and functions $\boldsymbol{w}_1(0, \cdot) : \Omega_1 \to \mathbb{R}^D$, $w_2(0, \cdot, \cdot) : \Omega_1 \times \Omega_2 \to \mathbb{R}$, $w_3(0, \cdot) : \Omega_2 \to \mathbb{R}$ such that, for any $n_1, n_2$,

$$\{\boldsymbol{w}_1(0, C_1(j_1)), w_2(0, C_1(j_1), C_2(j_2)), w_3(0, C_2(j_2)), \text{ for } j_1 \in [n_1], j_2 \in [n_2]\}$$
$$\stackrel{d}{=} \{\boldsymbol{w}_1(0, j_1), w_2(0, j_1, j_2), w_3(0, j_2), \text{ for } j_1 \in [n_1], j_2 \in [n_2]\},$$

where $\stackrel{d}{=}$ denotes equality in distribution, $C_1(j_1) \stackrel{i.i.d.}{\sim} \mathbb{P}_1$, $C_2(j_2) \stackrel{i.i.d.}{\sim} \mathbb{P}_2$, for $j_1 \in [n_1], j_2 \in [n_2]$ . Here, the tuple $\{(\Omega_1 \times \Omega_2, \mathcal{F}_1 \times \mathcal{F}_2, \mathbb{P}_1 \times \mathbb{P}_2), w_1(0, \cdot), w_2(0, \cdot, \cdot), w_3(0, \cdot)\}$ is called a *neuronal embedding*. With a slight abuse of notation, we use $\boldsymbol{w}_1, w_2, w_3$ to denote also the functions $\boldsymbol{w}_1(0, \cdot)$, $w_2(0, \cdot, \cdot)$, $w_3(0, \cdot)$. In later sections, we refer to the functions when we write $\boldsymbol{w}_1(0, c_1)$, $w_2(0, c_1, c_2)$, $w_3(0, c_2)$, while we refer to the weights of the neural network when we write $\boldsymbol{w}_1(0, j_1)$, $w_2(0, j_1, j_2)$, $w_3(0, j_2)$.

At this point, we are ready to define the mean-field ODE for the heavy ball method. This ODE tracks the functions $\boldsymbol{w}_1(0, \cdot), w_2(0, \cdot, \cdot), w_3(0, \cdot)$:

$$\begin{aligned} dw_3(t, c_2) &= r_3(t, c_2)\, dt, & dr_3(t, c_2) &= (-\gamma r_3(t, c_2) - \mathbb{E}_{\boldsymbol{z}} \Delta_3^W(\boldsymbol{z}, c_2; \boldsymbol{W}(t)))\, dt, \\ dw_2(t, c_1, c_2) &= r_2(t, c_1, c_2)\, dt, & dr_2(t, c_1, c_2) &= (-\gamma r_2(t, c_1, c_2) - \mathbb{E}_{\boldsymbol{z}} \Delta_2^W(\boldsymbol{z}, c_1, c_2; \boldsymbol{W}(t)))dt, \\ d\boldsymbol{w}_1(t, c_1) &= \boldsymbol{r}_1(t, c_1)\, dt, & d\boldsymbol{r}_1(t, c_1) &= (-\gamma \boldsymbol{r}_1(t, c_1) - \mathbb{E}_{\boldsymbol{z}} \Delta_1^W(\boldsymbol{z}, c_1; \boldsymbol{W}(t)))\, dt, \end{aligned} \quad (13)$$

where $c_1 \in \Omega_1, c_2 \in \Omega_2$ are dummy variables and $\boldsymbol{W}(t)$ refers to the collection of weights $(\boldsymbol{w}_1(t), w_2(t), w_3(t))$. The output of the neural network under the mean-field limit (13) is described via the following forward pass:

$$\begin{aligned} H_1(\boldsymbol{x}, c_1; \boldsymbol{W}(t)) &= \boldsymbol{w}_1(t, c_1)^T \boldsymbol{x}, \quad c_1 \in \Omega_1, \\ H_2(\boldsymbol{x}, c_2; \boldsymbol{W}(t)) &= \mathbb{E}_{C_1 \sim \mathbb{P}_1} w_2(t, C_1, c_2) \sigma_1(H_1(\boldsymbol{x}, C_1; \boldsymbol{W}(t))), \quad c_2 \in \Omega_2, \\ f(\boldsymbol{x}; \boldsymbol{W}(t)) &= \mathbb{E}_{C_2 \sim \mathbb{P}_2} w_3(t, C_2) \sigma_2(H_2(\boldsymbol{x}, C_2; \boldsymbol{W}(t))). \end{aligned} \quad (14)$$

Furthermore, the quantities $\Delta_3^W, \Delta_2^W, \Delta_1^W$ appearing in (13) are described via the backward pass:

$$\begin{aligned} \Delta_3^W(\boldsymbol{z}, c_2; \boldsymbol{W}(t)) &:= \partial_2 R(y; f(\boldsymbol{x}; \boldsymbol{W}(t))) \sigma_2(H_2(\boldsymbol{x}, c_2; \boldsymbol{W}(t))), \\ \Delta_2^H(\boldsymbol{z}, c_2; \boldsymbol{W}(t)) &:= \partial_2 R(y; f(\boldsymbol{x}; \boldsymbol{W}(t))) w_3(t, c_2) \sigma_2'(H_2(\boldsymbol{x}, c_2; \boldsymbol{W}(t))), \\ \Delta_2^W(\boldsymbol{z}, c_1, c_2; \boldsymbol{W}(t)) &:= \Delta_2^H(\boldsymbol{z}, c_2; \boldsymbol{W}(t)) \sigma_1(H_1(\boldsymbol{x}, c_1; \boldsymbol{W}(t))), \\ \Delta_1^H(\boldsymbol{z}, c_1; \boldsymbol{W}(t)) &:= \mathbb{E}_{C_2} \Delta_2^H(\boldsymbol{x}, C_2; \boldsymbol{W}(t)) w_2(t, c_1, C_2) \sigma_1'(H_1(\boldsymbol{x}, c_1; \boldsymbol{W}(t))), \\ \Delta_1^W(\boldsymbol{z}, c_1; \boldsymbol{W}(t)) &:= \Delta_1^H(\boldsymbol{x}, c_1; \boldsymbol{W}(t)) \boldsymbol{x}. \end{aligned} \quad (15)$$

For convenience, we will also use the lighter notation $H_1(t, \boldsymbol{x}, c_1)$, $H_2(t, \boldsymbol{x}, c_2)$, $\Delta_3^W(t, \boldsymbol{z}, c_2)$, $\Delta_2^H(t, \boldsymbol{z}, c_2)$, $\Delta_2^W(t, \boldsymbol{z}, c_1, c_2)$, $\Delta_1^H(t, \boldsymbol{z}, c_1)$, $\Delta_1^W(t, \boldsymbol{z}, c_1)$ to denote the quantities $H_1(\boldsymbol{x}, c_1; \boldsymbol{W}(t))$, $H_2(\boldsymbol{x}, c_1; \boldsymbol{W}(t))$, $\Delta_3^W(\boldsymbol{z}, c_2; \boldsymbol{W}(t))$, $\Delta_2^H(\boldsymbol{z}, c_2; \boldsymbol{W}(t))$, $\Delta_2^W(\boldsymbol{z}, c_1, c_2; \boldsymbol{W}(t))$, $\Delta_1^H(\boldsymbol{z}, c_1; \boldsymbol{W}(t))$, $\Delta_1^W(\boldsymbol{z}, c_1; \boldsymbol{W}(t))$, respectively.

We note that the neuronal embedding framework can recover the distributional dynamics for two-layer networks as a special case (see (Nguyen & Pham, 2020, Corollary 22) for more details). It is also possible to define the mean-field limit for three-layer neural networks directly as a distributional dynamics (Araújo et al., 2019; Sirignano & Spiliopoulos, 2020a), although this approach may require additional assumptions (namely, first and last layer not trained, see Araújo et al. (2019)).

At this point, we are ready to prove the existence and uniqueness of the mean-field differential equations.

**Theorem 4.1.** *For any $t < \infty$, we have: (i) Under Assumption 3.1, there exists a unique solution of the mean-field PDE (12).*
*(ii) Under Assumption 3.3, there exists a unique solution of the mean-field ODE (13).*

The proof of Theorem 4.1 follows from the analysis of a Picard type of iteration (Sznitman, 1991), and it is deferred to Appendix B. We note that Krichene et al. (2020) consider a mean-field limit for two-layer networks in which an additional Brownian motion is applied to the momentum term. This extra noise term – together with the additional assumption (A5) (see p. 8 of Krichene et al. (2020)) – allows them to prove the existence and uniqueness of the mean-field limit for any $t \in [0, \infty]$. This includes the existence and uniqueness of the limiting point (for $t = \infty$). In contrast, Theorem 4.1 proves the existence and uniqueness of the solution for any finite $t$, and we do not have guarantees on the limiting point.

## 5 Convergence to the mean-field limit

### 5.1 Two-layer networks

We recall that the mean-field ODE is defined in (12), and the SHB dynamics can be expressed as

$$\boldsymbol{\theta}^{SHB}(k+1, j) = \boldsymbol{\theta}^{SHB}(k, j) + (1 - \gamma\varepsilon)(\boldsymbol{\theta}^{SHB}(k, j) - \boldsymbol{\theta}^{SHB}(k-1, j)) - \varepsilon^2 \nabla_{\boldsymbol{\theta}} \widehat{\Psi}(\boldsymbol{z}(k), \boldsymbol{\theta}^{SHB}(k, j); \rho^{\boldsymbol{\theta}}_{SHB}(k)),$$

$$(16)$$

where $\boldsymbol{\theta}^{SHB}(k, j)$ denotes the parameter associated to the $j$-th neuron at step $k$, $\widehat{\Psi}$ is defined in (11), and $\rho^{\boldsymbol{\theta}}_{SHB}(k) = \frac{1}{n} \sum_{j=1}^{n} \delta_{\boldsymbol{\theta}^{SHB}(k,j)}$ denotes the empirical distribution of the parameters $\{\boldsymbol{\theta}^{SHB}(k, j)\}_{j \in [n]}$. We couple the mean-field ODE (12) and the SHB dynamics (16), in the sense that they share the same initialization: $\boldsymbol{\theta}(0) \sim \rho^{\boldsymbol{\theta}}(0)$ and $\boldsymbol{\theta}^{SHB}(0, j) \overset{i.i.d.}{\sim} \rho^{\boldsymbol{\theta}}(0)$. Let us define the following distance metric that measures the difference between the mean-field dynamics and the SHB dynamics:

$$\mathcal{D}_T(\boldsymbol{\theta}, \boldsymbol{\theta}^{SHB}) = \max_{j \in [n]} \sup_{t \in [0,T]} \|\boldsymbol{\theta}^{SHB}(\lfloor t/\varepsilon \rfloor, j) - \boldsymbol{\theta}(t)\|_2. \tag{17}$$

**Theorem 5.1.** *Let Assumption 3.1 hold. Consider the mean-field ODE (12), the SHB dynamics (16) and the distance metric (17). Then, with probability at least $1 - \exp(-\delta^2)$,*

$$\mathcal{D}_T(\boldsymbol{\theta}, \boldsymbol{\theta}^{SHB}) \leq K(\gamma, T) \left( \frac{(\sqrt{\log n} + \delta)}{\sqrt{n}} + \sqrt{\varepsilon}(\sqrt{D + \log n} + \delta) \right), \tag{18}$$

*where $K(\gamma, T)$ is a constant depending only on $\gamma, T$.*

We remark that the RHS of (18) is also an upper bound on $\sup_{t \in [0,T]} \mathcal{W}_2(\rho^{\boldsymbol{\theta}}(t), \rho^{\boldsymbol{\theta}}_{SHB}(\lfloor t/\varepsilon \rfloor))$, which follows directly from the definition of the Wasserstein $\mathcal{W}_2$ distance.

Theorem 5.1 gives a quantitative characterization of the approximation error between the mean-field limit and the stochastic heavy ball dynamics. In particular, it shows that this approximation error scales roughly as $\sqrt{\log n/n} + \sqrt{\varepsilon(D + \log n)}$, i.e., it vanishes as the number of neurons $n$ grows large and the step size $\varepsilon = o(1/\sqrt{D + \log n})$. We remark that the bound in (18) is *dimension-free* in the sense that $n$ does not need to scale with the input dimension $D$. This is aligned with the bound provided for SGD by Mei et al. (2019) which is also dimension-free. Note that the order of the upper bound $O(\frac{1}{\sqrt{n}})$ is tight due to large deviation theory. This implies that it is not possible to improve the corresponding dropout stability and connectivity guarantees (see Section 6), in terms of the number of neurons/input dimension. This means that the behavior of SGD and SHB is similar, as both algorithms are optimal in this regard. The constant $K(\gamma, T)$ scales rather poorly in $T$, i.e., $K(\gamma, T) = O(e^{e^T})$. This is a common shortcoming for "propagation of chaos"-type arguments: for example, in Mei et al. (2018; 2019), the scaling of the bound in $T$ is $O(e^T)$. An interesting open problem is to improve such dependence, e.g., by using ideas from Schuh (2022).

To the best of our knowledge, Theorem 5.1 provides the first consistency guarantee of the mean-field limit (12). In the noisy case, a similar (although non-quantitative) guarantee is proved by Krichene et al. (2020). The injection of noise in the dynamics often simplifies the analysis and it allows to prove stronger results

(e.g., existence and uniqueness of the limit). However, the noiseless dynamics is particularly interesting, since Brownian noise is typically *not* added in practice while training.

At the technical level, in order to deal with the second order dynamics arising from the heavy ball method, we exploit a second order Gronwall's lemma (cf. Lemma F.3) and use the Euler scheme to discretize the continuous dynamics. The detailed proof is provided in Appendix C.

### 5.2 Three-layer networks

We recall that the mean-field ODE is defined in (13), and the SHB dynamics can be expressed as

$$
w_3^{SHB}(k+1, j_2) = w_3^{SHB}(0, j_2) + (1 - \gamma\varepsilon)(w_3^{SHB}(k, j_2) - w_3^{SHB}(k-1, j_2)) - \varepsilon^2 \Delta_3^W(\boldsymbol{z}(k), j_2; \boldsymbol{W}^{SHB}(k)),
$$
$$
w_2^{SHB}(k+1, j_1, j_2) = w_2^{SHB}(0, j_1, j_2) + (1 - \gamma\varepsilon)(w_2^{SHB}(k, j_1, j_2) - w_2^{SHB}(k-1, j_1, j_2))
$$
$$
- \varepsilon^2 \Delta_2^W(\boldsymbol{z}(k), j_1, j_2; \boldsymbol{W}^{SHB}(k)),
$$
$$
\boldsymbol{w}_1^{SHB}(k+1, j_1) = \boldsymbol{w}_1^{SHB}(0, j_1) + (1 - \gamma\varepsilon)(\boldsymbol{w}_1^{SHB}(k, j_1) - \boldsymbol{w}_1^{SHB}(k-1, j_1)) - \varepsilon^2 \Delta_1^W(\boldsymbol{z}(k), j_1; \boldsymbol{W}^{SHB}(k)),
$$
(19)

where $\boldsymbol{W}^{SHB}(k) = \left((\boldsymbol{w}_1^{SHB}(k, j_1))_{j_1 \in [n_1]}, (w_2^{SHB}(k, j_1, j_2))_{j_1 \in [n_1], j_2 \in [n_2]}, (w_3^{SHB}(k, j_2))_{j_2 \in [n_2]}\right)$, $\boldsymbol{z}(k)$ is the data point sampled at time step $k$, and $\Delta_1^W, \Delta_2^W, \Delta_3^W$ are defined in (9).

Before stating our result, let us discuss how to couple the SHB dynamics (19) and the mean-field ODE (13). First, sample a finite neural network w.r.t. the neuronal embedding, i.e., $C_1(j_1) \overset{i.i.d.}{\sim} \mathbb{P}_1$, $C_2(j_2) \overset{i.i.d.}{\sim} \mathbb{P}_2$, for $j_1 \in [n_1], j_2 \in [n_2]$ and $\boldsymbol{w}_1(0, \cdot), w_2(0, \cdot, \cdot), w_3(0, \cdot)$. Given $\boldsymbol{w}_1(0, \cdot), w_2(0, \cdot, \cdot), w_3(0, \cdot)$, let the mean-field ODE (13) evolve, thus obtaining $\boldsymbol{w}_1(t, \cdot), w_2(t, \cdot, \cdot), w_3(t, \cdot)$. Next, initialize the weights corresponding to the SHB evolution according to the initialization of the mean-field ODE, i.e., $\boldsymbol{w}_1^{SHB}(0, j_1) = \boldsymbol{w}_1(0, C_1(j_1))$, $w_2^{SHB}(0, j_1, j_2) = w_2(0, C_1(j_1), C_2(j_2))$ and $w_3^{SHB}(0, j_2) = w_3(0, C_2(j_2))$, and let them evolve according to SHB dynamics (19), thus obtaining $\boldsymbol{w}_1^{SHB}(k, j_1), w_2^{SHB}(k, j_1, j_2), w_3^{SHB}(k, j_2)$. Finally, we define the following distance metric that measures the difference between the mean-field and the SHB dynamics:

$$
\mathcal{D}_T(\boldsymbol{W}, \boldsymbol{W}^{SHB}) = \max_{j_1 \in [n_1], j_2 \in [n_2]} \sup_{t \in [0,T]} \max\{\|\boldsymbol{w}_1(t, C_1(j_1)) - \boldsymbol{w}_1^{SHB}(\lfloor t/\varepsilon \rfloor, j_1)\|_2,
$$
$$
|w_2(t, C_1(j_1), C_2(j_2)) - w_2^{SHB}(\lfloor t/\varepsilon \rfloor, j_1, j_2)|,
$$
$$
|w_3(t, C_2(j_2)) - w_3^{SHB}(\lfloor t/\varepsilon \rfloor, j_2)|\}.
$$
(20)

**Theorem 5.2.** *Let Assumption 3.3 hold. Consider the coupled the SHB dynamics (19) and mean-field ODE (13), and the distance metric (20). Then, with probability at least $1 - \exp(-\delta^2)$,*

$$
\mathcal{D}_T(\boldsymbol{W}, \boldsymbol{W}^{SHB}) \le K(\gamma, T) \left(\frac{(\sqrt{\log n_{\max}} + \delta)}{\sqrt{n_{\min}}} + \sqrt{\varepsilon}(\sqrt{D + \log n_1 n_2} + \delta)\right),
$$
(21)

*where $n_{\max} = \max\{n_1, n_2\}$, $n_{\min} = \min\{n_1, n_2\}$, and $K(\gamma, T)$ is a constant depending only on $\gamma, T$.*

For three layers, Theorem 5.2 gives that the approximation error scales roughly as $\sqrt{\log n_{\max}/n_{\min}} + \sqrt{\varepsilon(D + \log n_1 n_2)}$, i.e., it vanishes as long as $n_1, n_2$ both grow large, with $n_{\max} = o(e^{n_{\min}})$ and $\varepsilon = o(1/\sqrt{D + \log n_1 n_2})$. As in the two-layer case, the bound is dimension-free (in the sense that $n_1, n_2$ do not need to scale with $D$), and $K(\gamma, T) = O(e^{e^T})$. We remark that the scaling of the bound in $T$ is also a shortcoming of the existing analysis for SGD in (Pham & Nguyen, 2021a). The detailed proof is provided in Appendix D.

## 6 Consequences of the mean-field analysis: Dropout stability and connectivity

The mean-field perspective put forward in this paper leads to a precise characterization of the SHB training dynamics. In particular, Theorems 5.1-5.2 offer a provable justification to two remarkable properties exhibited by solutions obtained via gradient-based methods, namely, *dropout-stability* and *connectivity*. The

fact that solutions (often resulting from algorithms that use momentum) can be connected via simple paths with low loss was empirically observed in (Garipov et al., 2018; Draxler et al., 2018), and this property was related to dropout-stability by Kuditipudi et al. (2019). In (Shevchenko & Mondelli, 2020), it was shown that SGD solutions enjoy dropout-stability and connectivity and, by combining this analysis with Theorems 5.1-5.2, similarly strong guarantees can be obtained for heavy ball methods. We will keep the discussion at an informal level, as the details are similar to those in (Shevchenko & Mondelli, 2020).

Let's start with the two-layer case. Given a non-empty set $A \subset [n]$, we say that $\boldsymbol{W}$ is $\epsilon_D$-dropout stable if

$$|R(\boldsymbol{W}) - R_{\mathrm{drop}}(\boldsymbol{W}; A)| \leq \epsilon_D, \tag{22}$$

where $R_{\mathrm{drop}}(\boldsymbol{W}; A)$ is obtained by replacing the two-layer network $f(\boldsymbol{x}; \boldsymbol{W})$ defined in (1) with the dropout network

$$f(\boldsymbol{x}; \boldsymbol{W}_A) = \frac{1}{|A|} \sum_{j \in A} w_2(j)\sigma(\boldsymbol{w}_1(j)^T \boldsymbol{x}), \tag{23}$$

where $\boldsymbol{W}_A = (\boldsymbol{w}_1(j), w_2(j))_{j \in A}$ and $|A|$ denotes the cardinality of the set $A$. Similarly, in the three-layer case, given two non-empty sets $A_1 \subset [n_1], A_2 \subset [n_2]$, the dropout network is given by

$$\begin{aligned}
H_2(\boldsymbol{x}, j_2; \boldsymbol{W}_{A_1, A_2}) &= \frac{1}{|A_1|} \sum_{j_1 \in A_1} w_2(j_1, j_2)\sigma_1(\boldsymbol{w}_1(j_1)^T \boldsymbol{x}), \\
f(\boldsymbol{x}; \boldsymbol{W}_{A_1, A_2}) &= \frac{1}{|A_2|} \sum_{j_2 \in A_2} w_3(j_2)\sigma_2(H_2(\boldsymbol{x}, j_2; \boldsymbol{W}_{A_1, A_2})),
\end{aligned} \tag{24}$$

and $\epsilon_D$-dropout stability is defined analogously. Furthermore, we say that two solutions $\boldsymbol{W}$ and $\boldsymbol{W}'$ are $\epsilon_C$-connected if there exists a continuous path in parameter space that starts at $\boldsymbol{W}$, ends at $\boldsymbol{W}'$ and along which the population risk is upper bounded by $\max\{R(\boldsymbol{W}), R(\boldsymbol{W}')\} + \epsilon_C$.

At this point, the quantitative convergence result to the mean-field limit provided by Theorem 5.1 and 5.2 leads to a quantitative bound on the dropout stability and connectivity of the solutions found by the stochastic heavy ball method. We remark that proving only the consistency of the mean-field limit, as done in (Krichene et al., 2020, Theorem 1), does not suffice to obtain such guarantees on the structure of the SHB solution. In particular, for two-layer networks, after $k \leq \lfloor T/\varepsilon \rfloor$ steps of the iteration (16), the resulting parameters are $\epsilon_D$-dropout stable and $\epsilon_C$-connected, where

$$\begin{aligned}
\epsilon_D &\leq K(\gamma, T)\left(\frac{\left(\sqrt{\log|A|} + \delta\right)}{\sqrt{|A|}} + \sqrt{\varepsilon}(\sqrt{D + \log n} + \delta)\right), \\
\epsilon_C &\leq K(\gamma, T)\left(\frac{\left(\sqrt{\log n} + \delta\right)}{\sqrt{n}} + \sqrt{\varepsilon}(\sqrt{D + \log n} + \delta)\right),
\end{aligned} \tag{25}$$

with probability at least $1 - \exp(-\delta^2)$. Here $K(\gamma, T)$ is a universal constant depending only on $\gamma, T$ as before. Similarly, for three-layer networks, after $k \leq \lfloor T/\varepsilon \rfloor$ steps of the iteration (19), the resulting parameters are $\epsilon_D$-dropout stable and $\epsilon_C$-connected, where

$$\begin{aligned}
\epsilon_D &= K(\gamma, T)\left(\frac{\left(\sqrt{\log A_{\max}} + \delta\right)}{\sqrt{A_{\min}}} + \sqrt{\varepsilon}(\sqrt{D + \log n_1 n_2} + \delta)\right), \\
\epsilon_C &= K(\gamma, T)\left(\frac{\left(\sqrt{\log n_{\max}} + \delta\right)}{\sqrt{n_{\min}}} + \sqrt{\varepsilon}(\sqrt{D + \log n_1 n_2} + \delta)\right),
\end{aligned} \tag{26}$$

with probability at least $1 - \exp(-\delta^2)$. Here, $A_{\max} = \max\{|A_1|, |A_2|\}$, $A_{\min} = \min\{|A_1|, |A_2|\}$, $n_{\max} = \max\{n_1, n_2\}$ and $n_{\min} = \min\{n_1, n_2\}$. We remark that the path connecting the two solutions can be explicitly constructed as in (Kuditipudi et al., 2019; Shevchenko & Mondelli, 2020). More specifically, this

path is piece-wise linear, and the number of linear segments is a fixed constant. We also note that the bounds for multi-layer networks in (Shevchenko & Mondelli, 2020) exhibit a linear dependence on the input dimension $D$. In contrast, our bounds (25)-(26) are dimension-free.

Finally, let us highlight that our mean-field viewpoint can shed light on the thought-provoking conjecture in (Entezari et al., 2022), where it is empirically observed that, after a suitable permutation, the solutions of the optimization algorithm enjoy *linear* connectivity. In fact, Theorems 5.1 and 5.2 show that, by running the corresponding SHB training algorithm multiple times, all the resulting solutions satisfy (18) and (21), respectively. This implies that, after a permutation of the neurons, the distance between such solutions can also be upper bounded by the RHS of (18) and (21), hence the linear connectivity is an immediate consequence of this closeness among the solutions.

## 7 Global convergence of the mean-field ODE for three-layer networks

In order to show the global convergence result, we first need to make some extra assumptions.

**Assumption 7.1.** *We make the following additional assumptions for the training of a three-layer neural network:*

- *(C1) (Universal approximation property of the activation) $\sigma_1$ exhibits a universal approximation property, that is: $\{\sigma_1(\langle \boldsymbol{w}, \cdot \rangle) : \boldsymbol{w} \in \mathbb{R}^D\}$ has dense span in $\mathcal{L}^2(\mathcal{D}_{\boldsymbol{x}})$, where $\mathcal{D}_{\boldsymbol{x}}$ denotes the $\boldsymbol{x}$-marginal of the data distribution $\mathcal{D}$.*

- *(C2) (Full support at initialization) $\rho_0^1$ has full support.*

- *(C3) (Mode of convergence) The mean-field ODE (13) converges to the limit $\big(\boldsymbol{w}_1(\infty, c_1), w_2(\infty, c_1, c_2), w_3(\infty, c_2)\big)$. Formally, we have that*

$$\mathbb{E}_{C_1, C_2}[(1 + |w_3(\infty, C_2)|) \cdot |w_3(\infty, C_2)| \cdot |w_2(\infty, C_1, C_2)| \cdot \|\boldsymbol{w}_1(t, C_1) - \boldsymbol{w}_1(\infty, C_1)\|_2] \xrightarrow{t \to \infty} 0,$$

$$\mathbb{E}_{C_1, C_2}[(1 + |w_3(\infty, C_2)|) \cdot |w_3(\infty, C_2)| \cdot |w_2(t, C_1, C_2) - w_2(\infty, C_1, C_2)|] \xrightarrow{t \to \infty} 0,$$

$$\mathbb{E}_{C_2}[(1 + |w_3(\infty, C_2)|) \cdot |w_3(t, C_2) - w_3(\infty, C_2)|] \xrightarrow{t \to \infty} 0,$$

$$\operatorname*{ess\,sup}_{C_1} \mathbb{E}_{C_2}[|\mathbb{E}_{\boldsymbol{z}} \Delta_2^W(t, \boldsymbol{z}, C_1, C_2)|] \xrightarrow{t \to \infty} 0,$$

$$\Pr[w_3(\infty, C_2) \neq 0] > 0.$$

The universal approximation property is the key assumption to obtain a global convergence result. This requirement is mild, since most activation functions used in practice are universal approximators. The assumption on full support is also mild, since widely used initialized schemes (e.g., He's or LeCun's initialization) employ a Gaussian distribution, which indeed has full support. The assumption on the mode of convergence is purely technical, and it is an open question whether it can be relaxed. More specifically, part (C3) is needed because of the lack of entropic and moment regularization, which makes it difficult to characterize the limiting points of the noiseless mean-field dynamics. We note that the uniform convergence of the gradient (the fourth assumption in (C3)) could be replaced by Morse-Sard type of regularity assumptions (see (Nguyen & Pham, 2020, Section 8)). We remark that these requirements also appear in Pham & Nguyen (2021a), with the exception of $\Pr[w_3(\infty, C_2) \neq 0] > 0$, which is needed to handle the heavy ball dynamics.

**Theorem 7.2.** *Let Assumptions 3.3 and 7.1 hold, and assume further that $R(y, f(\boldsymbol{x}; \boldsymbol{W}))$ is convex in $f(\boldsymbol{x}; \boldsymbol{W})$. Let $\boldsymbol{W}(t)$ be the solution of the mean-field ODE (13). Then, we have that*

$$\lim_{t \to \infty} \mathbb{E}_{\boldsymbol{z}} R(y, f(\boldsymbol{x}; \boldsymbol{W}(t))) = \inf_{\hat{y}: \mathbb{R}^D \to \mathbb{R}} \mathbb{E}_{\boldsymbol{z}} R(y, \widehat{y}(\boldsymbol{x})). \tag{27}$$

The detailed proof is deferred to Appendix E, and we provide here a sketch. First, we show a degeneracy property for the mean-field ODE, i.e., there exist deterministic functions $\boldsymbol{w}_1^*(\cdot, \cdot) : \mathbb{R}^{\geq 0} \times \mathbb{R}^D \to \mathbb{R}^D, w_2^*(\cdot, \cdot, \cdot, \cdot) :$

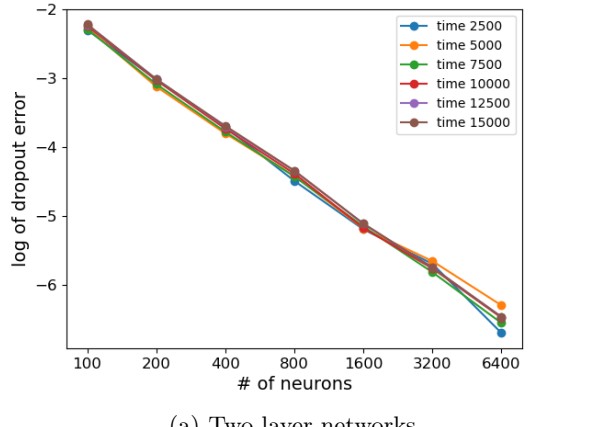
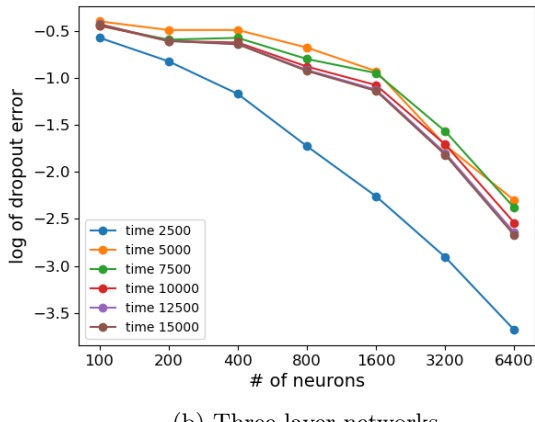

(a) Two-layer networks

(b) Three-layer networks

Figure 1: Dropout error plotted as a function of the network width. The log-log plot is close to be linear, matching the behavior of our theoretical predictions of Section 6.

$\mathbb{R}^{\geq 0} \times \mathbb{R}^D \times \mathbb{R} \times \mathbb{R} \to \mathbb{R}, w_3^*(\cdot, \cdot) : \mathbb{R}^{\geq 0} \times \mathbb{R} \to \mathbb{R}$ such that

$$\boldsymbol{w}_1(t, C_1) = \boldsymbol{w}_1^*(t, \boldsymbol{w}_1(0, C_1)),$$
$$w_2(t, C_1, C_2) = w_2^*(t, \boldsymbol{w}_1(0, C_1), w_2(0, C_1, C_2), w_3(0, C_2)),$$
$$w_3(t, C_2) = w_3^*(t, w_3(0, C_2)).$$

Next, we show that, for any finite $t$, $\boldsymbol{w}_1^*(\cdot, \cdot)$ is continuous in both arguments and $\boldsymbol{w}_1(t, C_1)$ has full support. Finally, the convergence to the global minimum is obtained by combining the argument that $\boldsymbol{w}_1(t, C_1)$ is full support for all finite $t$ with the assumption on the mode of convergence.

Theorem 7.2 is rather different from the global convergence result for the heavy ball method presented in (Krichene et al., 2020). In fact, Krichene et al. (2020) consider a *noisy* dynamics, and show the convergence of the mean-field ODE to the global minimum of a certain free energy, which represents an entropic regularization of the loss function. In this setup, the convergence is guaranteed by the noise term in the dynamics and by the regularization term in the free energy functional. In contrast, we consider a *noiseless* dynamics and do not prove its convergence. Instead, we show that, when the mean-field ODE converges, it must do so towards the global minimum of the un-regularized loss function. At the technical level, our proof strategy is an adaptation to the heavy ball case of the argument for SGD in (Pham & Nguyen, 2021a), which also crucially relies on the universal approximation property of the activation function. A similar idea was first proposed in (Lu et al., 2020), and it also appears e.g. in (Fang et al., 2021). However, our contribution is the first to tackle the case of optimization with momentum.

## 8 Numerical results

**Experimental setup.** We train a two-layer and a three-layer fully connected neural network in the mean-field regime on the MNIST dataset. The training algorithm is stochastic gradient descent with momentum, and we evaluate the dropout stability of the learnt models. For the two-layer network, we take the width $n = 100 \times 2^k$, where $k \in \{1, \dots, 7\}$; for the three-layer network, we take $n_1 = n_2 = n$ and use the same grid for $n$. We use the PyTorch default initialization, pick the learning rate $\varepsilon$ to be 0.05 and the momentum to be 0.9 (which implies that $\gamma = 2$). We rescale the learning rate, so that the scaling of the gradient does not depend on $n$, as required by our theory. The batch size is 100 and we train for 25 epochs, which means that the neural network is trained for 15000 steps (each epoch contain 600 steps and there are 25 epochs). For each model, we perform 10 i.i.d. experiments and report their average. We compute the population loss for both the original network and the dropout network, obtained by randomly dropping out half of the neurons. For each experiment, we take 10 random dropout networks and report the average population loss.

**Experimental results.** In Figure 1, we plot the log of the dropout error defined in (22) as a function of the number of neurons in each layer. Different curves correspond to different numbers of trained epochs. Two remarks are in order. First, the dependence of the dropout error on the time of the dynamics is rather mild (and, hence, our bounds on the constant $K(\gamma, T)$ appear to be pessimistic). Second, the dropout error scales as an inverse polynomial in the width, in agreement with (25)-(26).

## 9 Discussion and future direction

In this paper, we consider neural networks with two and three layers, and analyze the dynamics of stochastic gradient descent with momentum – also known as the stochastic heavy ball method (SHB) – from a mean-field viewpoint. After showing the existence and uniqueness of the mean-field limit, we provide a quantitative convergence result of the discrete SHB dynamics of a finite-width neural network to the corresponding limit differential equation, thus solving a problem raised by Krichene et al. (2020). Then, we exploit the power of our mean-field perspective by *(i)* proving that the solutions found by SHB enjoy desirable properties, such as dropout stability and connectivity, and *(ii)* showing a global convergence result for three-layer networks.

At the technical level, our proof strategies build on the work by Mei et al. (2019) and Pham & Nguyen (2021a) for neural networks with two and three layers, respectively. However, these papers focus on vanilla SGD, and dealing with SHB requires a number of delicate technical results. In particular, *(i)* we establish several boundedness and smoothness properties of the mean-field dynamics, *(ii)* we track various new quantities and exploit a second-order Gronwall lemma (Pachpatte's inequality) to bound them, *(iii)* we perform a discretization of the particle dynamics which is different from the SGD case, and *(iv)* we prove a key measurability property for the second-order dynamics, which characterizes the dependency between layers during training. Our strategy is tailored to the stochastic heavy ball method, but similar ideas could potentially be applied also to Nesterov's accelerated method, due to the similarity in the continuous limit. The study of other popular training algorithms (e.g., Adam) most likely requires an entirely different technical analysis, whose investigation is left as an interesting open problem.

Let us conclude by discussing some extensions and future directions.

**Beyond three layers and fully connected networks.** While it should be possible to extend our analysis of SGD with momentum to feed-forward networks with more than three layers, it remains unclear how to obtain global convergence guarantees in a more general setup. In fact, this is an open problem even for the case of optimization without momentum, see (Nguyen & Pham, 2020, Section 5). We also note that, although we only consider fully connected networks, our analysis could be generalized to other architectures, such as convolutional neural networks (CNNs) or ResNets, by slightly modifying the assumption. In particular, for CNNs, if we replace inner products by convolutions and let $n$ or $n_1, n_2$ be the number of filters in each layer, our analysis can be directly applied. For ResNets, the presence of the skip connection modifies the backward path, and the mean-field limit would need to reflect this modification.

**"Central limit theorem"-type of results.** Theorems 5.1-5.2 belong to a "law of large numbers"-type of results, in the sense that they show how close is the SHB dynamics to the mean-field limit. A parallel line of work has studied the distribution of the perturbation of the finite-width neural network around its mean-field limit, see (Chen et al., 2020; Sirignano & Spiliopoulos, 2020b) for two-layer networks, and (Pham & Nguyen, 2021b) for multi-layer ones. However, all the existing results concern the vanilla SGD algorithm, and understanding how momentum can affect such a distribution is an interesting future direction.

**Convergence of noisy dynamics.** In this work, we consider a noiseless dynamics, and the global convergence result for three-layer networks follows from the universality of the activation function. In contrast, in the noisy case, the mean-field limit for a two-layer network is an under-damped mean-field Langevin dynamics, whose convergence follows from the convexity of a related free-energy functional (Krichene et al., 2020). In a multi-layer setup, the free energy is not convex, which makes it challenging to obtain global convergence.

**Comparison between SGD and heavy ball methods.** Motivated by the observation that, in practice, adding momentum helps to generalize better (Sutskever et al., 2013), an exciting avenue for future research is to exploit the structure of the mean-field limit to draw insights concerning the solution found by heavy

ball methods. While the recent work by Jelassi & Li (2022) theoretically proved the improvements in generalization error provided by momentum under certain settings, we are not aware about whether such results hold in the mean-field training regime. A key difficulty here is that, for the noiseless dynamics, global convergence is not guaranteed in general, let alone an explicit characterization of the stationary solution. To circumvent this issue, one option may be to consider a suitably regularized noisy dynamics, which admits a stationary solution in a Gibbs form, in the limit of vanishingly small noise and regularization. Finally, heavy ball methods are known to enjoy faster rates of convergence in the convex setting. Thus, establishing a convergence rate for the mean-field dynamics is an exciting avenue for future research.

## Acknowledgements

D. Wu and M. Mondelli are partially supported by the 2019 Lopez-Loreta Prize. V. Kungurtsev was supported by the OP VVV project CZ.02.1.01/0.0/0.0/16_019/0000765 "Research Center for Informatics".

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

# A A-priori estimates

## A.1 Two-layer networks

**Lemma A.1.** *Assume that (A1) - (A3) hold, and let $f(\boldsymbol{x}; \rho), \Psi(\boldsymbol{\theta}; \rho), \nabla_{\boldsymbol{\theta}} \Psi(\boldsymbol{\theta}; \rho)$ be defined in (11). Then, for any fixed $T$, there exist universal constants $K, K_2(\gamma, T)$, where the latter depends only on $\gamma, T$, such that the following results hold.*

1. *(Boundedness) We have that, for any $\boldsymbol{\theta}, \rho$,*

$$
\begin{aligned}
f(\boldsymbol{x}; \rho) &\leq K \mathbb{E}_\rho |w_2|, \\
|\Psi(\boldsymbol{\theta}; \rho)| &\leq K |w_2|, \\
\|\nabla_{\boldsymbol{\theta}} \Psi(\boldsymbol{\theta}; \rho)\|_2 &\leq K(1 + |w_2|).
\end{aligned}
\tag{28}
$$

2. *(Boundedness for mean-field ODE) We have that, for any $t \leq T$, $w_2(t)$ as governed by (12) satisfies*

$$
|w_2(t)| \leq K_2(\gamma, T).
\tag{29}
$$

3. *(Lipschitz continuity):*

$$
|\Psi(\boldsymbol{\theta}; \rho) - \Psi(\boldsymbol{\theta}'; \rho')| \leq K(1 + |w_2|)\left(|w_2 - w_2'| + \|\boldsymbol{w}_1 - \boldsymbol{w}_1'\|_2 + \mathcal{W}_2(\rho, \rho')\right),
\tag{30}
$$

$$
\|\nabla_{\boldsymbol{\theta}} \Psi(\boldsymbol{\theta}; \rho) - \nabla_{\boldsymbol{\theta}} \Psi(\boldsymbol{\theta}'; \rho')\|_2 \leq K(1 + |w_2|)\left(|w_2 - w_2'| + \|\boldsymbol{w}_1 - \boldsymbol{w}_1'\|_2 + \mathcal{W}_2(\rho, \rho')\right).
\tag{31}
$$

*Proof.* 1. By the definition and assumption (A1), we have that

$$
\begin{aligned}
|f(\boldsymbol{x}; \rho)| &= |\mathbb{E}_\rho w_2 \sigma(\boldsymbol{w}_1^T \boldsymbol{x})| \leq K \mathbb{E}_\rho |w_2|, \\
|\Psi(\boldsymbol{\theta}; \rho)| &\leq |\mathbb{E}_{\boldsymbol{z}}\left[\partial_2 R(y, f(\boldsymbol{x}; \rho))\sigma(\boldsymbol{w}_1^T \boldsymbol{x})\right]| \cdot |w_2| \leq K |w_2|, \\
|\nabla_{w_2} \Psi(\boldsymbol{\theta}; \rho)| &= |\mathbb{E}_{\boldsymbol{z}}\left[\partial_2 R(y, f(\boldsymbol{x}; \rho))\sigma(\boldsymbol{w}_1^T \boldsymbol{x})\right]| \leq K, \\
\|\nabla_{\boldsymbol{w}_1} \Psi(\boldsymbol{\theta}; \rho)\|_2 &= \|\mathbb{E}_{\boldsymbol{z}}\left[\partial_2 R(y, f(\boldsymbol{x}; \rho))w_2 \sigma'(\boldsymbol{w}_1^T \boldsymbol{x})\boldsymbol{x}\right]\|_2 \leq K |w_2|.
\end{aligned}
$$

2. By writing down the integral form of the ODE, we have

$$|w_2(t)| \leq |w_2(0)| + \gamma \int_0^T (|w_2(s)| + |w_2(0)|) \, ds + \int_0^T \int_0^s |\nabla_{w_2} \Psi(\boldsymbol{\theta}(u); \rho(u))| \, du \, ds$$

$$\leq (K + KT + KT^2) + \gamma \int_0^T |w_2(s)| \, ds$$

$$\leq (K + KT + KT^2) e^{\gamma T}.$$

By setting $K_2(\gamma, T) := (K + KT + KT^2) e^{\gamma T}$, the proof of (29) is complete.

3. For the Lipschitz continuity argument, we have

$$\Psi(\boldsymbol{\theta}; \rho) = \mathbb{E}_{\boldsymbol{z}} \left[ \partial_2 R(y, f(\boldsymbol{x}; \rho)) w_2 \sigma(\boldsymbol{w}_1^T \boldsymbol{x}) \right],$$

$$\nabla_{\boldsymbol{\theta}} \Psi(\boldsymbol{\theta}; \rho) = \begin{pmatrix} \mathbb{E}_{\boldsymbol{z}} \left[ \partial_2 R(y, f(\boldsymbol{x}; \rho)) w_2 \sigma'(\boldsymbol{w}_1^T \boldsymbol{x}) \boldsymbol{x} \right] \\ \mathbb{E}_{\boldsymbol{z}} \left[ \partial_2 R(y, f(\boldsymbol{x}; \rho)) \sigma(\boldsymbol{w}_1^T \boldsymbol{x}) \right] \end{pmatrix}.$$

Thus,

$$|\Psi(\boldsymbol{\theta}; \rho) - \Psi(\boldsymbol{\theta}'; \rho')| \leq K|w_2| \|\boldsymbol{w}_1 - \boldsymbol{w}_1'\| + K|w_2 - w_2'| + K|w_2| |\mathbb{E}_{\boldsymbol{\theta} \sim \rho} \sigma(x; \boldsymbol{\theta}) - \mathbb{E}_{\boldsymbol{\theta} \sim \rho'} \sigma(x; \boldsymbol{\theta})|. \quad (32)$$

We define the Bounded Lipschitz (BL) divergence as follows:

$$d_{BL}(\rho, \rho') = \sup\{|\mathbb{E}_{\boldsymbol{\theta} \sim \rho} f(\boldsymbol{\theta}) - \mathbb{E}_{\boldsymbol{\theta} \sim \rho'} f(\boldsymbol{\theta})| : |f| \leq 1, \quad \|f\|_{\text{Lip}} \leq 1\}.$$

We have the following relationship between the BL-divergence and the Wasserstein distance (see for example (Chizat & Bach, 2018, Appendix A) for more details):

$$d_{BL}(\rho, \rho') \leq \mathcal{W}_2(\rho, \rho').$$

Hence,

$$|\mathbb{E}_\rho \sigma(x; \boldsymbol{\theta}) - \mathbb{E}_{\rho'} \sigma(x; \boldsymbol{\theta})| \leq K d_{BL}(\rho, \rho') \leq K \mathcal{W}_2(\rho, \rho'),$$

which implies that the RHS of (32) is upper bounded by

$$K|w_2|(\|\boldsymbol{w}_1 - \boldsymbol{w}_1'\|_2 + \mathcal{W}_2(\rho, \rho')) + K|w_2 - w_2'| \leq K(1 + |w_2|) \left( |w_2 - w_2'| + \|\boldsymbol{w}_1 - \boldsymbol{w}_1'\|_2 + \mathcal{W}_2(\rho, \rho') \right)$$

This concludes the proof of (30). The Lipschitz continuity of $\nabla_{\boldsymbol{\theta}} \Psi(\boldsymbol{\theta}; \rho)$ follows from the same argument.

$\square$

## A.2 Three-layer networks

**Lemma A.2.** *Assume that (B1)-(B2) hold, and let $H_2, f, \Delta_1^W, \Delta_2^W, \Delta_3^W, \Delta_1^H, \Delta_2^H$ be defined in (14) and (15). Then, for any fixed $T$, and given a neuronal embedding*

$$\{(\Omega_1 \times \Omega_2, \mathcal{F}_1 \times \mathcal{F}_2, \mathbb{P}_1 \times \mathbb{P}_2), w_1(0, \cdot), w_2(0, \cdot, \cdot), w_3(0, \cdot)\},$$

*there exists a universal constant $K$ and universal constants $K_{3,2}(\gamma, T)$, $K_{3,3}(\gamma, T)$ only depending on $\gamma, T$ such that the following results hold.*

1. *(Boundedness) We have that, for any $\boldsymbol{W}, z$, for any $t \in [0, T]$ and for any $c_1 \in \Omega_1, c_2 \in \Omega_2$,*

   - $|f(x; \boldsymbol{W}(t))| \leq K \operatorname{ess\,sup}_{C_2} |w_3(t, C_2)|$
   - $H_2(\boldsymbol{x}, c_2; \boldsymbol{W}(t))| \leq K \operatorname{ess\,sup}_{C_1, C_2} |w_2(t, C_1, C_2)|$
   - $|\Delta_3^W(\boldsymbol{z}, c_2; \boldsymbol{W}(t))| \leq K$

- $|\Delta_2^H(\boldsymbol{z}, c_2; \boldsymbol{W}(t))| \le K \operatorname{ess\,sup}_{C_2} |w_3(t, C_2)|$
- $|\Delta_2^W(\boldsymbol{z}, c_1, c_2; \boldsymbol{W}(t))| \le K \left( \operatorname{ess\,sup}_{C_1, C_2} |w_2(t, C_1, C_2)| \right) \operatorname{ess\,sup}_{C_2} |w_3(t, C_2)|$
- $|\Delta_1^H(\boldsymbol{x}, c_1; \boldsymbol{W}(t))| \le K \operatorname*{ess\,sup}_{C_2} |w_3(t, C_2)| \operatorname*{ess\,sup}_{C_1, C_2} |w_2(t, C_1, C_2)|$
- $\|\Delta_1^W(\boldsymbol{x}, c_1; \boldsymbol{W}(t))\|_2 \le K \operatorname{ess\,sup}_{C_2} |w_3(t, C_2)| \operatorname{ess\,sup}_{C_1, C_2} |w_2(t, C_1, C_2)|$

2. *(Boundedness for mean-field ODE)* We have that, for any $t \le T$,

$$\operatorname*{ess\,sup}_{C_2} |w_3(t, C_2)| \le K_{3,3}(\gamma, T), \quad \operatorname*{ess\,sup}_{C_1, C_2} |w_2(t, C_1, C_2)| \le K_{3,2}(\gamma, T). \tag{33}$$

3. *(Lipschitz continuity)* We have that, for any $t \le T$,

- $|H_1(\boldsymbol{x}, c_1; \boldsymbol{W}(t)) - H_1(\boldsymbol{x}, c_1; \tilde{\boldsymbol{W}}(t))| \le K \|\boldsymbol{w}_1(t, c_1) - \tilde{\boldsymbol{w}}_1(t, c_1)\|_2$
- $|H_2(\boldsymbol{x}, c_2; \boldsymbol{W}(t)) - H_2(\boldsymbol{x}, c_2; \tilde{\boldsymbol{W}}(t))|$
  $\le K \operatorname*{ess\,sup}_{C_1} (|w_2(t, C_1, c_2)| \|\boldsymbol{w}_1(t, C_1) - \tilde{\boldsymbol{w}}_1(t, C_1)\|_2 + |w_2(t, C_1, c_2) - \tilde{w}_2(t, C_1, c_2)|)$
- $|f(\boldsymbol{x}; \boldsymbol{W}(t)) - f(\boldsymbol{x}; \tilde{\boldsymbol{W}}(t))|$
  $\le K \operatorname*{ess\,sup}_{C_1, C_2}(|w_3(t, C_2)| \cdot |w_2(t, C_1, C_2)| \cdot \|\boldsymbol{w}_1(t, c_1) - \tilde{\boldsymbol{w}}_1(t, c_1)\|_2$
  $+ |w_3(t, c_2)| \cdot |w_2(t, c_1, c_2) - \tilde{w}_2(t, c_1, c_2)| + |w_3(t, c_2) - \tilde{w}_3(t, c_2)|)$
- $|\Delta_3^W(\boldsymbol{z}, c_2; \boldsymbol{W}(t)) - \Delta_3^W(\boldsymbol{z}, c_2; \tilde{\boldsymbol{W}}(t))|$
  $\le K \left( |H_2(\boldsymbol{x}, c_2; \boldsymbol{W}(t)) - H_2(\boldsymbol{x}, c_2; \tilde{\boldsymbol{W}}(t))| + |f(\boldsymbol{x}; \boldsymbol{W}(t)) - f(\boldsymbol{x}; \tilde{\boldsymbol{W}}(t))| \right)$
- $|\Delta_2^H(\boldsymbol{z}, c_2; \boldsymbol{W}(t)) - \Delta_2^H(\boldsymbol{z}, c_2; \tilde{\boldsymbol{W}}(t))|$
  $\le K|w_3(t, c_2)| \cdot |H_2(\boldsymbol{x}, c_2; \boldsymbol{W}(t)) - H_2(\boldsymbol{x}, c_2; \tilde{\boldsymbol{W}}(t))| + K|w_3(t, c_2) - \tilde{w}_3(t, c_2)|$
  $+ K|w_3(t, c_2)| \cdot |f(\boldsymbol{x}; \boldsymbol{W}(t)) - f(\boldsymbol{x}; \tilde{\boldsymbol{W}}(t))|$
- $|\Delta_2^W(\boldsymbol{z}, c_1, c_2; \boldsymbol{W}(t)) - \Delta_2^W(\boldsymbol{z}, c_1, c_2; \tilde{\boldsymbol{W}}(t))|$
  $\le K|\Delta_2^H(\boldsymbol{x}, c_2; \boldsymbol{W}(t)) - \Delta_2^H(\boldsymbol{z}, c_2; \tilde{\boldsymbol{W}}(t))|$
  $+ K|\Delta_2^H(\boldsymbol{z}, c_2; \boldsymbol{W}(t))| \cdot \|\boldsymbol{w}_1(t, c_1) - \tilde{\boldsymbol{w}}_1(t, c_1)\|_2$
- $\|\Delta_1^W(\boldsymbol{z}, c_1; \boldsymbol{W}(t)) - \Delta_1^W(\boldsymbol{z}, c_1; \tilde{\boldsymbol{W}}(t))\|_2$
  $\le K|\mathbb{E}_{C_2} \left[ \Delta_2^H(\boldsymbol{z}, C_2; \boldsymbol{W}(t)) w_2(t, c_1, C_2) \right]| \cdot \|\boldsymbol{w}_1(t, c_1) - \tilde{\boldsymbol{w}}_1(t, c_1)\|_2$
  $+ K|\mathbb{E}_{C_2} \left[ \Delta_2^H(\boldsymbol{z}, C_2; \tilde{\boldsymbol{W}}(t)) \tilde{w}_2(t, c_1, C_2) - \Delta_2^H(\boldsymbol{z}, C_2; \boldsymbol{W}(t)) w_2(t, c_1, C_2) \right]|.$

*Proof.*  1. By the definition and assumption (B1), we have

- $|f(x; \boldsymbol{W}(t))| = |\mathbb{E}_{C_2} w_3(t, C_2) \sigma_2(H_2(t, \boldsymbol{x}, C_2))|$
  $\le K|\mathbb{E}_{C_2} w_3(t, C_2)| \le \operatorname*{ess\,sup}_{C_2} |w_3(t, C_2)|$
- $|H_2(\boldsymbol{x}, c_2; \boldsymbol{W}(t))| = |\mathbb{E}_{C_1} w_2(t, C_1, c_2) \sigma_1(H_1(t, \boldsymbol{x}, C_1))|$
  $\le \operatorname*{ess\,sup}_{C_1, C_2} |w_2(t, C_1, C_2) \sigma_1(H_1(\boldsymbol{x}, C_1; \boldsymbol{W}(t)))|$
  $\le K \operatorname*{ess\,sup}_{C_1, C_2} |w_2(t, C_1, C_2)|$
- $|\Delta_3^W(\boldsymbol{x}, c_2; \boldsymbol{W}(t))| = |\partial_2 R(y; f(\boldsymbol{x}; \boldsymbol{W}(t))) \sigma_2(H_2(\boldsymbol{x}, c_2; \boldsymbol{W}(t)))| \le K$
- $|\Delta_2^H(\boldsymbol{x}, c_2; \boldsymbol{W}(t))| = |\partial_2 R(y; f(\boldsymbol{x}; \boldsymbol{W}(t))) w_3(t, c_2) \sigma_2'(H_2(\boldsymbol{x}, c_2; \boldsymbol{W}(t)))|$
  $\le \operatorname*{ess\,sup}_{C_2} |w_3(t, C_2) \sigma_2'(H_2(\boldsymbol{x}, C_2; \boldsymbol{W}(t)))| \le K \operatorname*{ess\,sup}_{C_2} |w_3(t, C_2)|$
- $|\Delta_2^W(\boldsymbol{x}, c_1, c_2; \boldsymbol{W}(t))| = |\Delta_2^H(\boldsymbol{x}, c_2; \boldsymbol{W}(t)) \sigma_2(H_1(\boldsymbol{x}, c_1; \boldsymbol{W}(t)))|$
  $\le K \operatorname*{ess\,sup}_{C_2} |\Delta_2^H(\boldsymbol{x}, C_2; \boldsymbol{W}(t))| \le K \operatorname*{ess\,sup}_{C_2} |w_3(t, C_2)|$

- $|\Delta_1^H(\boldsymbol{x}, c_1; \boldsymbol{W}(t))| = |\mathbb{E}_{C_2} \Delta_2^H(\boldsymbol{x}, C_2; \boldsymbol{W}(t)) w_2(t, c_1, C_2) \sigma_1'(H_1(\boldsymbol{x}, c_1; \boldsymbol{W}(t)))|$

$\leq \underset{C_2}{\mathrm{ess\,sup}} |\Delta_2^H(\boldsymbol{x}, C_2; \boldsymbol{W}(t))| \underset{C_1, C_2}{\mathrm{ess\,sup}} |w_2(t, C_1, C_2)| \underset{C_1}{\mathrm{ess\,sup}} |\sigma_1'(H_1(\boldsymbol{x}, C_1; \boldsymbol{W}(t)))|$

$\leq K \underset{C_2}{\mathrm{ess\,sup}} |w_3(t, C_2)| \underset{C_1, C_2}{\mathrm{ess\,sup}} |w_2(t, C_1, C_2)|$

- $\|\Delta_1^W(\boldsymbol{x}, c_1; \boldsymbol{W}(t))\|_2 = \|\Delta_1^H(\boldsymbol{x}, c_1; \boldsymbol{W}(t))\boldsymbol{x}\|_2 \leq \underset{C_1}{\mathrm{ess\,sup}} |\Delta_1^H(\boldsymbol{x}, C_1; \boldsymbol{W}(t))| \|\boldsymbol{x}\|_2$

$\leq K \underset{C_2}{\mathrm{ess\,sup}} |w_3(t, C_2)| \underset{C_1, C_2}{\mathrm{ess\,sup}} |w_2(t, C_1, C_2)|$

2. We have that, for any $t \leq T$,

$$|w_3(t, c_2)| \leq |w_3(0, c_2)| + \gamma \int_0^t (|w_3(0, c_2)| + |w_3(s, c_2)|)\, ds$$

$$+ \int_0^t \int_0^s |\mathbb{E}_{\boldsymbol{x}} \Delta_3^W(u, \boldsymbol{x}, c_2)|\, du\, ds$$

$$\leq K + K\gamma T + KT^2 + \gamma \int_0^t |w_3(s, c_2)|\, ds$$

$$\leq (K + K\gamma T + KT^2)e^{\gamma T} := K_{3,3}(\gamma, T),$$

which readily gives the first claim. Next, we write

$$|w_2(t, c_1, c_2)| \leq |w_2(0, c_1, c_2)| + \gamma \int_0^t (|w_2(0, c_1, c_2)| + |w_2(s, c_1, c_2)|)\, ds$$

$$+ \int_0^t \int_0^s |\mathbb{E}_{\boldsymbol{x}} \Delta_2^W(\boldsymbol{x}, c_1, c_2; \boldsymbol{W}(u))|\, du\, ds$$

$$\leq K + K\gamma T + \gamma \int_0^t |w_2(s, c_1, c_2)|\, ds + K_{3,3}(\gamma, T)T^2,$$

which by Gronwall's lemma, implies that

$$\underset{C_1, C_2}{\mathrm{ess\,sup}} |w_2(t, C_1, C_2)| \leq (K + K\gamma T + K_{3,3}(\gamma, T)T^2)e^{KT} := K_{3,2}(\gamma, T).$$

3. For the Lipschitz continuity argument, we have

- $|H_1(\boldsymbol{x}, c_1; \boldsymbol{W}(t)) - H_1(\boldsymbol{x}, c_1; \tilde{\boldsymbol{W}}(t))| = |\boldsymbol{x}^T(\boldsymbol{w}_1(t, c_1) - \tilde{\boldsymbol{w}}_1(t, c_1))|$

$\leq K\|\boldsymbol{w}_1(t, c_1) - \tilde{\boldsymbol{w}}_1(t, c_1)\|_2$

- $|H_2(\boldsymbol{x}, c_2; \boldsymbol{W}(t)) - H_2(\boldsymbol{x}, c_2; \tilde{\boldsymbol{W}}(t))|$

$= |\mathbb{E}_{C_1} w_2(t, C_1, c_2)\sigma_1(\boldsymbol{w}_1(t, C_1)^T \boldsymbol{x}) - \mathbb{E}_{C_1} \tilde{w}_2(t, C_1, c_2)\sigma_1(\tilde{\boldsymbol{w}}_1(t, C_1)^T \boldsymbol{x})|$

$\leq K \underset{C_1}{\mathrm{ess\,sup}} (|w_2(t, C_1, c_2)| \|\boldsymbol{w}_1(t, C_1) - \tilde{\boldsymbol{w}}_1(t, C_1)\|_2 + |w_2(t, C_1, c_2) - \tilde{w}_2(t, C_1, c_2)|)$

- $|f(\boldsymbol{x}; \boldsymbol{W}(t)) - f(\boldsymbol{x}; \tilde{\boldsymbol{W}}(t))|$

$\leq |\mathbb{E}_{C_2} w_3(t, C_2)\sigma_2(H_2(\boldsymbol{x}, C_2; \boldsymbol{W}(t))) - \mathbb{E}_{C_2} \tilde{w}_3(t, C_2)\sigma_2(H_2(\boldsymbol{x}, C_2; \tilde{\boldsymbol{W}}(t)))|$

$\leq K \underset{C_1, C_2}{\mathrm{ess\,sup}}(|w_3(t, C_2)| \cdot |w_2(t, C_1, C_2)| \cdot \|\boldsymbol{w}_1(t, C_1) - \tilde{\boldsymbol{w}}_1(t, C_1)\|_2$

$+ |w_3(t, C_2)| \cdot |w_2(t, C_1, C_2) - \tilde{w}_2(t, C_1, C_2)| + |w_3(t, C_2) - \tilde{w}_3(t, C_2)|)$

- $|\Delta_3^W(\boldsymbol{z}, c_2; \boldsymbol{W}(t)) - \Delta_3^W(\boldsymbol{z}, c_2; \tilde{\boldsymbol{W}}(t))|$

$= |\partial_2 R(y, f(\boldsymbol{x}, \boldsymbol{W}(t)))$

$\cdot \sigma_2(H_2(\boldsymbol{x}, c_2; \boldsymbol{W}(t))) - \partial_2 R(y, f(\boldsymbol{x}, \tilde{\boldsymbol{W}}(t)))\sigma_2(H_2(\boldsymbol{x}, c_2; \tilde{\boldsymbol{W}}(t)))|$

$\leq K \left(|H_2(\boldsymbol{x}, c_2; \boldsymbol{W}(t)) - H_2(\boldsymbol{x}, c_2; \tilde{\boldsymbol{W}}(t))| + |f(\boldsymbol{x}; \boldsymbol{W}(t)) - f(\boldsymbol{x}; \tilde{\boldsymbol{W}}(t))|\right)$

- $|\Delta_2^H(\boldsymbol{z}, c_2; \boldsymbol{W}(t)) - \Delta_2^H(\boldsymbol{z}, c_2; \tilde{\boldsymbol{W}}(t))|$

  $= |\partial_2 R(y, f(\boldsymbol{x}, \boldsymbol{W}(t))) w_3(t, c_2) \sigma_2'(H_2(\boldsymbol{x}, c_2; \boldsymbol{W}(t)))$

  $\qquad - \partial_2 R(y, f(\boldsymbol{x}, \tilde{\boldsymbol{W}}(t))) \tilde{w}_3(t, c_2) \sigma_2'(H_2(\boldsymbol{x}, c_2; \tilde{\boldsymbol{W}}(t)))|$

  $\leq K|w_3(t, c_2)| \cdot |H_2(\boldsymbol{x}, c_2; \boldsymbol{W}(t)) - H_2(\boldsymbol{x}, c_2; \tilde{\boldsymbol{W}}(t))| + K|w_3(t, c_2) - \tilde{w}_3(t, c_2)|$

  $\qquad + K|w_3(t, c_2)| \cdot |f(\boldsymbol{x}; \boldsymbol{W}(t)) - f(\boldsymbol{x}; \tilde{\boldsymbol{W}}(t))|$

- $|\Delta_2^W(\boldsymbol{z}, c_1, c_2; \boldsymbol{W}(t)) - \Delta_2^W(\boldsymbol{z}, c_1, c_2; \tilde{\boldsymbol{W}}(t))|$

  $= |\Delta_2^H(\boldsymbol{z}, c_2; \boldsymbol{W}(t)) \sigma_1(\boldsymbol{w}_1(t, c_1)^T \boldsymbol{x}) - \Delta_2^H(\boldsymbol{z}, c_2; \tilde{\boldsymbol{W}}(t)) \sigma_1(\tilde{\boldsymbol{w}}_1(t, c_1)^T \boldsymbol{x})|$

  $\leq K|\Delta_2^H(\boldsymbol{x}, c_2; \boldsymbol{W}(t)) - \Delta_2^H(\boldsymbol{z}, c_2; \tilde{\boldsymbol{W}}(t))|$

  $\qquad + K|\Delta_2^H(\boldsymbol{z}, c_2; \boldsymbol{W}(t))| \cdot \|\boldsymbol{w}_1(t, c_1) - \tilde{\boldsymbol{w}}_1(t, c_1)\|_2$

- $\|\Delta_1^W(\boldsymbol{z}, c_1; \boldsymbol{W}(t)) - \Delta_1^W(\boldsymbol{z}, c_1; \tilde{\boldsymbol{W}}(t))\|_2$

  $\leq K|\mathbb{E}_{C_2} \Delta_2^H(\boldsymbol{z}, C_2; \boldsymbol{W}(t)) w_2(t, c_1, C_2) \sigma_1'(\boldsymbol{w}_1(t, c_1)^T \boldsymbol{x})$

  $\qquad - \mathbb{E}_{C_2} \Delta_2^H(\boldsymbol{z}, C_2; \tilde{\boldsymbol{W}}(t)) w_2(t, c_1, C_2) \sigma_1'(\tilde{\boldsymbol{w}}_1(t, c_1)^T \boldsymbol{x})|$

  $\leq K|\mathbb{E}_{C_2} \left[ \Delta_2^H(\boldsymbol{z}, C_2; \boldsymbol{W}(t)) w_2(t, c_1, C_2) \right]| \cdot \|\boldsymbol{w}_1(t, c_1) - \tilde{\boldsymbol{w}}_1(t, c_1)\|_2$

  $+ K|\mathbb{E}_{C_2} \left[ \Delta_2^H(\boldsymbol{z}, C_2; \tilde{\boldsymbol{W}}(t)) \tilde{w}_2(t, c_1, C_2) - \Delta_2^H(\boldsymbol{z}, C_2; \boldsymbol{W}(t)) w_2(t, c_1, C_2) \right]|.$

$\qquad\qquad\qquad\qquad\qquad\qquad\qquad\qquad\qquad\qquad\qquad\qquad\qquad\qquad\qquad\qquad \square$

## B  Existence and uniqueness of the mean-field limit

### B.1  Two-layer networks

In this section, we prove the existence and uniqueness of the mean-field limit for two-layer networks. We recall the mean-field ODE again here:

$$
\begin{aligned}
d\boldsymbol{\theta}(t) &= \boldsymbol{r}(t)dt, \\
d\boldsymbol{r}(t) &= \left( -\gamma \boldsymbol{r}(t) - \nabla_{\boldsymbol{\theta}} \Psi(\boldsymbol{\theta}(t); \rho^{\boldsymbol{\theta}}(t)) \right) dt.
\end{aligned} \tag{34}
$$

The proof follows from constructing a Picard type of iteration, similarly to (Sirignano & Spiliopoulos, 2020a, Section 4), (Javanmard et al., 2020, Theorem C.4). Below is an adaptation of the strategy in (Sznitman, 1991, Theorem 1.1). We first write the integral form of the mean-field ODE:

$$
\boldsymbol{\theta}(t) = \boldsymbol{\theta}(0) - \gamma \int_0^t (\boldsymbol{\theta}(s) - \boldsymbol{\theta}(0))\, ds - \int_0^t \int_0^s \nabla_{\boldsymbol{\theta}} \Psi(\boldsymbol{\theta}(u); \rho^{\boldsymbol{\theta}}(u))\, du\, ds, \tag{35}
$$

$$
\boldsymbol{r}(t) = \boldsymbol{r}(0) - \gamma(\boldsymbol{\theta}(t) - \boldsymbol{\theta}(0)) - \int_0^t \Psi(\boldsymbol{\theta}(s); \rho^{\boldsymbol{\theta}}(s))\, ds, \tag{36}
$$

where $\rho(t)$ is the law of $(\boldsymbol{\theta}(t), \boldsymbol{r}(t))$, and we use $\rho^{\boldsymbol{\theta}}(t), \rho^{\boldsymbol{r}}(t)$ to denote the $\boldsymbol{\theta}$ and $\boldsymbol{r}$ marginals, respectively. We define the space $\mathcal{P}_2(\mathbb{R}^D \times \mathbb{R}^D)$ to be the space of probability measures on $\mathbb{R}^D \times \mathbb{R}^D$ equipped with Wasserstein metric $W_2$, and we have $\rho(t) \in \mathcal{P}_2(\mathbb{R}^D \times \mathbb{R}^D)$. We define the space $\mathcal{C}\left([0,T], \mathcal{P}_2(\mathbb{R}^D \times \mathbb{R}^D)\right)$ to be the space of continuous maps $\rho(\cdot; T) : [0,T] \to \mathcal{P}_2(\mathbb{R}^D \times \mathbb{R}^D)$. We omit $T$ when there's no confusion. The space is equipped with the following metric: $d_T(\rho_1, \rho_2) = \sup_{t \in [0,T]} \mathcal{W}_2(\rho_1(t), \rho_2(t))$.

Note that the space $\left(\mathcal{P}_2(\mathbb{R}^D \times \mathbb{R}^D), \mathcal{W}_2\right)$ is a complete space (Ambrosio et al., 2021, Theorem 8.7). Thus for any fixed $0 < T < \infty$, the space $\left(\mathcal{C}\left([0,T], \mathcal{P}_2(\mathbb{R}^D \times \mathbb{R}^D)\right) \times d_T\right)$ is also complete.

Next, we define the operator $H_T(\cdot, \boldsymbol{\theta}(0)) : \mathcal{C}\left([0,T], \mathcal{P}_2(\mathbb{R}^D \times \mathbb{R}^D)\right) \to \mathcal{C}\left([0,T], \mathcal{P}_2(\mathbb{R}^D \times \mathbb{R}^D)\right)$ as follows:

$$
H_T(\rho_1; \boldsymbol{\theta}(0)) := \tilde{\rho}, \quad \tilde{\rho}(t) := \{\text{Law}(\tilde{\boldsymbol{\theta}}(t), \tilde{\boldsymbol{r}}(t))\}_{t \leq T}
$$

$$
\tilde{\boldsymbol{\theta}}(t) = \boldsymbol{\theta}(0) - \gamma \int_0^t (\tilde{\boldsymbol{\theta}}(s) - \boldsymbol{\theta}(0))\, ds - \int_0^t \int_0^s \nabla_{\boldsymbol{\theta}} \Psi(\tilde{\boldsymbol{\theta}}(u); \rho_1^{\boldsymbol{\theta}}(u))\, du\, ds, \tag{37}
$$

where $\boldsymbol{\theta}(0)$ denotes the parameters of the mean-field ODE (35) at initialization, which means that the stochastic process we defined in (37) is coupled with the mean-field ODE.

Note that the $\rho_1^{\boldsymbol{\theta}}(t)$ in (37) is no longer the law of $\widetilde{\boldsymbol{\theta}}(t)$, but the input distribution. We use $H_T(\rho(t))$ to denote $H_T(\rho; \boldsymbol{\theta}(0))(t)$, that is the distribution of the solution (37) at time $t$. We omit $\boldsymbol{\theta}(0)$ when there is no confusion. The definition of $H_T$ can be interpreted as follows: it maps $\rho \in \mathcal{C}\left([0,T], \mathcal{P}_2(\mathbb{R}^D \times \mathbb{R}^D)\right)$ as input to output the law of $(\boldsymbol{\theta}(t), \boldsymbol{r}(t))$ which evolves according to the stochastic process induced by the probability measure $\rho(t)$.

It is easy to see that the fixed point of $H_T$ is the solution of the non-linear dynamics (35). Thus, our goal is to show that there exist a $T_0$ such that $H_{T_0}$ has unique fixed point, or equivalently that $H_{T_0}$ is a strict contraction.

**Proposition B.1.** *Under Assumptions (A1)-(A2), there exists a $T_0$ only depending on $K, \gamma$ and a $C(T_0) \in (0,1)$ such that, for all $\rho_1, \rho_2 \in \mathcal{C}\left([0,T_0], \mathcal{P}_2(\mathbb{R}^D \times \mathbb{R}^D)\right)$ with the same initialization $(\boldsymbol{\theta}_1(0), \boldsymbol{r}_1(0)) = (\boldsymbol{\theta}_2(0), \boldsymbol{r}_2(0))$, we have:*

$$d_{T_0}\left(H_{T_0}\left(\rho_1\right), H_{T_0}\left(\rho_2\right)\right) \leq C(T_0) d_{T_0}(\rho_1, \rho_2).$$

*Proof.* We first fix any $0 < T < \infty$, and the space $\mathcal{C}\left([0,T], \mathcal{P}_2(\mathbb{R}^D \times \mathbb{R}^D)\right)$. Given $\rho_1, \rho_2 \in \mathcal{C}\left([0,T], \mathcal{P}_2(\mathbb{R}^D \times \mathbb{R}^D)\right)$, we define two dynamics as follows:

$$\boldsymbol{\theta}_1(t) = \boldsymbol{\theta}_1(0) - \gamma \int_0^t \left(\boldsymbol{\theta}_1(s) - \boldsymbol{\theta}_1(0)\right) ds - \int_0^t \int_0^s \nabla_{\boldsymbol{\theta}} \Psi(\boldsymbol{\theta}_1(u); \rho_1^{\boldsymbol{\theta}}(u)) \, du \, ds,$$

$$\boldsymbol{\theta}_2(t) = \boldsymbol{\theta}_2(0) - \gamma \int_0^t \left(\boldsymbol{\theta}_2(s) - \boldsymbol{\theta}_2(0)\right) ds - \int_0^t \int_0^s \nabla_{\boldsymbol{\theta}} \Psi(\boldsymbol{\theta}_2(u); \rho_2^{\boldsymbol{\theta}}(u)) \, du \, ds.$$

where $\boldsymbol{\theta}_1(t) = \left(\boldsymbol{w}_1^{(1)}, w_2^{(1)}\right)$ and $\boldsymbol{\theta}_2(t) = \left(\boldsymbol{w}_1^{(2)}, w_2^{(2)}\right)$. We want to upper bound the difference between these two dynamics, which will give us an upper bound on

$$d_T\left(H_T\left(\rho_1\right), H_T\left(\rho_2\right)\right).$$

For all $t \in [0,T]$, we have

$$\begin{aligned}
\|\boldsymbol{\theta}_1(t) - \boldsymbol{\theta}_2(t)\|_2 \leq & \gamma \int_0^t \|\boldsymbol{\theta}_1(s) - \boldsymbol{\theta}_2(s)\|_2 \, ds \\
& + \int_0^t \int_0^s \|\nabla_{\boldsymbol{\theta}} \Psi(\boldsymbol{\theta}_1(u); \rho_1^{\boldsymbol{\theta}}(u)) - \nabla_{\boldsymbol{\theta}} \Psi(\boldsymbol{\theta}_2(u); \rho_2^{\boldsymbol{\theta}}(u))\|_2 \, du \, ds.
\end{aligned}$$

Now, by Lemma A.1, we have that

$$\begin{aligned}
\|\nabla_{\boldsymbol{\theta}} & \Psi(\boldsymbol{\theta}_1(t); \rho_1^{\boldsymbol{\theta}}(t)) - \nabla_{\boldsymbol{\theta}} \Psi(\boldsymbol{\theta}_2(t); \rho_2^{\boldsymbol{\theta}}(t))\|_2 \\
& \leq K(1 + |w_2^{(1)}(t)|) \left(|w_2^{(1)}(t) - w_2^{(2)}(t)| + \|\boldsymbol{w}_1^{(1)}(t) - \boldsymbol{w}_1^{(2)}(t)\|_2 + \mathcal{W}_2(\rho_1^{\boldsymbol{\theta}}(t), \rho_2^{\boldsymbol{\theta}}(t))\right) \\
& \leq 2K(1 + K_2(\gamma, T)) \left(\|\boldsymbol{\theta}_1(t) - \boldsymbol{\theta}_2(t)\|_2 + \max_{s \in [0,T]} \mathcal{W}_2(\rho_1^{\boldsymbol{\theta}}(s), \rho_2^{\boldsymbol{\theta}}(s))\right),
\end{aligned}$$

where $\boldsymbol{\theta}_i(t) = (\boldsymbol{w}_1^{(i)}(t), w_2^{(i)}(t))$, for $i \in 1, 2$. Thus, we have that:

$$\begin{aligned}
\|\boldsymbol{\theta}_1(t) - \boldsymbol{\theta}_2(t)\|_2 \leq & 2K(1 + K_2(\gamma, T)) T^2 \max_{t \in [0,T]} \mathcal{W}_2(\rho_1^{\boldsymbol{\theta}}(t), \rho_2^{\boldsymbol{\theta}}(t)) + \gamma \int_0^t \|\boldsymbol{\theta}_1(s) - \boldsymbol{\theta}_2(s)\|_2 \, ds \\
& + 2K(1 + K_2(\gamma, T)) \int_0^t \int_0^s \|\boldsymbol{\theta}_1(u) - \boldsymbol{\theta}_2(u)\|_2 \, du \, ds.
\end{aligned}$$

Similarly for $\|\boldsymbol{r}_1(t) - \boldsymbol{r}_2(t)\|_2$, we have that:

$$\|\boldsymbol{r}_1(t) - \boldsymbol{r}_2(t)\|_2 \leq \gamma \int_0^t \|\boldsymbol{r}_1(s) - \boldsymbol{r}_2(s)\|_2 \, ds + \int_0^t \|\Psi(\boldsymbol{\theta}_1(s); \rho_1^{\boldsymbol{\theta}}(s)) - \Psi(\boldsymbol{\theta}_2(s); \rho_2^{\boldsymbol{\theta}}(s))\|_2 \, ds$$

$$\leq 2K(1 + K_2(\gamma, T))T \max_{t \in [0,T]} \mathcal{W}_2(\rho_1^{\boldsymbol{\theta}}(t), \rho_2^{\boldsymbol{\theta}}(t)) + \gamma \int_0^t \|\boldsymbol{r}_1(s) - \boldsymbol{r}_2(s)\|_2 \, ds$$

$$+ 2K(1 + K_2(\gamma, T)) \int_0^t \|\boldsymbol{\theta}_1(s) - \boldsymbol{\theta}_2(s)\|_2 \, ds.$$

Putting these two results together we have:

$$\left\| \begin{pmatrix} \boldsymbol{\theta}_1(t) \\ \boldsymbol{r}_1(t) \end{pmatrix} - \begin{pmatrix} \boldsymbol{\theta}_2(t) \\ \boldsymbol{r}_2(t) \end{pmatrix} \right\|_2 \leq \|\boldsymbol{\theta}_1(t) - \boldsymbol{\theta}_2(t)\|_2 + \|\boldsymbol{r}_1(t) - \boldsymbol{r}_2(t)\|_2$$

$$\leq 4K(1 + K_2(\gamma, T))T^2 \max_{t \in [0,T]} \mathcal{W}_2(\rho_1^{\boldsymbol{\theta}}(t), \rho_2^{\boldsymbol{\theta}}(t))$$

$$+ \gamma \int_0^t (\|\boldsymbol{\theta}_1(s) - \boldsymbol{\theta}_2(s)\|_2 + \|\boldsymbol{r}_1(s) - \boldsymbol{r}_2(s)\|_2) \, ds$$

$$+ 4K(1 + K_2(\gamma, T)) \int_0^t \int_0^s \|\boldsymbol{\theta}_1(u) - \boldsymbol{\theta}_2(u)\|_2 \, du \, ds$$

$$\leq 4K(1 + K_2(\gamma, T))T^2 \max_{t \in [0,T]} \mathcal{W}_2(\rho_1^{\boldsymbol{\theta}}(t), \rho_2^{\boldsymbol{\theta}}(t)) + 2\gamma \int_0^t \left\| \begin{pmatrix} \boldsymbol{\theta}_1(s) \\ \boldsymbol{r}_1(s) \end{pmatrix} - \begin{pmatrix} \boldsymbol{\theta}_2(s) \\ \boldsymbol{r}_2(s) \end{pmatrix} \right\|_2 ds$$

$$+ 4K(1 + K_2(\gamma, T)) \int_0^t \int_0^s \left\| \begin{pmatrix} \boldsymbol{\theta}_1(u) \\ \boldsymbol{r}_1(u) \end{pmatrix} - \begin{pmatrix} \boldsymbol{\theta}_2(u) \\ \boldsymbol{r}_2(u) \end{pmatrix} \right\|_2 du \, ds.$$

By Corollary F.4, we have that:

$$\left\| \begin{pmatrix} \boldsymbol{\theta}_1(t) \\ \boldsymbol{r}_1(t) \end{pmatrix} - \begin{pmatrix} \boldsymbol{\theta}_2(t) \\ \boldsymbol{r}_2(t) \end{pmatrix} \right\|_2$$

$$\leq 4K(1 + K_2(\gamma, T))T^2 \left( 1 + \exp\left( \frac{4\gamma^2 + 4K(1 + K_2(\gamma, T))T}{2\gamma} \right) \right) \max_{t \in [0,T]} \mathcal{W}_2(\rho_1^{\boldsymbol{\theta}}(t), \rho_2^{\boldsymbol{\theta}}(t)).$$

Thus, we have that

$$\left\| \begin{pmatrix} \boldsymbol{\theta}_1(t) \\ \boldsymbol{r}_1(t) \end{pmatrix} - \begin{pmatrix} \boldsymbol{\theta}_2(t) \\ \boldsymbol{r}_2(t) \end{pmatrix} \right\|_2 \leq T^2 K(\gamma, T) \max_{t \in [0,T]} \mathcal{W}_2(\rho_1^{\boldsymbol{\theta}}(t), \rho_2^{\boldsymbol{\theta}}(t)),$$

which implies that

$$\max_{t \in [0,T]} \mathcal{W}_2(H_T(\rho_1(t)), H_T(\rho_2(t))) \leq T^2 K(\gamma, T) \max_{t \in [0,T]} \mathcal{W}_2(\rho_1^{\boldsymbol{\theta}}(t), \rho_2^{\boldsymbol{\theta}}(t)),$$

where we set $K(\gamma, T) = 4K(1 + K_2(\gamma, T)) \left( 1 + \exp \frac{4\gamma^2 + 4K(1 + K_2(\gamma, T))T}{4\gamma} \right)$.

Let $C(T) = T^2 K(\gamma, T)$. Then, we could always find a $T_0$ such that $C(T_0) < 1$ since $C(0) = 0$ and $C(T)$ is continuous in $T$, which finishes our proof. □

By Banach's fixed point theorem, there exist a $T_0 > 0$ such that the mean-field ODE has a unique solution in time interval $[0, T_0]$. Now, we show the existence and uniqueness of the solution of the mean-field ODE for any time period $[0, T]$.

**Theorem B.2.** *Under Assumptions (A1)-(A2), for any $T > 0$, there exists a unique solution for the mean-field ODE (12) in the interval $[0, T]$.*

*Proof.* The idea is to separate the time interval $[0, T]$ into subintervals of length $T_0$, that is, we consider the intervals $[0, T_0], [T_0, 2T_0], ..., [\lfloor \frac{T}{T_0} \rfloor T_0, T]$ . Note that the contraction property we proved in Proposition B.1 only depends on the length of the time interval, so the proof can be done recursively. That is:

1. In the interval $[0, T_0]$, (12) with initialization $(\theta(0), r(0))$ has a unique solution $\{\rho(t)\}_{t \in [0, T_0]}$.

2. In the interval $[T_0, 2T_0]$, we consider (12) with initial distribution $\rho(T_0)$, and it has a unique solution $\{\rho(t)\}_{t \in [T_0, 2T_0]}$.

3. Recursively do the above steps until the interval $[\lfloor \frac{T}{T_0} \rfloor T_0, T]$.

Thus we have that, for any $T > 0$, there exists a unique solution for (12) in the interval $[0, T]$. $\qquad \square$

## B.2 Three-layer networks

In this section, we prove the existence and the uniqueness of the mean-field ODE (13). The integral form of the mean-field ODE is given by

$$w_3(t, c_2) = w_3(0, c_2) - \gamma \int_0^t (w_3(s, c_2) - w_3(0, c_2))\, ds - \int_0^t \int_0^s \mathbb{E}_{\boldsymbol{z}} \Delta_3^W(u, \boldsymbol{x}, c_2)\, du\, ds, \tag{38}$$

$$w_2(t, c_1, c_2) = w_2(0, c_1, c_2) - \gamma \int_0^t (w_2(s, c_1, c_2) - w_2(0, c_1, c_2))\, ds - \int_0^t \int_0^s \mathbb{E}_{\boldsymbol{z}} \Delta_2^W(u, \boldsymbol{z}, c_1, c_2)\, du\, ds, \tag{39}$$

$$\boldsymbol{w}_1(t, c_1) = \boldsymbol{w}_1(0, c_1) - \gamma \int_0^t (\boldsymbol{w}_1(s, c_1) - \boldsymbol{w}_1(0, c_1))\, ds - \int_0^t \int_0^s \mathbb{E}_{\boldsymbol{z}} \Delta_1^W(u, \boldsymbol{z}, c_1)\, du\, ds. \tag{40}$$

In order to prove the existence and the uniqueness, we follow the same Picard's iteration arguments as for the two-layers case. We first define the following norm:

$$\|\boldsymbol{W}\|_T = \max \operatorname*{ess\,sup}_{C_1, C_2} \sup_{t \in [0, T]} \{|w_2(t, C_1, C_2)|, |w_3(t, C_2)|\}. \tag{41}$$

Next, we define the following metric for two sets of mean-field parameters:

$$\begin{aligned} \mathcal{D}_T(\boldsymbol{W}, \boldsymbol{W}') = \max \operatorname*{ess\,sup}_{C_1, C_2} \sup_{t \in [0, T]} \big\{ &\|\boldsymbol{w}_1'(t, C_1) - \boldsymbol{w}_1(t, C_1)\|_2, \\ &|w_2'(t, C_1, C_2) - w_2(t, C_1, C_2)|, |w_3'(t, C_2) - w_3(t, C_2)| \big\}. \end{aligned} \tag{42}$$

Note that the metric we define above is not the metric induced by the norm, since in the definition of the norm we only require the boundedness of $w_2$ and $w_3$.

We define the following functional space of the mean-field parameters:

$$\mathcal{W}_T(\boldsymbol{W}(0)) = \{\{\widetilde{\boldsymbol{W}}(t)\}_{t \in [0, T]} : \|\widetilde{\boldsymbol{W}}\|_T < \infty, \widetilde{\boldsymbol{W}}(0) = \boldsymbol{W}(0)\}, \tag{43}$$

which means that all the $\widetilde{\boldsymbol{W}} \in \mathcal{W}_T(\boldsymbol{W}(0))$ have the same initialization $\boldsymbol{W}(0)$. By Lemma A.2, we know that:

$$\operatorname*{ess\,sup}_{C_1, C_2} |w_2(t, C_1, C_2)| \le K_{3,2}(\gamma, T), \qquad \operatorname*{ess\,sup}_{C_1, C_2} |w_3(t, C_2)| \le K_{3,3}(\gamma, T). \tag{44}$$

It is easy to see that the space $\mathcal{W}_T(\boldsymbol{W}(0))$ is complete w.r.t. the metric $\mathcal{D}_T(\boldsymbol{W}, \boldsymbol{W}')$. Let us define the operator $H_T : \mathcal{W}_T(\boldsymbol{W}(0)) \to \mathcal{W}_T(\boldsymbol{W}(0))$ as follows:

*Input:* $\{\boldsymbol{W}(t)\}_{t \in [0, T]}$

*Output:* $\{\widetilde{\boldsymbol{W}}(t)\}_{t \in [0,T]}$, such that:

$$\widetilde{w}_3(t, c_2) = w_3(0, c_2) - \gamma \int_0^t (w_3(s, c_2) - w_3(0, c_2)) \, ds - \int_0^t \int_0^s \mathbb{E}_{\boldsymbol{z}} \Delta_3^W(\boldsymbol{x}, c_2; \boldsymbol{W}(u)) \, du \, ds \qquad (45)$$

$$\widetilde{w}_2(t, c_1, c_2) = w_2(0, c_1, c_2) - \gamma \int_0^t (w_2(s, c_1, c_2) - w_2(0, c_1, c_2)) \, ds - \int_0^t \int_0^s \mathbb{E}_{\boldsymbol{z}} \Delta_2^W(\boldsymbol{z}, c_1, c_2; \boldsymbol{W}(u)) \, du \, ds \qquad (46)$$

$$\widetilde{\boldsymbol{w}}_1(t, c_1) = \boldsymbol{w}_1(0, c_1) - \gamma \int_0^t (\boldsymbol{w}_1(s, c_1) - \boldsymbol{w}_1(0, c_1)) \, ds - \int_0^t \int_0^s \mathbb{E}_{\boldsymbol{z}} \Delta_1^W(\boldsymbol{z}, c_1; \boldsymbol{W}(u)) \, du \, ds. \qquad (47)$$

We aim to show the following proposition.

**Proposition B.3.** *Under Assumptions (B1)-(B3), there exists a $T_0$ only depending on $K, \gamma$ and $C(T_0) \in (0, 1)$, such that, for all $\boldsymbol{W}^1, \boldsymbol{W}^2 \in \mathcal{W}_T(\boldsymbol{W}(0))$, we have:*

$$\mathcal{D}_{T_0}(H_{T_0}(\boldsymbol{W}^1), H_{T_0}(\boldsymbol{W}^2)) \leq C(T_0) \mathcal{D}_{T_0}(\boldsymbol{W}^1, \boldsymbol{W}^2). \qquad (48)$$

*Proof of Proposition B.3.* For simplicity of notation, we denote the output of $H_T(\boldsymbol{W}^1)$ to be $\widetilde{\boldsymbol{W}}^1$, which is composed of $\widetilde{w}_3^1, \widetilde{w}_2^1, \widetilde{w}_1^1$. The output of $H_T(\boldsymbol{W}^2)$ is denoted similarly.

By the definition of the mean-field ODE, we have that, for any $t \leq T$,

$$|\widetilde{w}_3^1(t, c_2) - \widetilde{w}_3^2(t, c_2)| \leq \gamma \int_0^t |w_3^1(s, c_2) - w_3^2(s, c_2)| \, ds$$
$$+ \int_0^t \int_0^s \mathbb{E}_{\boldsymbol{z}} |\Delta_3^W(\boldsymbol{x}, c_2; \boldsymbol{W}^1(u)) - \Delta_3^W(\boldsymbol{x}, c_2; \boldsymbol{W}^2(u))| \, du \, ds,$$

$$|\widetilde{w}_2^1(t, c_1, c_2) - \widetilde{w}_2^2(t, c_1, c_2)| \leq \gamma \int_0^t |w_2^1(s, c_1, c_2) - w_2^2(s, c_1, c_2)| \, ds$$
$$+ \int_0^t \int_0^s \mathbb{E}_{\boldsymbol{z}} |\Delta_2^W(\boldsymbol{x}, c_1, c_2; \boldsymbol{W}^1(u)) - \Delta_2^W(\boldsymbol{x}, c_1, c_2; \boldsymbol{W}^2(u))| \, du \, ds,$$

$$\|\widetilde{w}_1^1(t, c_1) - \widetilde{w}_1^2(t, c_1)\|_2 \leq \gamma \int_0^t \|w_1^1(s, c_1) - w_1^2(s, c_1)\|_2 \, ds$$
$$+ \int_0^t \int_0^s \mathbb{E}_{\boldsymbol{z}} \|\Delta_1^W(\boldsymbol{x}, c_1; \boldsymbol{W}^1(u)) - \Delta_1^W(\boldsymbol{x}, c_1; \boldsymbol{W}^2(u))\|_2 \, du \, ds.$$

By Lemma A.2, we have that:

$$\max\{\mathbb{E}_{\boldsymbol{z}} |\Delta_3^W(\boldsymbol{x}, c_2; \boldsymbol{W}^1(u)) - \Delta_3^W(\boldsymbol{x}, c_2; \boldsymbol{W}^2(u))|,$$
$$\mathbb{E}_{\boldsymbol{z}} |\Delta_2^W(\boldsymbol{x}, c_1, c_2; \boldsymbol{W}^1(u)) - \Delta_2^W(\boldsymbol{x}, c_1, c_2; \boldsymbol{W}^2(u))|,$$
$$\mathbb{E}_{\boldsymbol{z}} \|\Delta_1^W(\boldsymbol{x}, c_1; \boldsymbol{W}^1(u)) - \Delta_1^W(\boldsymbol{x}, c_1; \boldsymbol{W}^2(u))\|_2\} \leq K(\gamma, T) \mathcal{D}_u(\boldsymbol{W}^1, \boldsymbol{W}^2).$$

Thus, we have:

$$\mathcal{D}_t(\widetilde{\boldsymbol{W}}^1, \widetilde{\boldsymbol{W}}^2) \leq \gamma \int_0^t \mathcal{D}_s(\boldsymbol{W}^1, \boldsymbol{W}^2) \, ds + K(\gamma, T) \int_0^t \int_{u=0}^s \mathcal{D}_u(\boldsymbol{W}^1, \boldsymbol{W}^2) \, du \, ds$$
$$\leq (\gamma t + t^2) K(\gamma, T) \mathcal{D}_t(\boldsymbol{W}^1, \boldsymbol{W}^2),$$

which implies that

$$\mathcal{D}_T(\widetilde{\boldsymbol{W}}^1, \widetilde{\boldsymbol{W}}^2) \leq (\gamma T + T^2) K(\gamma, T) \mathcal{D}_T(\boldsymbol{W}^1, \boldsymbol{W}^2). \qquad (49)$$

Since $(\gamma T + T^2) K(\gamma, T) = 0$ when $T = 0$, and $(\gamma T + T^2) K(\gamma, T)$ is continuous in $T$, we can pick a $T_0$ such that $(\gamma T_0 + T_0^2) K(\gamma, T_0) < 1$, which finishes the proof. $\qquad \square$

Since $\mathcal{W}_T(\boldsymbol{W}(0))$ is complete, by Banach's fixed point theorem, there exists a unique fixed point for the operator $H_{T_0}$, which implies that the mean-field ODE (13) has a unique solution in $[0, T_0]$. By following the same argument of the proof of Theorem B.2 (separate the interval $[0, T]$ into sub-intervals of length $T_0$ and successively apply Proposition B.3 to each of them), we readily obtain our main result concerning the existence and uniqueness of (13) in $[0, T]$.

**Theorem B.4.** *Under Assumptions (B1)-(B3), for any $T > 0$, there exists a unique solution for the mean-field ODE (13) in the interval $[0, T]$.*

## C  Convergence to the mean-field limit − Two-layer networks

In this section, we prove the convergence to the mean-field limit for two-layer neural networks (Theorem 5.1). Our proof's structure is inspired from Mei et al. (2019). Before going into the arguments, we first recall the definition of the mean-field ODE and the stochastic heavy ball method (SHB) for two-layer networks. Then, we define two auxiliary dynamics: the particle dynamics (PD) and the heavy ball dynamics (HB).

First, recall the mean-field ODE as follows:

$$
\begin{aligned}
d\boldsymbol{\theta}(t) &= \boldsymbol{r}(t)dt, \\
d\boldsymbol{r}(t) &= \left(-\gamma \boldsymbol{r}(t) - \nabla_{\boldsymbol{\theta}}\Psi(\boldsymbol{\theta}(t); \rho^{\boldsymbol{\theta}}(t))\right) dt,
\end{aligned}
\tag{50}
$$

and the corresponding integral form

$$
\boldsymbol{\theta}(t) = \boldsymbol{\theta}(0) - \gamma \int_0^t (\boldsymbol{\theta}(s) - \boldsymbol{\theta}(0))\, ds - \int_0^t \int_0^s \nabla_{\boldsymbol{\theta}}\Psi(\boldsymbol{\theta}(u); \rho^{\boldsymbol{\theta}}(t))\, du\, ds.
\tag{51}
$$

The SHB dynamics is as follows:

$$
\boldsymbol{\theta}^{SHB}(k+1, j) = \boldsymbol{\theta}^{SHB}(k, j) + (1 - \gamma\varepsilon)(\boldsymbol{\theta}^{SHB}(k, j) - \boldsymbol{\theta}^{SHB}(k-1, j)) - \varepsilon^2 \nabla_{\boldsymbol{\theta}}\widehat{\Psi}(\boldsymbol{z}, \boldsymbol{\theta}^{SHB}(k, j); \rho^{\boldsymbol{\theta}}_{SHB}(k)),
$$
$$
\forall j \in [n],
\tag{52}
$$

where $\rho^{\boldsymbol{\theta}}_{SHB}(k) = \frac{1}{n}\sum_{j=1}^n \delta_{\boldsymbol{\theta}(k,j)}$ is the empirical distribution.

In order to describe the convergence to mean-field limit, we define the following particle dynamics (PD):

$$
\begin{aligned}
d\boldsymbol{\theta}^{PD}(t, j) &= \boldsymbol{r}^{PD}(t, j)dt \\
d\boldsymbol{r}^{PD}(t, j) &= \left(-\gamma \boldsymbol{r}^{PD}(t, j) - \nabla_{\boldsymbol{\theta}}\Psi(\boldsymbol{\theta}^{PD}(t, j); \rho^{\boldsymbol{\theta}}_{PD}(t))\right) dt, \ \ \forall j \in [n],
\end{aligned}
\tag{53}
$$

where $\rho^{\boldsymbol{\theta}}_{PD}(t) = \frac{1}{n}\sum_{j=1}^n \delta_{\boldsymbol{\theta}^{PD}(t,j)}$ is the empirical distribution at time $t$. Furthermore, the heavy ball (HB) dynamics is defined as

$$
\boldsymbol{\theta}^{HB}(k+1, j) = \boldsymbol{\theta}^{HB}(k, j) + (1 - \gamma\varepsilon)(\boldsymbol{\theta}^{HB}(k, j) - \boldsymbol{\theta}^{HB}(k-1, j)) - \varepsilon^2 \nabla_{\boldsymbol{\theta}}\Psi(\boldsymbol{\theta}^{HB}(k, j); \rho^{\boldsymbol{\theta}}_{HB}(k)), \ \ \forall j \in [n].
\tag{54}
$$

We remark that (52), (53) and (54) have the same initialization, that is :

$$
\boldsymbol{\theta}^{PD}(0, j) = \boldsymbol{\theta}^{HB}(0, j) = \boldsymbol{\theta}^{SHB}(0, j), \ \ \forall j \in [n].
$$

Define the following distance metrics:

$$
\mathcal{D}_T(\boldsymbol{\theta}, \boldsymbol{\theta}^{PD}) := \max_{j \in [n]} \sup_{t \in [0, T]} \left\|\boldsymbol{\theta}^{PD}(t, j) - \boldsymbol{\theta}(t, j)\right\|_2,
\tag{55}
$$

$$
\mathcal{D}_T(\boldsymbol{\theta}^{PD}, \boldsymbol{\theta}^{HB}) := \max_{j \in [n]} \sup_{t \in [0, T]} \left\|\boldsymbol{\theta}^{HB}(\lfloor t/\varepsilon \rfloor, j) - \boldsymbol{\theta}^{PD}(t, j)\right\|_2,
\tag{56}
$$

$$
\mathcal{D}_{T,\varepsilon}(\boldsymbol{\theta}^{HB}, \boldsymbol{\theta}^{SHB}) := \max_{j \in [n]} \max_{k \in \lfloor T/\varepsilon \rfloor} \|\boldsymbol{\theta}^{HB}(k, j) - \boldsymbol{\theta}^{SHB}(k, j)\|_2.
\tag{57}
$$

## C.1 Bound between mean-field ODE and particle dynamics

In this section, we bound the difference between the mean-field ODE defined in (51) and the particle dynamics defined in (53), whose integral form is as follows:

$$\boldsymbol{\theta}^{PD}(t,j) = \boldsymbol{\theta}^{PD}(0,j) - \gamma \int_0^t (\boldsymbol{\theta}^{PD}(s,j) - \boldsymbol{\theta}^{PD}(0,j))\, ds - \int_0^t \int_0^s \nabla_{\boldsymbol{\theta}} \Psi(\boldsymbol{\theta}^{PD}(u,j); \rho_{PD}^{\boldsymbol{\theta}}(u))\, du\, ds.$$

**Proposition C.1.** *Under Assumptions (A1) - (A3), we have that, with probability at least $1 - \exp(-\delta^2)$,*

$$\mathcal{D}_T(\boldsymbol{\theta}, \boldsymbol{\theta}^{PD}) \le K(\gamma, T) \left( \frac{\delta + \sqrt{\log n}}{\sqrt{n}} \right), \tag{58}$$

*where $K(\gamma, T)$ is a constant depending only on $\gamma, T$.*

Before proving Proposition C.1, we first prove the following lemma, which characterizes the Lipschitz continuity of the mean-field ODE.

**Lemma C.2.** *Under Assumptions (A1) - (A3), there exists a universal constant $K(\gamma, T)$ depending only on $\gamma, T$ such that, for any $t, \tau > 0$ such that $t, t + \tau < T$,*

$$\begin{aligned}
\|\boldsymbol{\theta}(t + \tau) - \boldsymbol{\theta}(t)\|_2 &\le K(\gamma, T)\tau, \\
\mathcal{W}_2(\rho^{\boldsymbol{\theta}}(t + \tau), \rho^{\boldsymbol{\theta}}(t)) &\le K(\gamma, T)\tau.
\end{aligned} \tag{59}$$

*The same holds for the particle dynamics $\boldsymbol{\theta}^{PD}(t,j), \forall j \in [n]$.*

*Proof of Lemma C.2.* We only prove the results for the mean-field ODE, and the proof for the particle dynamics follows from the same arguments.

We first try to bound the increments $\|\boldsymbol{\theta}(t) - \boldsymbol{\theta}(0)\|_2$. By the definition of the mean-field dynamics, we have that:

$$\begin{aligned}
\|\boldsymbol{\theta}(t) - \boldsymbol{\theta}(0)\|_2 &\le \gamma \int_0^t \|\boldsymbol{\theta}(s) - \boldsymbol{\theta}(0)\|_2\, ds + \int_0^t \int_0^s \|\nabla_{\boldsymbol{\theta}} \Psi(\boldsymbol{\theta}(u); \rho^{\boldsymbol{\theta}}(u))\|_2\, du\, ds \\
&\le \gamma \int_0^t \|\boldsymbol{\theta}(s) - \boldsymbol{\theta}(0)\|_2\, ds + K_1(\gamma, T),
\end{aligned}$$

where in the last step we use that $\|\nabla_{\boldsymbol{\theta}} \Psi(\boldsymbol{\theta}(u); \rho^{\boldsymbol{\theta}}(u))\|_2 \le K_1(\gamma, T)$, which follows from Lemma A.1.

By Gronwall's lemma, this implies that, for any $t \le T$,

$$\|\boldsymbol{\theta}(t) - \boldsymbol{\theta}(0)\|_2 \le K_1(\gamma, T) \exp(\gamma T) := K_2(\gamma, T).$$

Next, by definition of the mean-field ODE, we have that:

$$\|\boldsymbol{\theta}(t + \tau) - \boldsymbol{\theta}(t)\|_2 \le \gamma \int_t^{t+\tau} \|\boldsymbol{\theta}(s) - \boldsymbol{\theta}(0)\|_2\, ds + \int_t^{t+\tau} \int_0^s \|\nabla_{\boldsymbol{\theta}} \Psi(\boldsymbol{\theta}(u); \rho^{\boldsymbol{\theta}}(u))\|_2\, du\, ds.$$

Thus,

$$\|\boldsymbol{\theta}(t + \tau) - \boldsymbol{\theta}(t)\|_2 \le K_4(\gamma, T)\tau,$$

where we use that fact that

$$\|\boldsymbol{\theta}(s) - \boldsymbol{\theta}(0)\|_2 \le K_3(\gamma, T),$$

$$\int_0^s \|\nabla_{\boldsymbol{\theta}}, \Psi(\boldsymbol{\theta}(u); \rho^{\boldsymbol{\theta}}(u))\|_2\, du \le K_3(\gamma, T).$$

For the Lipschitz continuity of $\rho^{\boldsymbol{\theta}}$, we just note that by definition of the $\mathcal{W}_2$ distance, we have:

$$\mathcal{W}_2(\rho^{\boldsymbol{\theta}}(t+\tau), \rho^{\boldsymbol{\theta}}(t)) \leq \mathbb{E}\left[\|\boldsymbol{\theta}(t+\tau) - \boldsymbol{\theta}(t)\|_2^2\right]^{1/2}.$$

$\square$

Now we are ready to prove Proposition C.1.

*Proof of Proposition C.1.* In order to bound the difference, we first define $n$ i.i.d mean-field dynamics:

$$\boldsymbol{\theta}(t,j) = \boldsymbol{\theta}(0,j) - \gamma \int_0^t (\boldsymbol{\theta}(s,j) - \boldsymbol{\theta}(0,j))\, ds - \int_0^t \int_0^s \nabla_{\boldsymbol{\theta}} \Psi(\boldsymbol{\theta}(u,j); \rho^{\boldsymbol{\theta}}(u,j))\, du\, ds,$$

where $\rho^{\boldsymbol{\theta}}(t,j)$ is the law of $\boldsymbol{\theta}(t,j)$, and we coupled the $n$ i.i.d dynamics with the particle dynamics at initialization, that is, we let:

$$\boldsymbol{\theta}(0,j) = \boldsymbol{\theta}^{PD}(0,j), \;\; \forall j \in [n].$$

We also define the empirical distribution of $\boldsymbol{\theta}(t,j)$, that is: $\widehat{\rho}^{\boldsymbol{\theta}}(t) = \frac{1}{n}\sum_{j=1}^n \delta_{\boldsymbol{\theta}(t,j)}$. Since the $n$ mean-field dynamics are i.i.d, we have that $\rho^{\boldsymbol{\theta}}(t,j) = \rho^{\boldsymbol{\theta}}(t)$, $\forall j \in [n]$, thus we use the notation of $\rho^{\boldsymbol{\theta}}(t)$ to denote the the law of $\boldsymbol{\theta}(t,j)$ for each $j \in [n]$.

By Lemma A.1 and Lemma C.2, we know that:

$$\sup_{t \in [T]} \max_{j \in [n]} \|w_2(t,j)\|_2 \leq K(\gamma, T),$$

$$\sup_{t \in [T]} \max_{j \in [n]} \|\boldsymbol{\theta}(t+\tau, j) - \boldsymbol{\theta}(t,j)\|_2 \leq K(\gamma, T)\tau.$$

We have that

$$\|\boldsymbol{\theta}^{PD}(t,j) - \boldsymbol{\theta}(t,j)\|_2 \leq (1+\gamma t)\|\boldsymbol{\theta}^{PD}(0,j) - \boldsymbol{\theta}(0,j)\|_2 + \gamma \int_0^t \|\boldsymbol{\theta}^{PD}(s,j) - \boldsymbol{\theta}(s,j)\|_2\, ds$$

$$+ \int_0^t \int_0^s \|\nabla_{\boldsymbol{\theta}} \Psi(\boldsymbol{\theta}^{PD}(u,j); \rho_{PD}^{\boldsymbol{\theta}}(u)) - \nabla_{\boldsymbol{\theta}} \Psi(\boldsymbol{\theta}(u,j); \rho^{\boldsymbol{\theta}}(u))\|_2\, du\, ds,$$

and our goal is to bound:

$$\sup_{t \in [0,T]} \max_{j \in [n]} \|\boldsymbol{\theta}^{PD}(t,j) - \boldsymbol{\theta}(t,j)\|_2.$$

Now we aim to bound the quantity

$$\sup_{t \in [0,T]} \max_{j \in [n]} \|\nabla_{\boldsymbol{\theta}} \Psi(\boldsymbol{\theta}^{PD}(u,j); \rho_{PD}^{\boldsymbol{\theta}}(u,j)) - \nabla_{\boldsymbol{\theta}} \Psi(\boldsymbol{\theta}(u,j); \rho^{\boldsymbol{\theta}}(u))\|_2.$$

An application of the triangle inequality gives

$$\|\nabla_{\boldsymbol{\theta}} \Psi(\boldsymbol{\theta}(u,j); \rho^{\boldsymbol{\theta}}(u)) - \nabla_{\boldsymbol{\theta}} \Psi(\boldsymbol{\theta}^{PD}(u,j); \rho_{PD}^{\boldsymbol{\theta}}(u,j))\|_2 \leq \|\nabla_{\boldsymbol{\theta}} \Psi(\boldsymbol{\theta}(u,j); \widehat{\rho}^{\boldsymbol{\theta}}(u)) - \nabla_{\boldsymbol{\theta}} \Psi(\boldsymbol{\theta}(u,j); \rho^{\boldsymbol{\theta}}(u))\|_2 \quad (60)$$

$$+ \|\nabla_{\boldsymbol{\theta}} \Psi(\boldsymbol{\theta}(u,j); \widehat{\rho}^{\boldsymbol{\theta}}(u)) - \nabla_{\boldsymbol{\theta}} \Psi(\boldsymbol{\theta}^{PD}(u,j); \rho_{PD}^{\boldsymbol{\theta}}(u))\|_2. \quad (61)$$

Recall by definition that

$$\nabla_{\boldsymbol{w}_1} \Psi(\boldsymbol{\theta}(u,j); \rho^{\boldsymbol{\theta}}(u)) = \mathbb{E}_{\boldsymbol{z}}\left[\partial_2 R(y, f(\boldsymbol{x}; \rho^{\boldsymbol{\theta}}(u)))w_2(u,j)\sigma'(\boldsymbol{w}_1(u,j)^T \boldsymbol{x})\boldsymbol{x}\right],$$

$$\nabla_{w_2} \Psi(\boldsymbol{\theta}(u,j); \rho^{\boldsymbol{\theta}}(u)) = \mathbb{E}_{\boldsymbol{z}}\left[\partial_2 R(y, f(\boldsymbol{x}; \rho^{\boldsymbol{\theta}}(u)))\sigma(\boldsymbol{w}_1(u,j)^T \boldsymbol{x})\right].$$

Similarly,

$$\nabla_{\boldsymbol{w}_1}\Psi(\boldsymbol{\theta}(u,j);\widehat{\rho}^{\boldsymbol{\theta}}(u)) = \mathbb{E}_{\boldsymbol{z}}\left[\partial_2 R(y,f(\boldsymbol{x};\widehat{\rho}^{\boldsymbol{\theta}}(u)))w_2(u,j)\sigma'(\boldsymbol{w}_1(u,j)^T\boldsymbol{x})\boldsymbol{x}\right],$$
$$\nabla_{w_2}\Psi(\boldsymbol{\theta}(u,j);\widehat{\rho}^{\boldsymbol{\theta}}(u)) = \mathbb{E}_{\boldsymbol{z}}\left[\partial_2 R(y,f(\boldsymbol{x};\widehat{\rho}^{\boldsymbol{\theta}}(u)))\sigma(\boldsymbol{w}_1(u,j)^T\boldsymbol{x})\right].$$

For the term in (60), we use concentration inequalities to give an upper bound. By the Lipschitz continuity of $\partial_2 R$ in Assumption 3.1, we have

$$\|\nabla_{\boldsymbol{\theta}}\Psi(\boldsymbol{\theta}(u,j);\widehat{\rho}^{\boldsymbol{\theta}}(u)) - \nabla_{\boldsymbol{\theta}}\Psi(\boldsymbol{\theta}(u,j);\rho^{\boldsymbol{\theta}}(u))\|_2$$
$$\leq K\left\|\mathbb{E}_{\boldsymbol{z}}\begin{pmatrix}w_2(u,j)\sigma'(\boldsymbol{w}_1(u,j)^T\boldsymbol{x})\boldsymbol{x}\\\sigma(\boldsymbol{w}_1(u,j)^T\boldsymbol{x})\end{pmatrix}\right\|_2|f(\boldsymbol{x};\rho^{\boldsymbol{\theta}}(u)) - f(\boldsymbol{x};\widehat{\rho}^{\boldsymbol{\theta}}(u))|$$
$$\leq K_1(\gamma,T)|f(\boldsymbol{x};\rho^{\boldsymbol{\theta}}(u)) - f(\boldsymbol{x};\widehat{\rho}^{\boldsymbol{\theta}}(u))|.$$

For the term $|f(\boldsymbol{x};\rho(u)) - f(\boldsymbol{x};\widehat{\rho}(u))|$, we have

$$|f(\boldsymbol{x};\rho^{\boldsymbol{\theta}}(u)) - f(\boldsymbol{x};\widehat{\rho}^{\boldsymbol{\theta}}(u))| = \left|\frac{1}{n}\sum_{j=1}^{n}w_2(u,j)\sigma(\boldsymbol{w}_1(u,j)^T\boldsymbol{x}) - \mathbb{E}_{\rho(u)}w_2(u)\sigma(\boldsymbol{w}_1(u)^T\boldsymbol{x})\right|.$$

Note that, by Lemma A.1, we know that

$$\left|w_2(u,j)\sigma(\boldsymbol{w}_1(u,j)^T\boldsymbol{x}) - \mathbb{E}_{\rho(u)}w_2(u)\sigma(\boldsymbol{w}_1(u)^T\boldsymbol{x})\right| \leq K_2(\gamma,T).$$

By Lemma F.1, we have that, with probability at least $1 - \exp(-n(\delta')^2)$,

$$\left|\frac{1}{n}\sum_{i=1}^{n}w_2(u,j)\sigma(\boldsymbol{w}_1(u,j)^T\boldsymbol{x}) - \mathbb{E}_{\rho(u)}w_2(u)\sigma(\boldsymbol{w}_1(u)^T\boldsymbol{x})\right| \leq K_2(\gamma,T)\left(\frac{1}{\sqrt{n}} + \delta'\right).$$

By Lemma C.2, we know that $\left|\frac{1}{n}\sum_{i=1}^{n}w_2(u,j)\sigma(\boldsymbol{w}_1(u,j)^T\boldsymbol{x}) - \mathbb{E}_{\rho(u)}w_2(u)\sigma(\boldsymbol{w}_1(u)^T\boldsymbol{x})\right|$ is $K(\gamma,T)$-Lipschitz continuous in $u$. Thus, by taking a union bound over $j \in [n]$ and $t \in \left\{0,\eta,...,\lfloor\frac{T}{\eta}\rfloor\eta\right\}$, we have that, with probability at least $1 - \frac{nT}{\eta}\exp(-n(\delta')^2)$,

$$\max_{j\in[n]}\sup_{t\in[0,T]}\left|\frac{1}{n}\sum_{i=1}^{n}w_2(u,j)\sigma(\boldsymbol{w}_1(u,j)^T\boldsymbol{x}) - \mathbb{E}_{\rho(u)}w_2(u)\sigma(\boldsymbol{w}_1(u)^T\boldsymbol{x})\right| \leq K_3(\gamma,T)\left(\frac{1}{\sqrt{n}} + \delta' + \eta\right).$$

Take $\eta = \frac{1}{\sqrt{n}}$, $\delta' = \sqrt{\frac{\delta^2+\log\left(n^{\frac{3}{2}}T\right)}{n}}$. Then, with probability at least $1 - \exp(-\delta^2)$,

$$\max_{j\in[n]}\sup_{t\in[0,T]}\left|\frac{1}{n}\sum_{i=1}^{n}\mathbb{E}_{\rho(u)}w_2(u)\sigma(\boldsymbol{w}_1(u)^T\boldsymbol{x}) - w_2(u,j)\sigma(\boldsymbol{w}_1(u,j)^T\boldsymbol{x})\right| \leq K_4(\gamma,T)\frac{\delta + \sqrt{\log n}}{\sqrt{n}},$$

which implies that, for term (60),

$$\max_{j\in[n]}\sup_{t\in[0,T]}\|\nabla_{\boldsymbol{\theta}}\Psi(\boldsymbol{\theta}(u,j);\widehat{\rho}^{\boldsymbol{\theta}}(u)) - \nabla_{\boldsymbol{\theta}}\Psi(\boldsymbol{\theta}(u,j);\rho^{\boldsymbol{\theta}}(u))\|_2 \leq K_5(\gamma,T)\frac{\delta + \sqrt{\log n}}{\sqrt{n}},$$

with probability $1 - \exp(-\delta^2)$.

For the term in (61), we use the Lipschitz continuity of $\nabla_{\boldsymbol{\theta}}\Psi$. By Lemma A.1, we have that, for each $j \in [n]$,

$$\|\nabla_{\boldsymbol{\theta}}\Psi(\boldsymbol{\theta}(t,j);\widehat{\rho}^{\boldsymbol{\theta}}(t)) - \nabla_{\boldsymbol{\theta}}\Psi(\boldsymbol{\theta}^{PD}(t,j);\rho_{PD}^{\boldsymbol{\theta}}(t))\|_2 \leq K_6(\gamma,T)(\mathcal{D}_t(\boldsymbol{\theta},\boldsymbol{\theta}^{PD}) + \mathcal{W}_2(\widehat{\rho}^{\boldsymbol{\theta}}(t),\rho_{PD}^{\boldsymbol{\theta}}(t))).$$

Note that $\widehat{\rho}^{\boldsymbol{\theta}}(t), \rho_{PD}^{\boldsymbol{\theta}}(t)$ are discrete measures, thus we have:

$$\mathcal{W}_2(\widehat{\rho}^{\boldsymbol{\theta}}(t), \rho_{PD}^{\boldsymbol{\theta}}(t)) \leq \left( \frac{1}{n} \sum_{j=1}^{n} \|\boldsymbol{\theta}(t,j) - \boldsymbol{\theta}^{PD}(t,j)\|_2^2 \right)^{1/2} \leq \mathcal{D}_t(\boldsymbol{\theta}, \boldsymbol{\theta}^{PD}).$$

Hence,

$$\|\nabla_{\boldsymbol{\theta}} \Psi(\boldsymbol{\theta}(t,j); \widehat{\rho}^{\boldsymbol{\theta}}(t)) - \nabla_{\boldsymbol{\theta}} \Psi(\boldsymbol{\theta}^{PD}(t,j); \rho_{PD}^{\boldsymbol{\theta}}(t))\|_2 \leq K_7(\gamma, T) \mathcal{D}_t(\boldsymbol{\theta}, \boldsymbol{\theta}^{PD}).$$

Combining the above results, we have that, with probability $1 - \exp(-\delta^2)$,

$$\mathcal{D}_t(\boldsymbol{\theta}, \boldsymbol{\theta}^{PD}) \leq K_8(\gamma, T) \frac{\delta + \sqrt{\log n}}{\sqrt{n}} + \gamma \int_0^t \mathcal{D}_s(\boldsymbol{\theta}, \boldsymbol{\theta}^{PD}) \, ds + K_8(\gamma, T) \int_0^t \int_0^s \mathcal{D}_u(\boldsymbol{\theta}, \boldsymbol{\theta}^{PD}) \, du \, ds.$$

An application of Corollary F.4 concludes the proof. $\qquad\square$

## C.2 Bound between particle dynamics and heavy ball dynamics

In this section, we bound the difference between the particle dynamics defined in (53) and the heavy ball dynamics defined in (54). We recall that the distance we aim to bound is defined in (56). Note that the heavy ball dynamics is a discretization of the particle dynamics. Thus we aim to bound the difference at time point $k\varepsilon$.

**Proposition C.3.** *Under Assumptions (A1) - (A3), there exist a universal constant $K(\gamma, T)$ depending only on $\gamma, T$, such that*

$$\mathcal{D}_T(\boldsymbol{\theta}^{PD}, \boldsymbol{\theta}^{HB}) \leq K(\gamma, T)\varepsilon. \tag{62}$$

*Proof of Proposition C.3.* By the Taylor expansion, we have the following approximation for the particle dynamics:

$$\boldsymbol{\theta}^{PD}((k+1)\varepsilon, j) = \boldsymbol{\theta}^{PD}(k\varepsilon, j) + \boldsymbol{r}^{PD}(k\varepsilon, j)\varepsilon + \frac{1}{2} \partial_t \boldsymbol{r}^{PD}(k\varepsilon, j)\varepsilon^2 + O(\varepsilon^3). \tag{63}$$

Also by Taylor expansion we have:

$$\boldsymbol{r}^{PD}(k\varepsilon, j)\varepsilon = \boldsymbol{\theta}^{PD}(k\varepsilon, j) - \boldsymbol{\theta}^{PD}((k-1)\varepsilon, j) + \frac{1}{2} \partial_t \boldsymbol{r}^{PD}(k\varepsilon, j)\varepsilon^2 + O(\varepsilon^3). \tag{64}$$

By plugging (61) into (63), we have that

$$\begin{aligned}
\boldsymbol{\theta}^{PD}((k+1)\varepsilon, j) &= \boldsymbol{\theta}^{PD}(k\varepsilon, j) + \boldsymbol{\theta}^{PD}(k\varepsilon, j) - \boldsymbol{\theta}^{PD}((k-1)\varepsilon, j) + \partial_t \boldsymbol{r}^{PD}(k\varepsilon, j)\varepsilon^2 + O(\varepsilon^3) \\
&= \boldsymbol{\theta}^{PD}(k\varepsilon, j) + \boldsymbol{\theta}^{PD}(k\varepsilon, j) - \boldsymbol{\theta}^{PD}((k-1)\varepsilon, j) + \left( -\gamma \boldsymbol{r}(k\varepsilon, j) - \nabla_{\boldsymbol{\theta}} \Psi(\boldsymbol{\theta}^{PD}(k\varepsilon, j); \rho_{PD}^{\boldsymbol{\theta}}(k\varepsilon)) \right) \varepsilon^2 + O(\varepsilon^3) \\
&= \boldsymbol{\theta}^{PD}(k\varepsilon, j) + (1 - \gamma\varepsilon)(\boldsymbol{\theta}^{PD}(k\varepsilon, j) - \boldsymbol{\theta}^{PD}((k-1)\varepsilon, j)) - \nabla_{\boldsymbol{\theta}} \Psi(\boldsymbol{\theta}^{PD}(k\varepsilon, j); \rho_{PD}^{\boldsymbol{\theta}}(k\varepsilon))\varepsilon^2 + O(\varepsilon^3).
\end{aligned}$$

Now we get a discrete iteration equation for the particle dynamics, with an approximation error of at most $O(\varepsilon^3)$. By accumulating the $\nabla_{\boldsymbol{\theta}} \Psi(\boldsymbol{\theta}^{PD}(l\varepsilon, j); \rho_{PD}^{\boldsymbol{\theta}}(l\varepsilon))$ term from $l = 1, ..., k$, we have

$$\boldsymbol{\theta}^{PD}(k\varepsilon, j) = \boldsymbol{\theta}^{PD}(0, j) - \sum_{l=0}^{k-1} c_l^{(k)} (\nabla_{\boldsymbol{\theta}} \Psi(\boldsymbol{\theta}^{PD}(l\varepsilon, j); \rho_{PD}^{\boldsymbol{\theta}}(l\varepsilon)) + O(\varepsilon)), \tag{65}$$

where $c_l^{(k)} = \varepsilon^2 \sum_{i=0}^{k-1-l} (1 - \gamma\varepsilon)^i = \varepsilon^2 \frac{1-(1-\gamma\varepsilon)^k}{\gamma\varepsilon} \leq \frac{\varepsilon}{\gamma}$.

The heavy ball dynamics can be written in a similar fashion:

$$\boldsymbol{\theta}^{HB}(k\varepsilon, j) = \boldsymbol{\theta}^{HB}(0, j) - \sum_{l=0}^{k-1} c_l^{(k)} \nabla_{\boldsymbol{\theta}} \Psi(\boldsymbol{\theta}^{HB}(l\varepsilon, j); \rho_{HB}^{\boldsymbol{\theta}}(l\varepsilon)). \tag{66}$$

Thus, we have that

$$\|\boldsymbol{\theta}^{PD}(k\varepsilon, j) - \boldsymbol{\theta}^{HB}(k\varepsilon, j)\|_2 \le \sum_{l=0}^{k-1} c_l^{(k)} \left( \|\nabla_{\boldsymbol{\theta}} \Psi(\boldsymbol{\theta}^{PD}(l\varepsilon, j); \rho_{PD}^{\boldsymbol{\theta}}(l\varepsilon)) - \nabla_{\boldsymbol{\theta}} \Psi(\boldsymbol{\theta}^{HB}(l\varepsilon, j); \rho_{HB}^{\boldsymbol{\theta}}(l\varepsilon))\|_2 + O(\varepsilon) \right). \tag{67}$$

By Lemma A.1, we have that

$$\|\nabla_{\boldsymbol{\theta}} \Psi(\boldsymbol{\theta}^{PD}(l\varepsilon, j); \rho_{PD}^{\boldsymbol{\theta}}(l\varepsilon)) - \nabla_{\boldsymbol{\theta}} \Psi(\boldsymbol{\theta}^{HB}(l\varepsilon, j); \rho_{HB}^{\boldsymbol{\theta}}(l\varepsilon))\|_2$$
$$\le K_1(\gamma, T)(\|\boldsymbol{\theta}^{PD}(l\varepsilon, j) - \boldsymbol{\theta}^{HB}(l\varepsilon, j)\|_2 + \mathcal{W}_2(\rho_{PD}^{\boldsymbol{\theta}}(l\varepsilon), \rho_{HB}^{\boldsymbol{\theta}}(l\varepsilon))).$$

Since $\rho_{PD}^{\boldsymbol{\theta}}(l\varepsilon), \rho_{HB}^{\boldsymbol{\theta}}(l\varepsilon))$ are discrete distributions, we have that

$$\mathcal{W}_2(\rho_{PD}^{\boldsymbol{\theta}}(l\varepsilon), \rho_{HB}^{\boldsymbol{\theta}}(l\varepsilon))) \le \left( \frac{1}{n} \sum_{j=1}^{n} \|\boldsymbol{\theta}^{PD}(l\varepsilon, j) - \boldsymbol{\theta}^{HB}(l\varepsilon, j)\|_2^2 \right)^{1/2} \le \mathcal{D}_{l\varepsilon}(\boldsymbol{\theta}^{PD}, \boldsymbol{\theta}^{HB}),$$

which implies that

$$\|\nabla_{\boldsymbol{\theta}} \Psi(\boldsymbol{\theta}^{PD}(l\varepsilon, j); \rho_{PD}^{\boldsymbol{\theta}}(l\varepsilon)) - \nabla_{\boldsymbol{\theta}} \Psi(\boldsymbol{\theta}^{HB}(l\varepsilon, j); \rho_{HB}^{\boldsymbol{\theta}}(l\varepsilon))\|_2 \le K_2(\gamma, T)\mathcal{D}_{l\varepsilon}(\boldsymbol{\theta}^{PD}, \boldsymbol{\theta}^{HB}).$$

As a result, we have

$$\mathcal{D}_{k\varepsilon}(\boldsymbol{\theta}^{PD}, \boldsymbol{\theta}^{HB}) \le K_2(\gamma, T)\frac{\varepsilon}{\gamma} \sum_{l=1}^{k-1} \left( \mathcal{D}_{l\varepsilon}(\boldsymbol{\theta}^{PD}, \boldsymbol{\theta}^{HB}) + O(\varepsilon) \right).$$

Finally, an application of the discrete Gronwall's lemma concludes the proof. $\qquad \square$

### C.3 Bound between heavy ball dynamics and stochastic heavy ball dynamics

In this section, we bound the difference between the heavy ball dynamics defined in (54) and the stochastic heavy ball dynamics defined in (52). We recall that the distance we aim to bound is defined in (57). The manipulations of the previous section imply that the heavy ball dynamics can be written as

$$\boldsymbol{\theta}^{HB}(k, j) = \boldsymbol{\theta}^{HB}(0, j) - \sum_{l=1}^{k-1} c_l^{(k)} \nabla_{\boldsymbol{\theta}} \Psi(\boldsymbol{\theta}^{HB}(l, j); \rho_{HB}^{\boldsymbol{\theta}}(l)). \tag{68}$$

Similarly, the stochastic heavy ball dynamics can be written as

$$\boldsymbol{\theta}^{SHB}(k, j) = \boldsymbol{\theta}^{SHB}(0, j) - \sum_{l=0}^{k-1} c_l^{(k)} \nabla_{\boldsymbol{\theta}} \widehat{\Psi}(\boldsymbol{z}(l), \boldsymbol{\theta}^{SHB}(l, j); \rho_{SHB}^{\boldsymbol{\theta}}(l)). \tag{69}$$

**Proposition C.4.** *Under Assumptions (A1) - (A3), there exists a universal constant $K(\gamma, T)$ depending only on $\gamma, T$, such that, with probability $1 - \exp(-\delta^2)$,*

$$\mathcal{D}_{T, \varepsilon}(\boldsymbol{\theta}^{HB}, \boldsymbol{\theta}^{SHB}) \le K(\gamma, T)\sqrt{\varepsilon}(\sqrt{D + \log n} + \delta). \tag{70}$$

*Proof.* By using (68) and (69), we have

$$\left\| \boldsymbol{\theta}^{HB}(k,j) - \boldsymbol{\theta}^{SHB}(k,j) \right\|_2 \leq \left\| \sum_{l=0}^{k-1} c_l^{(k)} (\nabla_{\boldsymbol{\theta}} \Psi(\boldsymbol{\theta}^{HB}(l,j); \rho_{HB}^{\boldsymbol{\theta}}(l)) - \nabla_{\boldsymbol{\theta}} \widehat{\Psi}(\boldsymbol{z}(l), \boldsymbol{\theta}^{SHB}(l,j); \rho_{SHB}^{\boldsymbol{\theta}}(l))) \right\|_2.$$

By triangle inequality, we have that

$$\left\| \sum_{l=0}^{k-1} c_l^{(k)} (\nabla_{\boldsymbol{\theta}} \Psi(\boldsymbol{\theta}^{HB}(l,j); \rho_{HB}^{\boldsymbol{\theta}}(l)) - \nabla_{\boldsymbol{\theta}} \widehat{\Psi}(\boldsymbol{z}(l), \boldsymbol{\theta}^{SHB}(l,j); \rho_{SHB}^{\boldsymbol{\theta}}(l))) \right\|_2$$

$$\leq \sum_{l=0}^{k-1} c_l^{(k)} \| \nabla_{\boldsymbol{\theta}} \widehat{\Psi}(\boldsymbol{z}(l), \boldsymbol{\theta}^{SHB}(l,j); \rho_{SHB}^{\boldsymbol{\theta}}(l)) - \nabla_{\boldsymbol{\theta}} \widehat{\Psi}(\boldsymbol{z}(l), \boldsymbol{\theta}^{HB}(l,j); \rho_{HB}^{\boldsymbol{\theta}}(l)) \|_2 \tag{71}$$

$$+ \left\| \sum_{l=0}^{k-1} c_l^{(k)} (\nabla_{\boldsymbol{\theta}} \widehat{\Psi}(\boldsymbol{z}(l), \boldsymbol{\theta}^{HB}(l,j); \rho_{HB}^{\boldsymbol{\theta}}(l)) - \nabla_{\boldsymbol{\theta}} \Psi(\boldsymbol{\theta}^{HB}(l,j); \rho_{HB}^{\boldsymbol{\theta}}(l))) \right\|_2. \tag{72}$$

For the term in (71), by the Lipschitz continuity of $\nabla_{\boldsymbol{\theta}} \widehat{\Psi}$, we obtain

$$\| \nabla_{\boldsymbol{\theta}} \widehat{\Psi}(\boldsymbol{z}(l), \boldsymbol{\theta}^{SHB}(l,j); \rho_{SHB}^{\boldsymbol{\theta}}(l)) - \nabla_{\boldsymbol{\theta}} \widehat{\Psi}(\boldsymbol{z}(l), \boldsymbol{\theta}^{HB}(l,j); \rho_{HB}^{\boldsymbol{\theta}}(l)) \|_2$$
$$\leq K_1(\gamma, T)(\mathcal{D}_{l\varepsilon,\varepsilon}(\boldsymbol{\theta}^{HB}, \boldsymbol{\theta}^{SHB}) + \mathcal{W}_2(\rho_{HB}^{\boldsymbol{\theta}}(l), \rho_{SHB}^{\boldsymbol{\theta}}(l))).$$

Since $\rho_{HB}^{\boldsymbol{\theta}}, \rho_{SHB}^{\boldsymbol{\theta}}$ are discrete distributions, we have that $\mathcal{W}_2(\rho_{HB}^{\boldsymbol{\theta}}(l), \rho_{SHB}^{\boldsymbol{\theta}}(l)) \leq \mathcal{D}_{l\varepsilon,\varepsilon}(\boldsymbol{\theta}^{HB}, \boldsymbol{\theta}^{SHB})$. Thus,

$$\| \nabla_{\boldsymbol{\theta}} \widehat{\Psi}(\boldsymbol{z}(l), \boldsymbol{\theta}^{SHB}(l,j); \rho_{SHB}^{\boldsymbol{\theta}}(l)) - \nabla_{\boldsymbol{\theta}} \widehat{\Psi}(\boldsymbol{z}(l), \boldsymbol{\theta}^{HB}(l,j); \rho_{HB}^{\boldsymbol{\theta}}(l)) \|_2 \leq K_2(\gamma, T) \mathcal{D}_{l\varepsilon,\varepsilon}(\boldsymbol{\theta}^{HB}, \boldsymbol{\theta}^{SHB}).$$

For the term in (72), note that, since the $\boldsymbol{z}(l)$'s are sampled i.i.d. at each step by definition, we have

$$\mathbb{E}_{\boldsymbol{z}(l)} \left[ \nabla_{\boldsymbol{\theta}} \widehat{\Psi}(\boldsymbol{z}(l), \boldsymbol{\theta}^{HB}(l,j); \rho_{HB}^{\boldsymbol{\theta}}(l)) \right] = \nabla_{\boldsymbol{\theta}} \Psi(\boldsymbol{\theta}^{HB}(l,j); \rho_{HB}^{\boldsymbol{\theta}}(l)).$$

Thus,

$$\nabla_{\boldsymbol{\theta}} \widehat{\Psi}(\boldsymbol{z}(l), \boldsymbol{\theta}^{HB}(l,j); \rho_{HB}^{\boldsymbol{\theta}}(l)) - \nabla_{\boldsymbol{\theta}} \Psi(\boldsymbol{\theta}^{HB}(l,j); \rho_{HB}^{\boldsymbol{\theta}}(l))$$

is a martingale difference. By Lemma A.1, we have that, for all $l \in \{1, ..., \lfloor T/\varepsilon \rfloor\}$,

$$\| \nabla_{\boldsymbol{\theta}} \widehat{\Psi}(\boldsymbol{z}(l), \boldsymbol{\theta}^{HB}(l,j); \rho_{HB}^{\boldsymbol{\theta}}(l)) - \nabla_{\boldsymbol{\theta}} \Psi(\boldsymbol{\theta}^{HB}(l,j); \rho_{HB}^{\boldsymbol{\theta}}(l)) \|_2 \leq K_3(\gamma, T).$$

Thus, an application of Lemma F.2 gives that, with probability at least $1 - \exp(-\delta^2)$,

$$\max_{l \in \{1, ..., \lfloor T/\varepsilon \rfloor\}} \| \nabla_{\boldsymbol{\theta}} \widehat{\Psi}(\boldsymbol{z}(l), \boldsymbol{\theta}^{HB}(l,j); \rho_{HB}^{\boldsymbol{\theta}}(l)) - \nabla_{\boldsymbol{\theta}} \Psi(\boldsymbol{\theta}^{HB}(l,j); \rho_{HB}^{\boldsymbol{\theta}}(l)) \|_2 \leq K_4(\gamma, T) \sqrt{\varepsilon}(\sqrt{D} + \delta).$$

By taking a union bounds on $j \in [n]$, we have that, with probability at least $1 - \exp(-\delta^2)$,

$$\max_{j \in [n]} \max_{l \in \{1, ..., \lfloor T/\varepsilon \rfloor\}} \| \nabla_{\boldsymbol{\theta}} \widehat{\Psi}(\boldsymbol{z}(l), \boldsymbol{\theta}^{HB}(l,j); \rho_{HB}^{\boldsymbol{\theta}}(l)) - \nabla_{\boldsymbol{\theta}} \Psi(\boldsymbol{\theta}^{HB}(l,j); \rho_{HB}^{\boldsymbol{\theta}}(l)) \|_2 \leq K_4(\gamma, T) \sqrt{\varepsilon}(\sqrt{D + \log n} + \delta).$$

By combining the above result, we conclude that

$$\mathcal{D}_{T,\varepsilon}(\boldsymbol{\theta}^{HB}, \boldsymbol{\theta}^{SHB}) \leq K_4(\gamma, T) \sqrt{\varepsilon}(\sqrt{D + \log n} + \delta) + \frac{K_2(\gamma, T)}{\gamma} \varepsilon \sum_{l=0}^{\lfloor \frac{T}{\varepsilon} \rfloor} \mathcal{D}_{l\varepsilon,\varepsilon}(\boldsymbol{\theta}^{HB}, \boldsymbol{\theta}^{SHB}). \tag{73}$$

Finally, an application of the discrete Gronwall's lemma concludes the proof. $\qquad\square$

### C.4 Proof of Theorem 5.1

*Proof of Theorem 5.1.* The proof follows from combining Proposition C.1, C.3, C.4, and the fact that:

$$\mathcal{D}_T(\boldsymbol{\theta}, \boldsymbol{\theta}^{SHB}) \leq \mathcal{D}_T(\boldsymbol{\theta}, \boldsymbol{\theta}^{PD}) + \mathcal{D}_T(\boldsymbol{\theta}^{PD}, \boldsymbol{\theta}^{HB}) + \mathcal{D}_{T,\varepsilon}(\boldsymbol{\theta}^{HB}, \boldsymbol{\theta}^{SHB}).$$

□

## D Convergence to the mean-field limit − Three-layer networks

In this section, we prove the convergence of the training dynamics to the mean-field limit for a three-layer neural network. Our proof's structure is inspired from Pham & Nguyen (2021a).

Before going into the proofs, let's first recall the definition of the mean-field ODE and the SHB dynamics, and then define two auxiliary dynamics, namely the HB dynamics and the particle dynamics. For the convenience of further computation, we define these continuous dynamics in integral form. We define the random variable corresponding to the stochastic heavy ball dynamics, the heavy ball dynamics, the particle dynamics, and the mean-field ODE as $\boldsymbol{W}^{SHB}, \boldsymbol{W}^{HB}, \boldsymbol{W}^{PD}, \boldsymbol{W}$ respectively.

The mean-field ODE (13) in integral form is the following:

$$w_3(t, c_2) = w_3(0, c_2) - \gamma \int_0^t (w_3(s, c_2) - w_3(0, c_2))\, ds - \int_0^t \int_0^s \mathbb{E}_{\boldsymbol{z}} \Delta_3^W(\boldsymbol{z}, c_2; \boldsymbol{W}(u))\, du\, ds,$$

$$w_2(t, c_1, c_2) = w_2(0, c_1, c_2) - \gamma \int_0^t (w_2(s, c_1, c_2) - w_2(0, c_1, c_2))\, ds - \int_0^t \int_0^s \mathbb{E}_{\boldsymbol{z}} \Delta_2^W(\boldsymbol{z}, c_1, c_2; \boldsymbol{W}(u))\, du\, ds,$$

$$\boldsymbol{w}_1(t, c_1) = \boldsymbol{w}_1(0, c_1) - \gamma \int_0^t (\boldsymbol{w}_1(s, c_1) - \boldsymbol{w}_1(0, c_1))\, ds - \int_0^t \int_0^s \mathbb{E}_{\boldsymbol{z}} \Delta_1^W(\boldsymbol{z}, c_1; \boldsymbol{W}(u))\, du\, ds.$$

(74)

The SHB dynamics is as follows:

$$w_3^{SHB}(k+1, j_2) = w_3^{SHB}(k, j_2) + (1 - \gamma\varepsilon)(w_3^{SHB}(k, j_2) - w_3^{SHB}(k-1, j_2)) - \varepsilon^2 \Delta_3^W(\boldsymbol{z}(k), j_2; \boldsymbol{W}^{SHB}(k)),$$

$$w_2^{SHB}(k+1, j_1, j_2) = w_2^{SHB}(k, j_1, j_2) + (1 - \gamma\varepsilon)(w_2^{SHB}(k, j_1, j_2) - w_2^{SHB}(k-1, j_1, j_2)),$$
$$- \varepsilon^2 \Delta_2^W(\boldsymbol{z}(k), j_1, j_2; \boldsymbol{W}^{SHB}(k))$$

$$\boldsymbol{w}_1^{SHB}(k+1, j_1) = \boldsymbol{w}_1^{SHB}(k, j_1) + (1 - \gamma\varepsilon)(\boldsymbol{w}_1^{SHB}(k, j_1) - \boldsymbol{w}_1^{SHB}(k-1, j_1)) - \varepsilon^2 \Delta_1^W(\boldsymbol{z}(k), j_1; \boldsymbol{W}^{SHB}(k)),$$

(75)

where $\boldsymbol{z}(k)$ is the data point sampled at time step $k$.

We define the particle dynamics as a continuous dynamics without mean-field interaction:

$$w_3^{PD}(t, j_2) = w_3^{PD}(0, j_2) - \gamma \int_0^t (w_3^{PD}(s, j_2) - w_3^{PD}(0, j_2))\, ds - \int_0^t \int_0^s \mathbb{E}_{\boldsymbol{z}} \Delta_3^W(\boldsymbol{z}, j_2; \boldsymbol{W}^{PD}(u))\, du\, ds,$$

$$w_2^{PD}(t, j_1, j_2) = w_2^{PD}(0, j_1, j_2) - \gamma \int_0^t (w_2^{PD}(s, j_1, j_2) - w_2^{PD}(0, j_1, j_2))\, ds$$

$$- \int_0^t \int_0^s \mathbb{E}_{\boldsymbol{z}} \Delta_2^W(\boldsymbol{z}, j_1, j_2; \boldsymbol{W}^{PD}(u))\, du\, ds,$$

$$\boldsymbol{w}_1^{PD}(t, j_1) = \boldsymbol{w}_1^{PD}(0, j_1) - \gamma \int_0^t (\boldsymbol{w}_1^{PD}(s, j_1) - \boldsymbol{w}_1^{PD}(0, j_1))\, ds - \int_0^t \int_0^s \mathbb{E}_{\boldsymbol{z}} \Delta_1^W(\boldsymbol{z}, j_1; \boldsymbol{W}^{PD}(u))\, du\, ds.$$

(76)

We define the HB dynamics by replacing the stochastic gradient in the SHB dynamics by the true gradient. That is:

$$
\begin{aligned}
w_3^{HB}(k+1, j_2) &= w_3^{HB}(k, j_2) + (1-\gamma\varepsilon)(w_3^{HB}(k, j_2) - w_3^{HB}(k-1, j_2)) - \varepsilon^2 \mathbb{E}_{\boldsymbol{z}} \Delta_3^W(\boldsymbol{z}, j_2; \boldsymbol{W}^{HB}(k)), \\
w_2^{HB}(k+1, j_1, j_2) &= w_2^{HB}(k, j_1, j_2) + (1-\gamma\varepsilon)(w_2^{HB}(k, j_1, j_2) - w_2^{HB}(k-1, j_1, j_2)) \\
&\qquad\qquad\qquad\qquad\qquad\qquad\qquad - \varepsilon^2 \mathbb{E}_{\boldsymbol{z}} \Delta_2^W(\boldsymbol{z}, j_1, j_2; \boldsymbol{W}^{HB}(k)), \\
\boldsymbol{w}_1^{HB}(k+1, j_1) &= \boldsymbol{w}_1^{HB}(k, j_1) + (1-\gamma\varepsilon)(\boldsymbol{w}_1^{HB}(k, j_1) - \boldsymbol{w}_1^{HB}(k-1, j_1)) - \varepsilon^2 \mathbb{E}_{\boldsymbol{z}} \Delta_1^W(\boldsymbol{z}, j_1; \boldsymbol{W}^{HB}(k)).
\end{aligned}
\tag{77}
$$

Note that the HB dynamics can be viewed as the discrete version of the PD dynamics.

In order to present our main theoretical results in this section, we first define a distance metric to quantify the level of correspondence of these dynamics. We define the following distance metrics:

$$
\begin{aligned}
\mathcal{D}_T(\boldsymbol{W}, \boldsymbol{W}^{PD}) = \max \sup_{t \in [0,T]} \{ &\left\| \boldsymbol{w}_1^{PD}(t, j_1) - \boldsymbol{w}_1(t, C_1(j_1)) \right\|_2, \\
&\left| w_2^{PD}(t, j_1, j_2) - w_2(t, C_1(j_1), C_2(j_2)) \right|, \\
&\left| w_3^{PD}(t, j_2) - w_3(t, C_2(j_2)) \right| : j_1 \in [n_1], j_2 \in [n_2] \}
\end{aligned}
\tag{78}
$$

$$
\begin{aligned}
\mathcal{D}_T(\boldsymbol{W}^{PD}, \boldsymbol{W}^{HB}) = \max \sup_{t \in [0,T]} \{ &\left\| \boldsymbol{w}_1^{HB}(\lfloor t/\varepsilon \rfloor, j_1) - \boldsymbol{w}_1^{PD}(t, j_1) \right\|_2, \\
&\left| w_2^{HB}(\lfloor t/\varepsilon \rfloor, j_1, j_2) - w_2^{PD}(t, j_1, j_2) \right|, \\
&\left| w_3^{HB}(\lfloor t/\varepsilon \rfloor, j_2) - w_3^{PD}(t, j_2) \right| : j_1 \in [n_1], j_2 \in [n_2] \}
\end{aligned}
\tag{79}
$$

$$
\begin{aligned}
\mathcal{D}_{T,\varepsilon}(\boldsymbol{W}^{HB}, \boldsymbol{W}^{SHB}) = \max \max_{k \in \lfloor T/\varepsilon \rfloor} \{ &\| \boldsymbol{w}_1^{HB}(k, j_1) - \boldsymbol{w}_1^{SHB}(k, j_1) \|_2, \\
&| w_2^{HB}(k, j_1, j_2) - w_2^{SHB}(k, j_1, j_2) |, \\
&| w_3^{HB}(k, j_2) - w_3^{SHB}(k, j_2) | : j_1 \in [n_1], j_2 \in [n_2] \}
\end{aligned}
\tag{80}
$$

We acknowledge the abuse in notation from reusing $\mathcal{D}_T$ for multiple metrics, which is done to avoid proliferation of notation. It is clear that:

$$
\mathcal{D}_T(\boldsymbol{W}, \boldsymbol{W}^{SHB}) \leq \mathcal{D}_T(\boldsymbol{W}, \boldsymbol{W}^{PD}) + \mathcal{D}_T(\boldsymbol{W}^{PD}, \boldsymbol{W}^{HB}) + \mathcal{D}_{T,\varepsilon}(\boldsymbol{W}^{HB}, \boldsymbol{W}^{SHB}),
\tag{81}
$$

where $\mathcal{D}_T(\boldsymbol{W}, \boldsymbol{W}^{SHB})$ is defined in (17).

In the following subsections, we bound the terms $\mathcal{D}_T(\boldsymbol{W}, \boldsymbol{W}^{PD})$, $\mathcal{D}_T(\boldsymbol{W}^{PD}, \boldsymbol{W}^{HB})$ and $\mathcal{D}_{T,\varepsilon}(\boldsymbol{W}^{HB}, \boldsymbol{W}^{SHB})$.

### D.1 Bound between mean-field ODE and particle dynamics

In this section, we bound the difference between the mean-field ODE defined in (74) and the particle dynamics defined in (76). We recall that the distance we aim to bound is defined in (78).

**Proposition D.1.** *Under Assumptions (B1)-(B3), we have that, with probability at least $1 - \exp(-\delta^2)$,*

$$
\mathcal{D}_T(\boldsymbol{W}, \boldsymbol{W}^{PD}) \leq \frac{K(\gamma, T)}{\sqrt{n_{\min}}} \left( \sqrt{\log n_{\max}} + \delta \right),
\tag{82}
$$

*where $K(\gamma, T)$ is a universal constant depending only on $\gamma, T$, $n_{\min} = \min\{n_1, n_2\}$ and $n_{\max} = \max\{n_1, n_2\}$.*

In order to prove Proposition D.1, we need the following auxiliary lemma, which characterizes the Lipschitz continuity of the mean-field ODE and of the particle dynamics.

**Lemma D.2.** *Under Assumptions (B1)-(B3), there exists a universal constant $K(\gamma, T)$ depending only on $\gamma, T$ such that, for any $t, \tau > 0$ with $t, t + \tau \leq T$,*

$$
\begin{aligned}
\operatorname*{ess\,sup}_{c_2} |w_3(t+\tau, c_2) - w_3(t, c_2)| &\leq K(\gamma, T)\tau, \\
\operatorname*{ess\,sup}_{c_1, c_2} |w_2(t+\tau, c_1, c_2) - w_2(t, c_1, c_2)| &\leq K(\gamma, T)\tau, \\
\operatorname*{ess\,sup}_{c_1} \|\boldsymbol{w}_1(t+\tau, c_1) - \boldsymbol{w}_1(t, c_1)\|_2 &\leq K(\gamma, T)\tau.
\end{aligned}
\tag{83}
$$

*The same holds for the particle dynamics $\boldsymbol{w}_1^{PD}(t, j_1), w_2^{PD}(t, j_1, j_2), w_3^{PD}(t, j_2)$.*

*Proof of Lemma D.2.* We do the proof for $\boldsymbol{w}_1(t, c_1)$, and the same argument applies to $w_2(t, c_1, c_2), w_3(t, c_2)$. First, we derive a bound on the increments up to time $t$, $\|\boldsymbol{w}_1(t, c_1) - \boldsymbol{w}_1(0, c_1)\|_2$. By the definition of the mean-field ODE (74), we have that

$$
\boldsymbol{w}_1(t, c_1) - \boldsymbol{w}_1(0, c_1) = -\gamma \int_0^t (\boldsymbol{w}_1(s, c_1) - \boldsymbol{w}_1(0, c_1))\, ds - \int_0^t \int_0^s \mathbb{E}_{\boldsymbol{z}} \Delta_1^W(\boldsymbol{z}, c_1; \boldsymbol{W}(u))\, du\, ds,
$$

which implies that

$$
\begin{aligned}
\|\boldsymbol{w}_1(t, c_1) - \boldsymbol{w}_1(0, c_1)\|_2 &= \gamma \int_0^t \|\boldsymbol{w}_1(s, c_1) - \boldsymbol{w}_1(0, c_1)\|_2\, ds + \int_0^t \int_0^s \|\mathbb{E}_{\boldsymbol{z}} \Delta_1^W(\boldsymbol{z}, c_1; \boldsymbol{W}(u))\|_2\, du\, ds \\
&\leq \gamma \int_0^t \|\boldsymbol{w}_1(s, c_1) - \boldsymbol{w}_1(0, c_1)\|_2\, ds + T^2 K_1(\gamma, T),
\end{aligned}
$$

where in the last step we use that, for some constant $K_1(\gamma, T)$ depending only on $\gamma, T$, $\|\mathbb{E}_{\boldsymbol{z}} \operatorname*{ess\,sup}_{c_1} \sup_{u \in [0,T]} \Delta_1^W(\boldsymbol{z}, c_1; \boldsymbol{W}(u))\|_2 \leq K_1(\gamma, T)$ by Lemma A.2. Thus, by Gronwall's lemma, we have that

$$
\sup_{t \in [0,T]} \|\boldsymbol{w}_1(t, c_1) - \boldsymbol{w}_1(0, c_1)\|_2 \leq e^{\gamma T} T^2 K_1(\gamma, T) := K_2(\gamma, T).
\tag{84}
$$

Now, by using again the definition of the mean-field ODE (74), we have that:

$$
\begin{aligned}
\|\boldsymbol{w}_1(t+\tau, c_1) - \boldsymbol{w}_1(t, c_1)\|_2 &= \left\| -\gamma \int_t^{t+\tau} (\boldsymbol{w}_1(s, c_1) - \boldsymbol{w}_1(0, c_1))\, ds - \int_t^{t+\tau} \int_0^s \mathbb{E}_{\boldsymbol{z}} \Delta_1^W(\boldsymbol{z}, c_1; \boldsymbol{W}(u))\, du\, ds \right\|_2 \\
&\leq \gamma \sup_{t \in [0,T]} \|\boldsymbol{w}_1(t, c_1) - \boldsymbol{w}_1(0, c_1)\|_2 \tau + K_1(\gamma, T) T \tau \\
&\leq K_2(\gamma, T)\tau + K_1(\gamma, T) T \tau,
\end{aligned}
$$

where in the second line we use that $\|\mathbb{E}_{\boldsymbol{z}} \operatorname*{ess\,sup}_{c_1} \sup_{u \in [0,T]} \Delta_1^W(\boldsymbol{z}, c_1; \boldsymbol{W}(u))\|_2 \leq K_1(\gamma, T)$ by Lemma A.2, and in the last passage we use (84). By setting $K(\gamma, T) = K_2(\gamma, T) + K_1(\gamma, T)T$, we obtain the desired result and the proof is complete. $\square$

By the above Lemma D.2 and Lemma A.2, we immediately get the following corollary.

**Corollary D.3.** *Under Assumptions (B1)-(B3), there exists a universal constant $K(\gamma, T)$ depending only on $\gamma, T$ such that, for any $t, \tau > 0$ with $t, t + \tau \leq T$, the following functions*

$$
f(\boldsymbol{x}; \boldsymbol{W}(t)), \quad H_2(\boldsymbol{x}, c_2; \boldsymbol{W}(t)), \quad \mathbb{E}_{C_2}\left[ \Delta_2^H(\boldsymbol{z}, C_2; \boldsymbol{W}(t)) w_2(t, c_1, C_2) \right]
$$

*are $K(\gamma, T)$-Lipschitz continuous in $t$. The same holds for the particle dynamics, i.e., the functions*

$$
f(\boldsymbol{x}; \boldsymbol{W}(t)), \quad H_2(\boldsymbol{x}, j_2; \boldsymbol{W}(t)), \quad \frac{1}{n_2} \sum_{j_2=1}^{n_2} \Delta_2^H(\boldsymbol{z}, j_2; \boldsymbol{W}(t)) w_2(t, j_1, j_2)
$$

*are $K(\gamma, T)$-Lipschitz continuous in $t$.*

*Proof of Corollary D.3.* We do the proof for $f(\boldsymbol{x}; \boldsymbol{W}(t))$, and the same argument applies to the other cases. By Lemma A.2, we have that

$$|f(\boldsymbol{x}; \boldsymbol{W}(t+\tau)) - f(\boldsymbol{x}; \boldsymbol{W}(t))| \leq K \operatorname*{ess\,sup}_{c_1, c_2}(|w_3(t+\tau, c_2)| \cdot |w_2(t+\tau, c_1, c_2)| \cdot \|\boldsymbol{w}_1(t+\tau, c_1) - \boldsymbol{w}_1(t, c_1)\|_2$$
$$+ |w_3(t+\tau, c_2)| \cdot |w_2(t+\tau, c_1, c_2) - w_2(t, c_1, c_2)| + |w_3(t+\tau, c_2) - w_3(t, c_2)|).$$

By Lemma D.2, we have that

$$\max(\|\boldsymbol{w}_1(t+\tau, c_1) - \boldsymbol{w}_1(t, c_1)\|_2, |w_2(t+\tau, c_1, c_2) - w_2(t, c_1, c_2)|, |w_3(t+\tau, c_2) - w_3(t, c_2)|) \leq K(\gamma, T)\tau.$$

Furthermore, by Lemma A.2, we have that:

$$\operatorname*{ess\,sup}_{c_2} |w_3(t+\tau, c_2)| \leq \sup_{t \in [0,T]} \operatorname*{ess\,sup}_{c_2} |w_3(t, c_2)| \leq K_{3,3}(\gamma, T),$$
$$\operatorname*{ess\,sup}_{c_1, c_2} |w_2(t+\tau, c_1, c_2)| \leq \sup_{t \in [0,T]} \operatorname*{ess\,sup}_{c_1, c_2} |w_2(t, c_1, c_2)| \leq K_{3,2}(\gamma, T).$$

Thus, we conclude

$$|f(\boldsymbol{x}; \boldsymbol{W}(t+\tau)) - f(\boldsymbol{x}; \boldsymbol{W}(t))| \leq K(\gamma, T)\tau,$$

which gives the desired result. $\square$

Now we are ready to prove Proposition D.1.

*Proof of Proposition D.1.* Let us recall that the quantity $\Delta_3^W$ is defined in (15). We start with computing the difference in the term $\Delta_3^W$:

$$|\mathbb{E}_{\boldsymbol{z}} \Delta_3^W(\boldsymbol{z}, C_2(j_2); \boldsymbol{W}(t)) - \mathbb{E}_{\boldsymbol{z}} \Delta_3^W(\boldsymbol{z}, j_2; \boldsymbol{W}^{PD}(t))|$$
$$\leq \mathbb{E}_{\boldsymbol{z}} |\Delta_3^W(\boldsymbol{z}, C_2(j_2); \boldsymbol{W}(t)) - \Delta_3^W(\boldsymbol{z}, j_2; \boldsymbol{W}^{PD}(t))|$$
$$= \mathbb{E}_{\boldsymbol{z}} |\partial_2 R(y, f(\boldsymbol{x}; \boldsymbol{W}(t)))\sigma_2(H_2(\boldsymbol{x}, C_2(j_2); \boldsymbol{W}(t))) - \partial_2 R(y, f(\boldsymbol{x}; \boldsymbol{W}^{PD}(t)))\sigma_2(H_2(\boldsymbol{x}, j_2; \boldsymbol{W}^{PD}(t)))| \tag{85}$$
$$\leq K\mathbb{E}_{\boldsymbol{z}} |f(\boldsymbol{x}; \boldsymbol{W}(t)) - f(\boldsymbol{x}; \boldsymbol{W}^{PD}(t))| + K|H_2(\boldsymbol{x}, C_2(j_2); \boldsymbol{W}(t)) - H_2(\boldsymbol{x}, j_2; \boldsymbol{W}^{PD}(t))|,$$

where in the last inequality we use the boundedness and Lipschitz continuity of $\partial_2 R$ and $\sigma_2$ obtained from Lemma A.2.

Similarly, for $\Delta_1^W, \Delta_2^W$, we have that

$$|\mathbb{E}_{\boldsymbol{z}} \Delta_2^W(\boldsymbol{z}, C_1(j_1), C_2(j_2); \boldsymbol{W}(t)) - \mathbb{E}_{\boldsymbol{z}} \Delta_2^W(\boldsymbol{z}, j_1, j_2; \boldsymbol{W}^{PD}(t))|$$
$$\leq \mathbb{E}_{\boldsymbol{z}} |\Delta_2^W(\boldsymbol{z}, C_1(j_1), C_2(j_2); \boldsymbol{W}(t)) - \Delta_2^W(\boldsymbol{z}, j_1, j_2; \boldsymbol{W}^{PD}(t))|$$
$$\leq \mathbb{E}_{\boldsymbol{z}} K|w_3(t, C_2(j_2))| \cdot \big(|f(\boldsymbol{x}; \boldsymbol{W}(t)) - f(\boldsymbol{x}; \boldsymbol{W}^{PD}(t))| + |H_2(\boldsymbol{x}, C_2(j_2); \boldsymbol{W}(t)) - H_2(\boldsymbol{x}, j_2; \boldsymbol{W}^{PD}(t))|\big)$$
$$+ |w_3(t, C_2(j_2))| \cdot \|\boldsymbol{w}_1(t, C_1(j_1)) - \boldsymbol{w}_1^{PD}(t, j_1)\|_2 + K|w_3(t, C_2(j_2)) - w_3^{PD}(t, j_2)| \tag{86}$$
$$\leq \mathbb{E}_{\boldsymbol{z}} K_{3,3}(\gamma, T) \big(|f(\boldsymbol{x}; \boldsymbol{W}(t)) - f(\boldsymbol{x}; \boldsymbol{W}^{PD}(t))| + |H_2(\boldsymbol{x}, C_2(j_2); \boldsymbol{W}(t)) - H_2(\boldsymbol{x}, j_2; \boldsymbol{W}^{PD}(t))|\big)$$
$$+ K_{3,3}(\gamma, T)\|\boldsymbol{w}_1(t, C_1(j_1)) - \boldsymbol{w}_1^{PD}(t, j_1)\|_2 + K|w_3(t, C_2(j_2)) - w_3^{PD}(t, j_2)|,$$

and that

$$|\mathbb{E}_{\boldsymbol{z}} \Delta_1^W(\boldsymbol{z}, C_1(j_1); \boldsymbol{W}(t)) - \mathbb{E}_{\boldsymbol{z}} \Delta_1^W(\boldsymbol{z}, j_1; \boldsymbol{W}^{PD}(t))|$$
$$\leq K \cdot \mathbb{E}_{\boldsymbol{z}} \left[ \left| \mathbb{E}_{C_2} \left[ \Delta_2^H(\boldsymbol{z}, C_2; \boldsymbol{W}(t)) w_2(t, C_1(j_1), C_2) \right] - \frac{1}{n_2} \sum_{j_2=1}^{n_2} \Delta_2^H(\boldsymbol{z}, j_2; \boldsymbol{W}^{PD}(t)) w_2^{PD}(t, j_1, j_2) \right| \right]$$
$$+ K \cdot \mathbb{E}_{\boldsymbol{z}} \left[ |\mathbb{E}_{C_2} \Delta_2^H(\boldsymbol{z}, C_2; \boldsymbol{W}(t)) w_2(t, C_1(j_1), C_2)| \right] \cdot \|\boldsymbol{w}_1(t, C_1(j_1)) - \boldsymbol{w}_1^{PD}(t, j_1)\|_2 \tag{87}$$
$$\leq K \cdot \mathbb{E}_{\boldsymbol{z}} \left[ \left| \mathbb{E}_{C_2} \left[ \Delta_2^H(\boldsymbol{z}, C_2; \boldsymbol{W}(t)) w_2(t, C_1(j_1), C_2) \right] - \frac{1}{n_2} \sum_{j_2=1}^{n_2} \Delta_2^H(\boldsymbol{z}, j_2; \boldsymbol{W}^{PD}(t)) w_2^{PD}(t, j_1, j_2) \right| \right]$$
$$+ K(T, \gamma) \cdot \|\boldsymbol{w}_1(t, C_1(j_1)) - \boldsymbol{w}_1^{PD}(t, j_1)\|_2.$$

Here, we remark that the expectation $\mathbb{E}_{C_2}[\Delta_2^H(\cdot, C_2; \cdot)w_2(\cdot, \cdot, C_2)]$ for the mean-field ODE corresponds to the average $\frac{1}{n_2}\sum_{j_2=1}^{n_2}\Delta_2^H(\cdot, j_2; \cdot)w_2(\cdot, \cdot, j_2)$ for the particle dynamics.

Now, our goal is to upper bound the following quantities:

$$|f(\boldsymbol{x}; \boldsymbol{W}(t)) - f(\boldsymbol{x}; \boldsymbol{W}^{PD}(t))|,$$
$$|H_2(\boldsymbol{x}, C_2(j_2); \boldsymbol{W}(t)) - H_2(\boldsymbol{x}, j_2; \boldsymbol{W}^{PD}(t))|,$$
$$\left|\mathbb{E}_{C_2}\left[\Delta_2^H(\boldsymbol{z}, C_2; \boldsymbol{W}(t))w_2(t, C_1(j_1), C_2)\right] - \frac{1}{n_2}\sum_{j_2=1}^{n_2}\Delta_2^H(\boldsymbol{z}, j_2; \boldsymbol{W}^{PD}(t))w_2^{PD}(t, j_1, j_2)\right|$$

To do so, we follow (Pham & Nguyen, 2021a, Appendix C.2, Proof of Theorem 14, Claim 2). Then, we have that, for any $\delta_1, \delta_2, \delta_3 > 0$,

$$\max\left\{|f(\boldsymbol{x}; \boldsymbol{W}(t)) - f(\boldsymbol{x}; \boldsymbol{W}^{PD}(t))|,\right.$$
$$\max_{j_2}|H_2(\boldsymbol{x}, C_2(j_2); \boldsymbol{W}(t)) - H_2(\boldsymbol{x}, j_2; \boldsymbol{W}^{PD}(t))|,$$
$$\left.\max_{j_1}\left|\mathbb{E}_{C_2}\left[\Delta_2^H(\boldsymbol{z}, C_2; \boldsymbol{W}(t))w_2(t, C_1(j_1), C_2)\right] - \frac{1}{n_2}\sum_{j_2=1}^{n_2}\Delta_2^H(\boldsymbol{z}, j_2; \boldsymbol{W}^{PD}(t))w_2^{PD}(t, j_1, j_2)\right|\right\}$$
$$\leq K_1(\gamma, T)\left(\mathcal{D}_T(\boldsymbol{W}, \boldsymbol{W}^{PD}) + \delta_1 + \delta_2 + \delta_3\right),$$

with probability at least

$$1 - \left(\frac{n_2}{\delta_1}\exp\left\{-\frac{n_1\delta_1^2}{K_1(\gamma, T)}\right\} + \frac{1}{\delta_2}\exp\left\{-\frac{n_2\delta_2^2}{K_1(\gamma, T)}\right\} + \frac{n_1}{\delta_3}\exp\left\{-\frac{n_2\delta_3^2}{K_1(\gamma, T)}\right\}\right).$$

By Corollary D.3, we know that $f(\boldsymbol{x}; \boldsymbol{W}(t)), H_2(\boldsymbol{x}, c_2; \boldsymbol{W}(t)), \mathbb{E}_{C_2}\left[\Delta_2^H(\boldsymbol{z}, C_2; \boldsymbol{W}(t))w_2(t, c_1, C_2)\right]$ are $K(\gamma, T)$-Lipschitz continuous, and the corresponding quantities for the particle dynamics are also $K(\gamma, T)$-Lipschitz continuous. Thus, by taking a union bound on $t \in \{0, \eta, ..., \lfloor T/\eta\rfloor\}$, we have

$$\max\sup_{t\in[0,T]}\left\{|f(\boldsymbol{x}; \boldsymbol{W}(t)) - f(\boldsymbol{x}; \boldsymbol{W}^{PD}(t))|,\right.$$
$$\max_{j_2}|H_2(\boldsymbol{x}, C_2(j_2); \boldsymbol{W}(t)) - H_2(\boldsymbol{x}, j_2; \boldsymbol{W}^{PD}(t))|,$$
$$\left.\max_{j_1}\left|\mathbb{E}_{C_2}\left[\Delta_2^H(\boldsymbol{z}, C_2; \boldsymbol{W}(t))w_2(t, C_1(j_1), C_2)\right] - \frac{1}{n_2}\sum_{j_2=1}^{n_2}\Delta_2^H(\boldsymbol{z}, j_2; \boldsymbol{W}^{PD}(t))w_2^{PD}(t, j_1, j_2)\right|\right\}$$
$$\leq K_2(\gamma, T)\left(\mathcal{D}_T(\boldsymbol{W}, \boldsymbol{W}^{PD}) + \delta_1 + \delta_2 + \delta_3 + \eta\right),$$

with probability at least

$$1 - \frac{T}{\eta}\left(\frac{n_2}{\delta_1}\exp\left\{-\frac{n_1\delta_1^2}{K_2(\gamma, T)}\right\} + \frac{1}{\delta_2}\exp\left\{-\frac{n_2\delta_2^2}{K_2(\gamma, T)}\right\} + \frac{n_1}{\delta_3}\exp\left\{-\frac{n_2\delta_3^2}{K_2(\gamma, T)}\right\}\right).$$

In particular, we pick

$$\eta = \frac{1}{\sqrt{n_{\max}}}, \quad \delta_1 = \frac{K_3(\gamma, T)}{\sqrt{n_1}}\left(\sqrt{\log n_{\max}} + \delta\right), \quad \delta_2 = \delta_3 = \frac{K_3(\gamma, T)}{\sqrt{n_2}}\left(\sqrt{\log n_{\max}} + \delta\right).$$

Then,

$$
\begin{aligned}
\max \sup_{t \in [0,T]} \Bigg\{ &|f(\boldsymbol{x}; \boldsymbol{W}(t)) - f(\boldsymbol{x}; \boldsymbol{W}^{PD}(t))|, \\
&\max_{j_2} |H_2(\boldsymbol{x}, C_2(j_2); \boldsymbol{W}(t)) - H_2(\boldsymbol{x}, j_2; \boldsymbol{W}^{PD}(t))|, \\
&\max_{j_1} \left| \mathbb{E}_{C_2} \left[ \Delta_2^H(\boldsymbol{z}, C_2; \boldsymbol{W}(t)) w_2(t, C_1(j_1), C_2) \right] - \frac{1}{n_2} \sum_{j_2=1}^{n_2} \Delta_2^H(\boldsymbol{z}, j_2; \boldsymbol{W}^{PD}(t)) w_2^{PD}(t, j_1, j_2) \right| \Bigg\} \\
&\leq K_4(\gamma, T) \left( \mathcal{D}_T(\boldsymbol{W}, \boldsymbol{W}^{PD}) + \frac{K_4(\gamma, T)}{\sqrt{n_{\min}}} \left( \sqrt{\log n_{\max}} + \delta \right) \right)
\end{aligned}
\tag{88}
$$

with probability at least $1 - \exp(-\delta^2)$.

Next, we combine (88) with (85), (86) and (87) to provide high-probability bounds on $\Delta_3^W, \Delta_2^W, \Delta_1^W$. By recalling the definition of the mean-field ODE (74) and the analogous definition of the particle dynamics (76), we finally obtain that, for all $t \leq T$, with probability at least $1 - \exp(-\delta^2)$,

$$
\mathcal{D}_t(\boldsymbol{W}, \boldsymbol{W}^{PD}) \leq \frac{K(\gamma, T)}{\sqrt{n_{\min}}} \left( \sqrt{\log n_{\max} T} + \delta \right) + \gamma \int_0^t \mathcal{D}_s(\boldsymbol{W}, \boldsymbol{W}^{PD}) \, ds + \int_0^t \int_0^s \mathcal{D}_u(\boldsymbol{W}, \boldsymbol{W}^{PD}) \, du \, ds.
$$

An application of Corollary F.4 gives the desired result (82) and concludes the proof. $\square$

### D.2 Bound between particle dynamics and heavy ball dynamics

In this part we bound the difference between the particle dynamics defined in (76) and the heavy ball dynamics defined in (77). We recall that the distance we aim to bound is defined in (79).

**Proposition D.4.** *Under Assumptions (B1)-(B3), there exists a universal constant $K(\gamma, T)$ depending only on $\gamma, T$ such that*

$$
\mathcal{D}_T(\boldsymbol{W}^{PD}, \boldsymbol{W}^{HB}) \leq K(\gamma, T)\varepsilon.
\tag{89}
$$

*Proof of Proposition D.4.* Note that the heavy ball dynamics is just a discretization of the particle dynamics, so we first bound the difference at each time point $k\varepsilon$. By a second order Taylor expansion, we have the following approximation for the particle dynamics. We do the computation for $w_3$ as a representative, and the proofs for $\boldsymbol{w}_1, w_2$ are the same.

We have that

$$
w_3^{PD}((k+1)\varepsilon, j_2) = w_3^{PD}(k\varepsilon, j_2) + \partial_t w_3^{PD}(k\varepsilon, j_2)\varepsilon + \frac{1}{2}\partial_t^2 w_3^{PD}(k\varepsilon, j_2)\varepsilon^2 + O(\varepsilon^3).
\tag{90}
$$

Also by Taylor expansion, we have that

$$
\partial_t w_3^{PD}(k\varepsilon, j_2)\varepsilon = w_3^{PD}(k\varepsilon, j_2) - w_3^{PD}((k-1)\varepsilon, j_2) + \frac{1}{2}\partial_t^2 w_3^{PD}(k\varepsilon, j_2)\varepsilon^2 + O(\varepsilon^3).
\tag{91}
$$

By plugging (91) into (90), we obtain

$$
\begin{aligned}
w_3^{PD}((k+1)\varepsilon, j_2) &= w_3^{PD}(k\varepsilon, j_2) + w_3^{PD}(k\varepsilon, j_2) - w_3^{PD}((k-1)\varepsilon, j_2) + \partial_t^2 w_3^{PD} k\varepsilon, j_2)\varepsilon^2 + O(\varepsilon^3) \\
&= w_3^{PD}(k\varepsilon, j_2) + w_3^{PD}(k\varepsilon, j_2) - w_3^{PD}((k-1)\varepsilon, j_2) + (-\gamma\partial_t w_3^{PD}(\varepsilon, j_2) - \mathbb{E}_{\boldsymbol{z}}\Delta_3^W(\boldsymbol{z}, j_2; \boldsymbol{W}^{PD}(k\varepsilon)))\varepsilon^2 + O(\varepsilon^3) \\
&= w_3^{PD}(k\varepsilon, j_2) + (1 - \gamma\varepsilon)(w_3^{PD}(k\varepsilon, j_2) - w_3^{PD}((k-1)\varepsilon, j_2)) - \mathbb{E}_{\boldsymbol{z}}\Delta_3^W(\boldsymbol{z}, j_2; \boldsymbol{W}^{PD}(k\varepsilon))\varepsilon^2 + O(\varepsilon^3),
\end{aligned}
$$

where in the last step we use again (91).

By unrolling the recursion, we can write the particle dynamics in the following form:

$$w_3^{PD}(k\varepsilon, j_2) = w_3^{PD}(0, j_2) - \sum_{l=0}^{k-1} c_l^{(k)} \mathbb{E}_{\boldsymbol{z}} \Delta_3^W(\boldsymbol{z}, j_2; \boldsymbol{W}^{PD}(l\varepsilon)) + O(\varepsilon),$$

where

$$c_l^{(k)} = \varepsilon^2 \sum_{i=0}^{k-1-l} (1 - \gamma\varepsilon)^i = \frac{1 - (1 - \gamma\varepsilon)^{k-l}}{\gamma\varepsilon} \varepsilon^2 \le \frac{\varepsilon}{\gamma}.$$

Similarly for $w_2, \boldsymbol{w}_1$, we have:

$$w_2^{PD}(k\varepsilon, j_1, j_2) = w_2^{PD}(0, j_1, j_2) - \sum_{l=0}^{k-1} c_l^{(k)} \mathbb{E}_{\boldsymbol{z}} \Delta_2^W(\boldsymbol{z}, j_1, j_2; \boldsymbol{W}^{PD}(l\varepsilon)) + O(\varepsilon),$$

$$\boldsymbol{w}_1^{PD}(k\varepsilon, j_1) = \boldsymbol{w}_1^{PD}(0, j_1) - \sum_{l=0}^{k-1} c_l^{(k)} \mathbb{E}_{\boldsymbol{z}} \Delta_1^W(\boldsymbol{z}, j_1; \boldsymbol{W}^{PD}(l\varepsilon)) + O(\varepsilon).$$

We can write analogous expressions for the heavy ball dynamics:

$$w_3^{HB}(k, j_2) = w_3^{HB}(0, j_2) - \sum_{l=0}^{k-1} c_l^{(k)} \mathbb{E}_{\boldsymbol{z}} \Delta_3^W(\boldsymbol{z}, j_2; \boldsymbol{W}^{HB}(l)),$$

$$w_2^{HB}(k, j_1, j_2) = w_2^{HB}(0, j_1, j_2) - \sum_{l=0}^{k-1} c_l^{(k)} \mathbb{E}_{\boldsymbol{z}} \Delta_2^W(\boldsymbol{z}, j_1, j_2; \boldsymbol{W}^{HB}(l)), \qquad (92)$$

$$\boldsymbol{w}_1^{HB}(k, j_1) = \boldsymbol{w}_1^{HB}(0, j_1) - \sum_{l=0}^{k-1} c_l^{(k)} \mathbb{E}_{\boldsymbol{z}} \Delta_1^W(\boldsymbol{z}, j_1; \boldsymbol{W}^{HB}(l)).$$

Let us define the following notation, for $k \in \left\{1, ..., \lfloor \frac{T}{\varepsilon} \rfloor\right\}$,

$$\mathcal{D}_T(\boldsymbol{W}^{PD}, \boldsymbol{W}^{HB}; k) = \max\{\|\boldsymbol{w}_1^{HB}(k, j_1) - \boldsymbol{w}_1^{PD}(k\varepsilon, j_1)\|_2,$$
$$|w_2^{HB}(k, j_1, j_2) - w_2^{PD}(k\varepsilon, j_1, j_2)|,$$
$$|w_3^{HB}(k, j_2) - w_3^{PD}(k\varepsilon, j_2)| : j_1 \in [n_1], j_2 \in [n_2]\}\}.$$

Recall that, by construction, $w_3^{PD}(0, j_2) = w_3^{HB}(0, j_2)$, $w_2^{PD}(0, j_1, j_2) = w_2^{HB}(0, j_1, j_2)$ and $\boldsymbol{w}_1^{PD}(0, j_1) = \boldsymbol{w}_1^{HB}(0, j_1)$ for all $j_1, j_2$. Thus, by computing the difference between $\boldsymbol{w}_1^{PD}, w_2^{PD}, w_3^{PD}$ and $\boldsymbol{w}_1^{HB}, w_2^{HB}, w_3^{HB}$, we have that $\mathcal{D}_T(\boldsymbol{W}^{PD}, \boldsymbol{W}^{HB}; k)$ satisfies the following induction inequality:

$$\mathcal{D}_T(\boldsymbol{W}^{PD}, \boldsymbol{W}^{HB}; k) \le \sum_{l=0}^{k-1} c_l^{(k)} K_1(\gamma, T) \mathcal{D}_T(\boldsymbol{W}^{PD}, \boldsymbol{W}^{HB}; l) + O(\varepsilon), \qquad (93)$$

where we have used the Lipschitz continuity of $\Delta_3^W, \Delta_2^W$ and $\Delta_1^W$ obtained via Lemma A.2. Thus, by the discrete Gronwall's lemma, we obtain that, for any $k \in \left\{1, ..., \lfloor \frac{T}{\varepsilon} \rfloor\right\}$,

$$\mathcal{D}_T(\boldsymbol{W}^{PD}, \boldsymbol{W}^{HB}; k) \le K_2(\gamma, T)\varepsilon$$

Finally, an application of Lemma D.2 gives that $\boldsymbol{w}_1^{PD}, w_2^{PD}, w_3^{PD}$ are $K_3(\gamma, T)$-Lipschitz continuous in time. Thus, for any $t \le T$,

$$|w_3^{PD}(t, j_2) - w_3^{HB}(\lfloor t/\varepsilon \rfloor, j_2)| \le |w_3^{PD}(t, j_2) - w_3^{PD}(\lfloor t/\varepsilon \rfloor \varepsilon, j_2)| + |w_3^{PD}(\lfloor t/\varepsilon \rfloor \varepsilon, j_2) - w_3^{HB}(\lfloor t/\varepsilon \rfloor, j_2)|$$
$$\le |w_3^{PD}(\lfloor t/\varepsilon \rfloor \varepsilon, j_2) - w_3^{HB}(\lfloor t/\varepsilon \rfloor, j_2)| + K_3(\gamma, T)\varepsilon.$$

Similar results hold also for $|w_2^{PD}(t, j_1, j_2) - w_2^{HB}(\lfloor t/\varepsilon \rfloor, j_1, j_2)|$ and $|w_1^{PD}(t, j_1) - w_1^{HB}(\lfloor t/\varepsilon \rfloor, j_1)|$. As a result, we conclude that

$$\mathcal{D}_T(\boldsymbol{W}^{PD}, \boldsymbol{W}^{HB}) \le \max_{k \in \left\{1, ..., \lfloor \frac{T}{\varepsilon} \rfloor\right\}} \mathcal{D}_T(\boldsymbol{W}^{PD}, \boldsymbol{W}^{HB}; k) + K_3(\gamma, T)\varepsilon \le K(\gamma, T)\varepsilon,$$

which gives the desired result. □

### D.3 Bound between heavy ball dynamics and stochastic heavy ball dynamics

In this part we bound the difference between the heavy ball dynamics defined in (77) and the stochastic heavy ball dynamics defined in (75). We recall that the distance we aim to bound is defined in (80).

**Proposition D.5.** *Under Assumptions (B1)-(B3), we have that, with probability at least $1 - \exp(-\delta^2)$,*

$$\mathcal{D}_{T,\epsilon}(\boldsymbol{W}^{HB}, \boldsymbol{W}^{SHB}) \leq K(\gamma, T)\sqrt{\varepsilon}(\sqrt{D + \log(n_1 n_2)} + \delta), \tag{94}$$

*where $K(\gamma, T)$ is a universal constant depending only on $\gamma, T$.*

Before proving Proposition D.5, we state and prove a result concerning the boundedness of the SHB dynamics.

**Lemma D.6** (Boundedness of the SHB dynamics)**.** *Under Assumptions (B1)-(B3), we have that, for any $k \in \lfloor \frac{T}{\varepsilon} \rfloor$,*

$$|w_3^{SHB}(k, j_2)| \leq K\left(1 + \frac{1}{\gamma}\right)T,$$

$$|w_2^{SHB}(k, j_1, j_2)| \leq K\left(1 + \frac{1}{\gamma}\right)\left(1 + \frac{T^2}{\gamma}\right),$$

*where $K$ is a universal constant.*

*Proof.* By following passages analogous to those leading to (92), we have that the SHB dynamics can be written as

$$w_3^{SHB}(k, j_2) = w_3^{SHB}(0, j_2) - \sum_{l=0}^{k-1} c_l^{(k)} \Delta_3^W(\boldsymbol{z}(l), j_2; \boldsymbol{W}^{SHB}(l)),$$

$$w_2^{SHB}(k, j_1, j_2) = w_2^{SHB}(0, j_1, j_2) - \sum_{l=0}^{k-1} c_l^{(k)} \Delta_2^W(\boldsymbol{z}(l), j_1, j_2; \boldsymbol{W}^{SHB}(l)). \tag{95}$$

Recall that

$$\Delta_3^W(\boldsymbol{z}(l), j_2; \boldsymbol{W}^{SHB}(l)) = \partial_2 R(y(l), f(\boldsymbol{x}(l); \boldsymbol{W}^{SHB}(l))) \cdot \sigma_2(H_2(\boldsymbol{x}(l), j_2; \boldsymbol{W}^{SHB}(l))),$$

which implies that

$$|\Delta_3^W(\boldsymbol{z}(l), j_2; \boldsymbol{W}^{SHB}(l))| = |\partial_2 R(y(l), f(\boldsymbol{x}(l); \boldsymbol{W}^{SHB}(l))) \cdot \sigma_2(H_2(\boldsymbol{x}(l), j_2; \boldsymbol{W}^{SHB}(l)))| \leq K.$$

Thus, we have

$$|w_3^{SHB}(k, j_2)| \leq |w_3^{SHB}(0, j_2)| + \sum_{l=0}^{k-1} c_l^{(k)} K \leq K + \frac{k\varepsilon}{\gamma} K \leq K\left(1 + \frac{1}{\gamma}\right)T,$$

where in the last step we use that $k\varepsilon \leq T$.

For $|w_2^{SHB}(k, j_1, j_2)|$, we recall that

$$|\Delta_2^W(\boldsymbol{z}(l), j_1, j_2; \boldsymbol{W}^{SHB}(l))|$$
$$= |\partial_2 R(y(l), f(\boldsymbol{x}(l); \boldsymbol{W}^{SHB}(l))) \cdot w_3^{SHB}(l, j_2) \cdot \sigma_2'(H_2(\boldsymbol{x}(l), j_2; \boldsymbol{W}^{SHB}(l)))\sigma_1((\boldsymbol{w}_1^{SHB}(l, j_1))^T \boldsymbol{x}(l))|$$
$$\leq K|w_3^{SHB}(l, j_2)|.$$

Thus, we have

$$|w_2^{SHB}(k, j_1, j_2)| \leq |w_2^{SHB}(0, j_1, j_2)| + \sum_{l=0}^{k-1} c_l^{(k)} |w_3^{SHB}(l, j_2)|$$

$$\leq K + K\left(1 + \frac{1}{\gamma}\right)T\frac{k\varepsilon}{\gamma}$$

$$\leq K\left(1 + \frac{1}{\gamma}\right)\left(1 + \frac{T^2}{\gamma}\right),$$

which gives the desired result. $\qquad \square$

*Proof of Proposition D.5.* Throughout this argument, we fix $\varepsilon$ and consider $k \in \lfloor \frac{T}{\varepsilon} \rfloor$. Recall that the HB and SHB dynamics can be written as in (92) and (95), respectively. Furthermore,

$$\boldsymbol{w}_1^{SHB}(k, j_1) = \boldsymbol{w}_1^{SHB}(0, j_1) - \sum_{l=0}^{k-1} c_l^{(k)} \Delta_1^W(\boldsymbol{z}(l), j_1; \boldsymbol{W}^{SHB}(l)). \tag{96}$$

Recall that, by construction, $w_3^{HB}(0, j_2) = w_3^{SHB}(0, j_2)$, $w_2^{HB}(0, j_1, j_2) = w_2^{SHB}(0, j_1, j_2)$ and $\boldsymbol{w}_1^{HB}(0, j_1) = \boldsymbol{w}_1^{SHB}(0, j_1)$ for all $j_1, j_2$. Thus, by computing the difference between the expressions in (92) and (95)-(96), we have

$$|w_3^{HB}(k, j_2) - w_3^{SHB}(k, j_2)| = \left| \sum_{l=0}^{k-1} c_l^{(k)} \left( \mathbb{E}_{\boldsymbol{z}}[\Delta_3^W(\boldsymbol{z}, j_2; \boldsymbol{W}^{HB}(l))] - \Delta_3^W(\boldsymbol{z}(l), j_2; \boldsymbol{W}^{SHB}(l)) \right) \right|$$

$$\leq \left| \sum_{l=0}^{k-1} c_l^{(k)} \left( \mathbb{E}_{\boldsymbol{z}}[\Delta_3^W(\boldsymbol{z}, j_2; \boldsymbol{W}^{HB}(l))] - \mathbb{E}_{\boldsymbol{z}}[\Delta_3^W(\boldsymbol{z}, j_2; \boldsymbol{W}^{SHB}(l))] \right) \right| \tag{97}$$

$$+ \left| \sum_{l=0}^{k-1} c_l^{(k)} \left( \mathbb{E}_{\boldsymbol{z}}[\Delta_3^W(\boldsymbol{z}, j_2; \boldsymbol{W}^{SHB}(l))] - \Delta_3^W(\boldsymbol{z}(l), j_2; \boldsymbol{W}^{SHB}(l)) \right) \right|, \tag{98}$$

$$|w_2^{HB}(k, j_1, j_2) - w_2^{SHB}(k, j_1, j_2)| = \left| \sum_{l=0}^{k-1} c_l^{(k)} \left( \mathbb{E}_{\boldsymbol{z}}[\Delta_2^W(\boldsymbol{z}, j_1, j_2; \boldsymbol{W}^{HB}(l))] - \Delta_2^W(\boldsymbol{z}(l), j_1, j_2; \boldsymbol{W}^{SHB}(l)) \right) \right|$$

$$\leq \left| \sum_{l=0}^{k-1} c_l^{(k)} \left( \mathbb{E}_{\boldsymbol{z}}[\Delta_2^W(\boldsymbol{z}, j_1, j_2; \boldsymbol{W}^{HB}(l))] - \mathbb{E}_{\boldsymbol{z}}[\Delta_2^W(\boldsymbol{z}, j_1, j_2; \boldsymbol{W}^{SHB}(l))] \right) \right| \tag{99}$$

$$+ \left| \sum_{l=0}^{k-1} c_l^{(k)} \left( \mathbb{E}_{\boldsymbol{z}}[\Delta_2^W(\boldsymbol{z}, j_1, j_2; \boldsymbol{W}^{SHB}(l))] - \Delta_2^W(\boldsymbol{z}(l), j_1, j_2; \boldsymbol{W}^{SHB}(l)) \right) \right|, \tag{100}$$

$$\|\boldsymbol{w}_1^{HB}(k, j_1) - \boldsymbol{w}_1^{SHB}(k, j_1)\|_2 = \left\| \sum_{l=0}^{k-1} c_l^{(k)} \left( \mathbb{E}_{\boldsymbol{z}}[\Delta_1^W(\boldsymbol{z}, j_1; \boldsymbol{W}^{HB}(l))] - \Delta_1^W(\boldsymbol{z}(l), j_1; \boldsymbol{W}^{SHB}(l)) \right) \right\|_2$$

$$\leq \left\| \sum_{l=0}^{k-1} c_l^{(k)} \left( \mathbb{E}_{\boldsymbol{z}}[\Delta_1^W(\boldsymbol{z}, j_1; \boldsymbol{W}^{HB}(l))] - \mathbb{E}_{\boldsymbol{z}}[\Delta_1^W(\boldsymbol{z}, j_1; \boldsymbol{W}^{SHB}(l))] \right) \right\|_2 \tag{101}$$

$$+ \left\| \sum_{l=0}^{k-1} c_l^{(k)} \left( \mathbb{E}_{\boldsymbol{z}}[\Delta_1^W(\boldsymbol{z}, j_1; \boldsymbol{W}^{SHB}(l))] - \Delta_1^W(\boldsymbol{z}(l), j_1; \boldsymbol{W}^{SHB}(l)) \right) \right\|_2. \tag{102}$$

To bound (97), (99) and (101), we use the Lipschitz continuity of $\Delta_3^W$, $\Delta_2^W$ and $\Delta_1^W$, together with the fact that $c_l^{(k)} \leq \varepsilon/\gamma$. In particular,

$$\left| \sum_{l=0}^{k-1} c_l^{(k)} \left( \mathbb{E}_{\boldsymbol{z}} \Delta_3^W(\boldsymbol{z}, j_2; \boldsymbol{W}^{HB}(l)) - \mathbb{E}_{\boldsymbol{z}} \Delta_3^W(\boldsymbol{z}, j_2; \boldsymbol{W}^{SHB}(l)) \right) \right|$$

$$\leq \frac{\varepsilon}{\gamma} \sum_{l=0}^{k-1} \left| \left( \mathbb{E}_{\boldsymbol{z}}[\Delta_3^W(\boldsymbol{z}, j_2; \boldsymbol{W}^{HB}(l))] - \mathbb{E}_{\boldsymbol{z}}[\Delta_3^W(\boldsymbol{z}, j_2; \boldsymbol{W}^{SHB}(l))] \right) \right| \tag{103}$$

$$\leq K(\gamma, T) \frac{\varepsilon}{\gamma} \sum_{l=0}^{k-1} \mathcal{D}_{l\varepsilon, \varepsilon}(\boldsymbol{W}^{HB}, \boldsymbol{W}^{SHB}).$$

Similarly, we have

$$
\left| \sum_{l=0}^{k-1} c_l^{(k)} \left( \mathbb{E}_{\boldsymbol{z}}[\Delta_2^W(\boldsymbol{z}, j_1, j_2; \boldsymbol{W}^{HB}(l))] - \mathbb{E}_{\boldsymbol{z}}[\Delta_2^W(\boldsymbol{z}, j_1, j_2; \boldsymbol{W}^{SHB}(l))] \right) \right| \leq K(\gamma, T) \frac{\varepsilon}{\gamma} \sum_{l=0}^{k-1} \mathcal{D}_{l\varepsilon, \varepsilon}(\boldsymbol{W}^{HB}, \boldsymbol{W}^{SHB}),
$$
$$
\left\| \sum_{l=0}^{k-1} c_l^{(k)} \left( \mathbb{E}_{\boldsymbol{z}} \Delta_1^W(\boldsymbol{z}, j_1; \boldsymbol{W}^{HB}(l)) - \mathbb{E}_{\boldsymbol{z}} \Delta_1^W(\boldsymbol{z}, j_1; \boldsymbol{W}^{SHB}(l)) \right) \right\|_2 \leq K(\gamma, T) \frac{\varepsilon}{\gamma} \sum_{l=0}^{k-1} \mathcal{D}_{l\varepsilon, \varepsilon}(\boldsymbol{W}^{HB}, \boldsymbol{W}^{SHB}).
$$
$$(104)$$

To bound (102),(100) and (98), we first define the filtration $\mathcal{F}_3(k)$ as the sigma-algebra generated by $(\{w_3(0, j_2)\}_{j_2 \in [n_2]}, \boldsymbol{z}(0), ..., \boldsymbol{z}(k))$. We define the filtration $\mathcal{F}_2(k), \mathcal{F}_1(k)$ in the same way. Let us recall that, in a one-pass algorithm, we take i.i.d. samples at each step and, hence, we can write, for all $l \in \{1, ..., \lfloor \frac{T}{\varepsilon} \rfloor\}$,

$$
\mathbb{E}_{\boldsymbol{z}(l)} \left[ \Delta_3^W(\boldsymbol{z}(l), j_2; \boldsymbol{W}^{SHB}(l)) \big| \mathcal{F}_3(l-1) \right] = \mathbb{E}_{\boldsymbol{z}} \Delta_3^W(\boldsymbol{z}, j_2; \boldsymbol{W}^{SHB}(l)),
$$
$$
\mathbb{E}_{\boldsymbol{z}(l)} \left[ \Delta_2^W(\boldsymbol{z}(l), j_1, j_2; \boldsymbol{W}^{SHB}(l)) \big| \mathcal{F}_2(l-1) \right] = \mathbb{E}_{\boldsymbol{z}} \Delta_2^W(\boldsymbol{z}, j_1, j_2; \boldsymbol{W}^{SHB}(l)),
$$
$$
\mathbb{E}_{\boldsymbol{z}(l)} \left[ \Delta_1^W(\boldsymbol{z}(l), j_1; \boldsymbol{W}^{SHB}(l)) \big| \mathcal{F}_1(l-1) \right] = \mathbb{E}_{\boldsymbol{z}} \Delta_1^W(\boldsymbol{z}, j_1; \boldsymbol{W}^{SHB}(l)).
$$

Clearly, we have that $\{\Delta_3^W(\boldsymbol{z}(l), j_2; \boldsymbol{W}^{SHB}(l)), l \in \{1, ..., \lfloor \frac{T}{\varepsilon} \rfloor\}\}$ are mutually independent, which implies that

$$
\Delta_3^W(\boldsymbol{z}(l), j_2; \boldsymbol{W}^{SHB}(l)) - \mathbb{E}_{\boldsymbol{z}} \Delta_3^W(\boldsymbol{z}, j_2; \boldsymbol{W}^{SHB}(l))
$$

is a martingale difference with respect to the filtration $\mathcal{F}_3(l)$. Thus, $\{\sum_{l=0}^{k-1} c_l^{(k)} \Delta_3^W(\boldsymbol{z}(l), j_2; \boldsymbol{W}^{SHB}(l)) - \mathbb{E}_{\boldsymbol{z}} \Delta_3^W(\boldsymbol{z}, j_2; \boldsymbol{W}^{SHB}(l)) | k \in \lfloor \frac{T}{\varepsilon} \rfloor\}$ is a martingale (same for $\Delta_2^W$ and $\Delta_1^W$). Next, we show that the martingale differences are bounded, so that we can use martingale convergence results to bound these terms.

Combining Lemma D.6 with the same strategy of the a-priori estimations of Lemma A.2, we have the following upper bounds:

$$
|\mathbb{E}_{\boldsymbol{z}} \Delta_3^W(\boldsymbol{z}, j_2; \boldsymbol{W}^{SHB}(k)) - \Delta_3^W(\boldsymbol{z}(k), j_2; \boldsymbol{W}^{SHB}(k))| \leq K_1,
$$
$$
|\mathbb{E}_{\boldsymbol{z}} \Delta_2^W(\boldsymbol{z}, j_1, j_2; \boldsymbol{W}^{SHB}(k)) - \Delta_2^W(\boldsymbol{z}(k), j_1, j_2; \boldsymbol{W}^{SHB}(k))| \leq K_1(\gamma, T),
$$
$$
|\mathbb{E}_{\boldsymbol{z}} \Delta_1^W(\boldsymbol{z}, j_1; \boldsymbol{W}^{SHB}(k)) - \Delta_1^W(\boldsymbol{z}(k), j_1; \boldsymbol{W}^{SHB}(k))| \leq K_1(\gamma, T).
$$

Thus, an application of Lemma F.2 gives

$$
\Pr \left[ \max_{k \in \lfloor T/\varepsilon \rfloor} \left| \sum_{l=0}^{k-1} c_l^{(k)} \left( \mathbb{E}_{\boldsymbol{z}} \Delta_3^W(\boldsymbol{z}, j_2; \boldsymbol{W}^{SHB}(l)) - \Delta_3^W(\boldsymbol{z}(l), j_2; \boldsymbol{W}^{SHB}(l)) \right) \right| \geq K\sqrt{T}\epsilon(1 + \delta_3) \right] \leq \exp(-\delta_3^2),
$$
$$
\Pr \left[ \max_{k \in \lfloor T/\varepsilon \rfloor} \left| \sum_{l=0}^{k-1} c_l^{(k)} \left( \mathbb{E}_{\boldsymbol{z}} \Delta_2^W(\boldsymbol{z}, j_1, j_2; \boldsymbol{W}^{SHB}(l)) - \Delta_2^W(\boldsymbol{z}(l), j_1, j_2; \boldsymbol{W}^{SHB}(l)) \right) \right| \geq K(\gamma, T)\sqrt{T}\epsilon(1 + \delta_2) \right]
$$
$$
\leq \exp(-\delta_2^2),
$$
$$
\Pr \left[ \max_{k \in \lfloor T/\varepsilon \rfloor} \left\| \sum_{l=0}^{k-1} c_l^{(k)} \left( \mathbb{E}_{\boldsymbol{z}} \Delta_1^W(\boldsymbol{z}, j_1; \boldsymbol{W}^{SHB}(l)) - \Delta_1^W(\boldsymbol{z}(l), j_1; \boldsymbol{W}^{SHB}(l)) \right) \right\|_2 \geq K(\gamma, T)\sqrt{T}\epsilon(\sqrt{D} + \delta_1) \right]
$$
$$
\leq \exp(-\delta_1^2).
$$

By taking a union bound over $j_1, j_2$, we have that, with probability at least $1 - \exp(-\delta^2)$,

$$
\max_{j_1, j_2} \max_{k \in \lfloor T/\varepsilon \rfloor} \left\{ \left| \sum_{l=0}^{k-1} c_l^{(k)} \left( \mathbb{E}_{\boldsymbol{z}} \Delta_3^W(\boldsymbol{z}, j_2; \boldsymbol{W}^{SHB}(l)) - \Delta_3^W(\boldsymbol{z}(l), j_2; \boldsymbol{W}^{SHB}(l)) \right) \right|, \right.
$$
$$
\left| \sum_{l=0}^{k-1} c_l^{(k)} \left( \mathbb{E}_{\boldsymbol{z}} \Delta_2^W(\boldsymbol{z}, j_1, j_2; \boldsymbol{W}^{SHB}(l)) - \Delta_2^W(\boldsymbol{z}(l), j_1, j_2; \boldsymbol{W}^{SHB}(l)) \right) \right|,
$$
$$
\left. \left\| \sum_{l=0}^{k-1} c_l^{(k)} \left( \mathbb{E}_{\boldsymbol{z}} \Delta_1^W(\boldsymbol{z}, j_1; \boldsymbol{W}^{SHB}(l)) - \Delta_1^W(\boldsymbol{z}(l), j_1; \boldsymbol{W}^{SHB}(l)) \right) \right\|_2 \right\} \leq K(\gamma, T) \sqrt{T} \epsilon (\sqrt{D \log(n_1 n_2)} + \delta).
$$

Combining the results, we conclude that, with probability at least $1 - \exp(-\delta^2)$,

$$
\mathcal{D}_{T,\varepsilon}(\boldsymbol{W}^{HB}, \boldsymbol{W}^{SHB}) \leq K(\gamma, T) \sqrt{T} \epsilon (\sqrt{D \log(n_1 n_2)} + \delta) + K(\gamma, T) \frac{\varepsilon}{\gamma} \sum_{l=0}^{k-1} \mathcal{D}_{l\varepsilon,\varepsilon}(\boldsymbol{W}^{HB}, \boldsymbol{W}^{SHB}).
$$

An application of the discrete Gronwall's lemma gives the desired result (94) and concludes the proof. $\quad\square$

### D.4 Proof of Theorem 5.2

*Proof of Theorem 5.2.* The proof follows from combining Proposition D.1, D.4, D.5 and the fact that:

$$
\mathcal{D}_T(\boldsymbol{W}, \boldsymbol{W}^{SHB}) \leq \mathcal{D}_T(\boldsymbol{W}, \boldsymbol{W}^{PD}) + \mathcal{D}_T(\boldsymbol{W}^{PD}, \boldsymbol{W}^{HB}) + \mathcal{D}_{T,\varepsilon}(\boldsymbol{W}^{HB}, \boldsymbol{W}^{SHB}).
$$

$\hfill\square$

## E Global convergence of the mean-field ODE

In this section, we aim to prove the global convergence result through the recipe below:

1. We show the following degeneracy property for the mean-field ODE: there exist deterministic functions $\boldsymbol{w}_1^*(\cdot, \cdot) : \mathbb{R}^{\geq 0} \times \mathbb{R}^D \to \mathbb{R}^D, w_2^*(\cdot, \cdot, \cdot, \cdot) : \mathbb{R}^{\geq 0} \times \mathbb{R}^D \times \mathbb{R} \times \mathbb{R} \to \mathbb{R}, w_3^*(\cdot, \cdot) : \mathbb{R}^{\geq 0} \times \mathbb{R} \to \mathbb{R}$ such that

$$
\begin{aligned}
\boldsymbol{w}_1(t, C_1) &= \boldsymbol{w}_1^*(t, \boldsymbol{w}_1(0, C_1)), \\
w_2(t, C_1, C_2) &= w_2^*(t, \boldsymbol{w}_1(0, C_1), w_2(0, C_1, C_2), w_3(0, C_2)), \\
w_3(t, C_2) &= w_3^*(t, w_3(0, C_2)).
\end{aligned} \tag{105}
$$

2. We show that *(i)* $\boldsymbol{w}_1^*(\cdot, \cdot)$ is continuous in both arguments for any finite $t$, and that *(ii)* if $\boldsymbol{w}_1(0, C_1)$ is full support, then $\boldsymbol{w}_1(t, C_1)$ is full support for any finite $t$.

3. Combining the argument that $\boldsymbol{w}_1(t, C_1)$ is full support for all finite $t$ and the mode of convergence assumption, we show that the mean-field ODE must converge to the global minimum.

We first show the degeneracy property of the mean-field ODE in the following lemma:

**Lemma E.1.** *Under Assumptions (B1) - (B3), there exist deterministic functions $\boldsymbol{w}_1^*(\cdot, \cdot) : \mathbb{R}^{\geq 0} \times \mathbb{R}^D \to \mathbb{R}^D, w_2^*(\cdot, \cdot, \cdot, \cdot) : \mathbb{R}^{\geq 0} \times \mathbb{R}^D \times \mathbb{R} \times \mathbb{R} \to \mathbb{R}, w_3^*(\cdot, \cdot) : \mathbb{R}^{\geq 0} \times \mathbb{R} \to \mathbb{R}$ such that*

$$
\begin{aligned}
\boldsymbol{w}_1(t, C_1) &= \boldsymbol{w}_1^*(t, \boldsymbol{w}_1(0, C_1)), \\
w_2(t, C_1, C_2) &= w_2^*(t, \boldsymbol{w}_1(0, C_1), w_2(0, C_1, C_2), w_3(0, C_2)), \\
w_3(t, C_2) &= w_3^*(t, w_3(0, C_2)).
\end{aligned}
$$

*Proof of Lemma E.1.* We follow the proof in (Pham & Nguyen, 2021a, Appendix D.2). To shorten the notations, we make the following definition: we define the sigma-algebras generated by $\boldsymbol{w}_1(0, C_1)), (\boldsymbol{w}_1(0, C_1), w_2(0, C_1, C_2), w_3(0, C_2)), w_3(0, C_2)$ as $S_1, S_{123}, S_3$ respectively. The lemma is equivalent to prove that $\boldsymbol{w}_1(t, C_1), w_2(t, C_1, C_2), w_3(t, C_2)$ are $S_1, S_{123}, S_3$-measurable, respectively.

In order to prove the measurability result, we define a reduced dynamics as follows:

$$w_3^{RD}(t, c_2) = w_3^{RD}(0, c_2) - \gamma \int_0^t (w_3^{RD}(s, c_2) - w_3^{RD}(0, c_2))\, ds - \int_0^t \int_0^s \mathbb{E}\left[\Delta_3^W(\boldsymbol{z}, C_2; \boldsymbol{W}(u)) | S_3\right]\, du\, ds,$$

$$w_2^{RD}(t, c_1, c_2) = w_2^{RD}(0, c_1, c_2) - \gamma \int_0^t (w_2^{RD}(s, c_1, c_2) - w_2^{RD}(0, c_1, c_2))\, ds$$
$$- \int_0^t \int_0^s \mathbb{E}\left[\Delta_2^W(\boldsymbol{z}, C_1, C_2; \boldsymbol{W}(u)) | S_{123}\right]\, du\, ds,$$

$$\boldsymbol{w}_1^{RD}(t, c_1) = \boldsymbol{w}_1^{RD}(0, c_1) - \gamma \int_0^t (\boldsymbol{w}_1^{RD}(s, c_1) - \boldsymbol{w}_1^{RD}(0, c_1))\, ds - \int_0^t \int_0^s \mathbb{E}\left[\Delta_1^W(\boldsymbol{z}, C_1; \boldsymbol{W}(u)) | S_1\right]\, du\, ds.$$

Note the reduced dynamics $\boldsymbol{w}_1^{RD}, w_2^{RD}, w_3^{RD}$ is clearly $S_1, S_{123}, S_3$-measurable. Furthermore, the reduced dynamics is not self-contained, in the sense that the gradient terms $\mathbb{E}\left[\Delta_3^W(\boldsymbol{z}, C_2; \boldsymbol{W}(t)) | S_3\right]$, $\mathbb{E}\left[\Delta_2^W(\boldsymbol{z}, C_1, C_2; \boldsymbol{W}(t)) | S_{123}\right]$ and $\mathbb{E}\left[\Delta_1^W(\boldsymbol{z}, C_1; \boldsymbol{W}(t)) | S_1\right]$ are induced by the mean-field ODE $\boldsymbol{W}(t)$.

In order to state the next result, we define the following metric:

$$\mathcal{D}_T(\boldsymbol{W}, \boldsymbol{W}') = \max\Big\{ \sup_{t\in[0,T]} \operatorname{ess\,sup}_{c_1} \|\boldsymbol{w}_1(t, c_1) - \boldsymbol{w}_1'(t, c_1)\|_2,$$
$$\sup_{t\in[0,T]} \operatorname{ess\,sup}_{c_1, c_2} |w_2(t, c_1, c_2) - w_2'(t, c_1, c_2)|,$$
$$\sup_{t\in[0,T]} \operatorname{ess\,sup}_{c_2} |w_3(t, c_2) - w_3'(t, c_2)|\Big\}.$$

Next, we aim to show that the reduced dynamics is equivalent to the mean-field ODE, i.e., for any $T > 0$,

$$\mathcal{D}_T(\boldsymbol{W}, \boldsymbol{W}^{RD}) = 0.$$

The key step is to prove that

$$\operatorname{ess\,sup} \sup_{t\in[0,T]} |\mathbb{E}\left[\Delta_3^W(\boldsymbol{z}, C_2; \boldsymbol{W}(t)) | S_3\right] - \mathbb{E}_{\boldsymbol{z}}\Delta_3^W(\boldsymbol{z}, C_2; \boldsymbol{W}(t))| \le K(\gamma, T)\mathcal{D}_T(\boldsymbol{W}, \boldsymbol{W}^{RD}), \quad (106)$$

$$\operatorname{ess\,sup} \sup_{t\in[0,T]} |\mathbb{E}\left[\Delta_2^W(\boldsymbol{z}, C_1, C_2; \boldsymbol{W}(t)) | S_{123}\right] - \mathbb{E}_{\boldsymbol{z}}\Delta_2^W(\boldsymbol{z}, C_1, C_2; \boldsymbol{W}(t))| \le K(\gamma, T)\mathcal{D}_T(\boldsymbol{W}, \boldsymbol{W}^{RD}), \quad (107)$$

$$\operatorname{ess\,sup} \sup_{t\in[0,T]} \|\mathbb{E}\left[\Delta_1^W(\boldsymbol{z}, C_1; \boldsymbol{W}(t)) | S_1\right] - \mathbb{E}_{\boldsymbol{z}}\Delta_1^W(\boldsymbol{z}, C_1; \boldsymbol{W}(t))\|_2 \le K(\gamma, T)\mathcal{D}_T(\boldsymbol{W}, \boldsymbol{W}^{RD}), \quad (108)$$

where $K(\gamma, T)$ is a universal constant depending only on $T, \gamma$. Here, $|\mathbb{E}\left[\Delta_3^W(\boldsymbol{z}, C_2; \boldsymbol{W}(t)) | S_3\right] - \mathbb{E}_{\boldsymbol{z}}\Delta_3^W(\boldsymbol{z}, C_2; \boldsymbol{W}(t))|$ is a random variable, and the ess sup in (106) is taken with respect to it. The same remark applies to the ess sup in (107) and in (108), which are intended to be taken with respect to the corresponding random variables.

We now prove that (106) holds. Note that

$$|\mathbb{E}\left[\Delta_3^W(\boldsymbol{z}, C_2; \boldsymbol{W}(t)) | S_3\right] - \mathbb{E}_{\boldsymbol{z}}\Delta_3^W(\boldsymbol{z}, C_2; \boldsymbol{W}(t))| \le |\mathbb{E}\left[\Delta_3^W(\boldsymbol{z}, C_2; \boldsymbol{W}(t)) | S_3\right] - \mathbb{E}\left[\Delta_3^W(\boldsymbol{z}, C_2; \boldsymbol{W}^{RD}(t)) | S_3\right]|$$
$$+ |\mathbb{E}\left[\Delta_3^W(\boldsymbol{z}, C_2; \boldsymbol{W}^{RD}(t)) | S_3\right] - \mathbb{E}_{\boldsymbol{z}}\Delta_3^W(\boldsymbol{z}, C_2; \boldsymbol{W}^{RD}(t))|$$
$$+ |\mathbb{E}_{\boldsymbol{z}}\Delta_3^W(\boldsymbol{z}, C_2; \boldsymbol{W}^{RD}(t)) - \mathbb{E}_{\boldsymbol{z}}\Delta_3^W(\boldsymbol{z}, C_2; \boldsymbol{W}(t))|. \quad (109)$$

Using the Lipschitz continuous property of $\Delta_3^W$, we have that:

$$|\mathbb{E}\left[\Delta_3^W(\boldsymbol{z}, C_2; \boldsymbol{W}(t)) | S_3\right] - \mathbb{E}\left[\Delta_3^W(\boldsymbol{z}, C_2; \boldsymbol{W}^{RD}(t)) | S_3\right]| \le K(\gamma, T)\mathcal{D}_T(\boldsymbol{W}, \boldsymbol{W}^{RD}),$$
$$|\Delta_3^W(\boldsymbol{z}, C_2; \boldsymbol{W}^{RD}(t)) - \Delta_3^W(\boldsymbol{z}, C_2; \boldsymbol{W}(t))| \le K(\gamma, T)\mathcal{D}_T(\boldsymbol{W}, \boldsymbol{W}^{RD}). \quad (110)$$

By following the argument in (Pham & Nguyen, 2021a, Appendix D.2) (which does not depend on the dynamics, but only on the structure of the gradient), we have that $\mathbb{E}_{\boldsymbol{z}}\Delta_3^W(\boldsymbol{z}, C_2; \boldsymbol{W}^{RD}(t))$ is $S_3$-measurable, i.e.,

$$|\mathbb{E}\left[\Delta_3^W(\boldsymbol{z}, C_2; \boldsymbol{W}^{RD}(t))|S_3\right] - \mathbb{E}_{\boldsymbol{z}}\Delta_3^W(\boldsymbol{z}, C_2; \boldsymbol{W}^{RD}(t))| = 0. \tag{111}$$

By combining (109), (110) and (111), we obtain that (106) holds. The arguments giving (107) and (108) are analogous.

From this, we can compute the difference between the reduced dynamics and the mean-field ODE as

$$\mathcal{D}_T(\boldsymbol{W}, \boldsymbol{W}^{RD}) \le \gamma \int_0^T \mathcal{D}_s(\boldsymbol{W}, \boldsymbol{W}^{RD})\,ds + K(\gamma, T)\int_0^T \int_0^s \mathcal{D}_u(\boldsymbol{W}, \boldsymbol{W}^{RD})\,dv\,ds,$$

which, after applying Corollary F.4, gives that $\mathcal{D}_T(\boldsymbol{W}, \boldsymbol{W}^{RD}) = 0$. This implies that $\boldsymbol{W} = \boldsymbol{W}^{RD}$ and, hence, $\boldsymbol{w}_1(t, C_1), w_2(t, C_1, C_2), w_3(t, C_2)$ are $S_1, S_{123}, S_3$-measurable, respectively. $\qquad\square$

Next, we show the continuity of the function $\boldsymbol{w}_1^*(\cdot, \cdot) : \mathbb{R}^{\ge 0} \times \mathbb{R}^D \to \mathbb{R}^D$ in both arguments.

**Lemma E.2.** *Under Assumptions (B1) - (B3), we have that, for all $t \in [0, T]$ and for all $\boldsymbol{u}_1, \boldsymbol{u}_1' \in \mathbb{R}^D$,*

$$\|\boldsymbol{w}_1^*(t, \boldsymbol{u}_1) - \boldsymbol{w}_1^*(t', \boldsymbol{u}_1)\|_2 \le K(\gamma, T)|t - t'|, \tag{112}$$

$$\|\boldsymbol{w}_1^*(t, \boldsymbol{u}_1) - \boldsymbol{w}_1^*(t, \boldsymbol{u}_1')\|_2 \le K(\gamma, T)\|\boldsymbol{u}_1 - \boldsymbol{u}_1'\|_2. \tag{113}$$

*Proof.* In order to prove the lemma, we first need to derive the dynamics that characterize the evolution of the functions $\boldsymbol{w}_1^*(t, \boldsymbol{u}_1), w_2^*(t, \boldsymbol{u}_1, u_2, u_3), w_3^*(t, u_3)$. This dynamics is induced by the mean-field ODE, whose form we recall below:

$$w_3(t, c_2) = w_3(0, c_2) - \gamma \int_0^t (w_3(s, c_2) - w_3(0, c_2))\,ds - \int_0^t \int_0^s \mathbb{E}_{\boldsymbol{z}}\Delta_3^W(\boldsymbol{z}, c_2; \boldsymbol{W}(v))\,dv\,ds, \tag{114}$$

$$w_2(t, c_1, c_2) = w_2(0, c_1, c_2) - \gamma \int_0^t (w_2(s, c_1, c_2) - w_2(0, c_1, c_2))\,ds - \int_0^t \int_0^s \mathbb{E}_{\boldsymbol{z}}\Delta_2^W(\boldsymbol{z}, c_1, c_2; \boldsymbol{W}(v))\,dv\,ds, \tag{115}$$

$$\boldsymbol{w}_1(t, c_1) = \boldsymbol{w}_1(0, c_1) - \gamma \int_0^t (\boldsymbol{w}_1(s, c_1) - \boldsymbol{w}_1(0, c_1))\,ds - \int_0^t \int_0^s \mathbb{E}_{\boldsymbol{z}}\Delta_1^W(\boldsymbol{z}, c_1; \boldsymbol{W}(v))\,dv\,ds. \tag{116}$$

Recall also that $w_3(t, c_2) = w_3^*(t, w_3(0, c_2))$. Thus, in order to get the dynamics of $w_3^*(t, u_3)$, we replace $w_3(0, c_2)$ by $u_3$, $w_2(0, c_1, c_2)$ by $u_2$, and $\boldsymbol{w}_1(0, c_1)$ by $\boldsymbol{u}_1$ into (114). By doing the same replacements into (115) and (116) for $w_2(t, c_1, c_2)$ and $\boldsymbol{w}_1(t, c_1)$, respectively, we obtain

$$w_3^*(t, u_3) = u_3 - \gamma \int_0^t (w_3^*(s, u_2) - u_3)\,ds - \int_0^t \int_0^s \mathbb{E}_{\boldsymbol{z}}\Delta_3^W(v, \boldsymbol{z}, u_3)\,dv\,ds,$$

$$w_2^*(t, \boldsymbol{u}_1, u_2, u_3) = u_2 - \gamma \int_0^t (w_2^*(s, \boldsymbol{u}_1, u_2, u_3) - u_2)\,ds - \int_0^t \int_0^s \mathbb{E}_{\boldsymbol{z}}\Delta_2^W(v, \boldsymbol{z}, \boldsymbol{u}_1, u_2, u_3)\,dv\,ds,$$

$$\boldsymbol{w}_1^*(t, \boldsymbol{u}_1) = \boldsymbol{u}_1 - \gamma \int_0^t (\boldsymbol{w}_1^*(s, \boldsymbol{u}_1) - \boldsymbol{u}_1)\,ds - \int_0^t \int_0^s \mathbb{E}_{\boldsymbol{z}}\Delta_1^W(v, \boldsymbol{z}, \boldsymbol{u}_1)\,dv\,ds,$$

where we have the following modified forward and backward paths:

$$H_1(t, \boldsymbol{x}, \boldsymbol{u}_1) = (\boldsymbol{w}_1^*(t, \boldsymbol{u}_1))^T \boldsymbol{x},$$
$$H_2(t, \boldsymbol{x}, u_3) = \mathbb{E}_{\boldsymbol{u}_1 \sim \rho_0^1, u_2 \sim \rho_0^2} w_2^*(t, \boldsymbol{u}_1, u_2, u_3)\sigma_1(H_1(t, \boldsymbol{x}, \boldsymbol{u}_1)),$$
$$f(\boldsymbol{x}; \boldsymbol{W}(t)) = \mathbb{E}_{u_3 \sim \rho_0^3} w_3(t, u_3)H_2(t, \boldsymbol{x}, u_3),$$

$$\Delta_3^W(t, \boldsymbol{z}, u_3) = \partial_2 R(y, f(\boldsymbol{x}; \boldsymbol{W}(t)))\sigma_2(H_2(t, \boldsymbol{x}, u_3)),$$
$$\Delta_2^W(t, \boldsymbol{z}, \boldsymbol{u}_1, u_2, u_3) = \partial_2 R(y, f(\boldsymbol{x}; \boldsymbol{W}(t)))w_3(t, u_3)\sigma_2'(H_2(t, \boldsymbol{x}, u_3))\sigma_1(H_1(t, \boldsymbol{x}, \boldsymbol{u}_1)),$$
$$\Delta_1^W(t, \boldsymbol{z}, \boldsymbol{u}_1) = \mathbb{E}_{u_2, u_3}\partial_2 R(y, f(\boldsymbol{x}; \boldsymbol{W}(t)))w_3(t, u_3)\sigma_2'(H_2(t, \boldsymbol{x}, u_3))w_2(t, \boldsymbol{u}_1, u_2, u_3)\sigma_1'(H_1(t, \boldsymbol{x}, \boldsymbol{u}_1))\boldsymbol{x}.$$

Thus, we have that

$$\|\boldsymbol{w}_1^*(t, \boldsymbol{u}_1) - \boldsymbol{w}_1^*(t, \boldsymbol{u}_1')\|_2 \leq (1 + \gamma t)\|\boldsymbol{u}_1 - \boldsymbol{u}_1'\|_2 + \gamma \int_0^t \|\boldsymbol{w}_1^*(s, \boldsymbol{u}_1) - \boldsymbol{w}_1^*(s, \boldsymbol{u}_1')\|_2 \, ds$$
$$+ \int_0^t \int_0^s \|\mathbb{E}_{\boldsymbol{z}}\Delta_1^W(v, \boldsymbol{z}, \boldsymbol{u}_1) - \mathbb{E}_{\boldsymbol{z}}\Delta_1^W(v, \boldsymbol{z}, \boldsymbol{u}_1')\|_2 \, dv \, ds. \tag{117}$$

An application of Lemma A.2 gives that

$$\|\mathbb{E}_{\boldsymbol{z}}\Delta_1^W(v, \boldsymbol{z}, \boldsymbol{u}_1) - \mathbb{E}_{\boldsymbol{z}}\Delta_1^W(v, \boldsymbol{z}, \boldsymbol{u}_1')\|_2 \leq K_1(\gamma, T)(|w_2^*(v, \boldsymbol{u}_1, u_2, u_3) - w_2^*(v, \boldsymbol{u}_1', u_2, u_3)| + \|\boldsymbol{w}_1^*(v, \boldsymbol{u}_1) - \boldsymbol{w}_1^*(v, \boldsymbol{u}_1')\|_2). \tag{118}$$

Similarly for $w_2^*$, we have that

$$|w_2^*(t, \boldsymbol{u}_1, u_2, u_3) - w_2^*(t, \boldsymbol{u}_1', u_2, u_3)| \leq \gamma \int_0^t |w_2^*(s, \boldsymbol{u}_1, u_2, u_3) - w_2^*(s, \boldsymbol{u}_1', u_2, u_3)| \, ds$$
$$+ \int_0^t \int_0^s \|\mathbb{E}_{\boldsymbol{z}}\Delta_2^W(v, \boldsymbol{z}, \boldsymbol{u}_1, u_2, u_3) - \mathbb{E}_{\boldsymbol{z}}\Delta_2^W(v, \boldsymbol{z}, \boldsymbol{u}_1', u_2, u_3)\|_2 \, dv \, ds, \tag{119}$$

and another application of Lemma A.2 gives that

$$\|\mathbb{E}_{\boldsymbol{z}}\Delta_2^W(v, \boldsymbol{z}, \boldsymbol{u}_1, u_2, u_3) - \mathbb{E}_{\boldsymbol{z}}\Delta_2^W(v, \boldsymbol{z}, \boldsymbol{u}_1', u_2, u_3)\|_2 \leq K_2(\gamma, T)(|w_2^*(v, \boldsymbol{u}_1, u_2, u_3) - w_2^*(v, \boldsymbol{u}_1', u_2, u_3)| + \|\boldsymbol{w}_1^*(v, \boldsymbol{u}_1) - \boldsymbol{w}_1^*(v, \boldsymbol{u}_1')\|_2). \tag{120}$$

By combining (117), (118), (119) and (120), we obtain

$$|w_2^*(t, \boldsymbol{u}_1, u_2, u_3) - w_2^*(t, \boldsymbol{u}_1', u_2, u_3)| + \|\boldsymbol{w}_1^*(t, \boldsymbol{u}_1) - \boldsymbol{w}_1^*(t, \boldsymbol{u}_1')\|_2$$
$$\leq (1 + \gamma t)\|\boldsymbol{u}_1 - \boldsymbol{u}_1'\|_2 + \gamma \int_0^t (|w_2^*(s, \boldsymbol{u}_1, u_2, u_3) - w_2^*(s, \boldsymbol{u}_1', u_2, u_3)| + \|\boldsymbol{w}_1^*(s, \boldsymbol{u}_1) - \boldsymbol{w}_1^*(s, \boldsymbol{u}_1')\|_2) \, ds$$
$$+ K_3(\gamma, T) \int_0^t \int_0^s (|w_2^*(v, \boldsymbol{u}_1, u_2, u_3) - w_2^*(v, \boldsymbol{u}_1', u_2, u_3)| + \|\boldsymbol{w}_1^*(v, \boldsymbol{u}_1) - \boldsymbol{w}_1^*(v, \boldsymbol{u}_1')\|_2) \, dv \, ds.$$

Thus, by Corollary F.4, we have that:

$$|w_2^*(t, \boldsymbol{u}_1, u_2, u_3) - w_2^*(t, \boldsymbol{u}_1', u_2, u_3)| + \|\boldsymbol{w}_1^*(t, \boldsymbol{u}_1) - \boldsymbol{w}_1^*(t, \boldsymbol{u}_1')\|_2 \leq K_4(\gamma, T)\|\boldsymbol{u}_1 - \boldsymbol{u}_1'\|_2,$$

which implies that $\|\boldsymbol{w}_1^*(t, \boldsymbol{u}_1) - \boldsymbol{w}_1^*(t, \boldsymbol{u}_1')\|_2 \leq K_4(\gamma, T)|\boldsymbol{u}_1 - \boldsymbol{u}_1'|$, and concludes the proof of (113). The Lipschitz continuity (112) of $\boldsymbol{w}_1^*(t, \boldsymbol{u}_1)$ is already proved in Lemma D.2. $\square$

At this point, we show that, if $w_1(0, c_1) : \Omega \to \mathbb{R}^D$ has full support, then $w_1^*(t, \boldsymbol{u}_1)$ has full support.

**Lemma E.3.** *Under Assumptions (B1)-(B3) and Assumption 7.1, we have that $\boldsymbol{w}_1^*(t, \boldsymbol{u}_1)$ has full support for any $t < \infty$.*

*Proof.* By the continuity argument in Lemma E.2, we have that

$$\|\boldsymbol{w}_1^*(t, \boldsymbol{u}_1) - \boldsymbol{u}_1\|_2 = \|\boldsymbol{w}_1^*(t, \boldsymbol{u}_1) - \boldsymbol{w}_1^*(0, \boldsymbol{u}_1)\|_2 \leq K(\gamma, T)t, \tag{121}$$
$$\|\boldsymbol{w}_1^*(t, \boldsymbol{u}_1) - \boldsymbol{w}_1^*(t, \boldsymbol{u}_1')\|_2 \leq K(\gamma, T)\|\boldsymbol{u}_1 - \boldsymbol{u}_1'\|_2. \tag{122}$$

We want to show that, for any $\boldsymbol{x} \in \mathbb{R}^D$, there exist a $\boldsymbol{v}$ such that $\boldsymbol{w}_1^*(t, \boldsymbol{v}) = \boldsymbol{x}$. For any $\boldsymbol{x} \in \mathbb{R}^D$, define a map $g_{\boldsymbol{x}}(t, \boldsymbol{v}) = \boldsymbol{x} - (\boldsymbol{w}_1^*(t, \boldsymbol{v}) - \boldsymbol{v})$. It is easy to see that if $\boldsymbol{v}$ a fixed point of $g_{\boldsymbol{x}}(t, \cdot)$, then $\boldsymbol{w}_1^*(t, \boldsymbol{v}) = \boldsymbol{x}$ as

$$g_{\boldsymbol{x}}(t, \boldsymbol{v}) = \boldsymbol{v} \iff \boldsymbol{x} - (\boldsymbol{w}_1^*(t, \boldsymbol{v}) - \boldsymbol{v}) = \boldsymbol{v} \iff \boldsymbol{w}_1^*(t, \boldsymbol{v}) = \boldsymbol{x}.$$

By (121), we have that $g_{\boldsymbol{x}}(t, \cdot) : \mathbb{R}^D \to \mathcal{B}(\boldsymbol{x}, K(\gamma, T)t)$, where $\mathcal{B}(\boldsymbol{x}, K(\gamma, T)t)$ is the closed ball centered at $\boldsymbol{x}$ with radius $K(\gamma, T)t$. Now, if we restrict $g_{\boldsymbol{x}}(t, \boldsymbol{v})$ on $\mathcal{B}(\boldsymbol{x}, K(\gamma, T)t)$, we have that it is a map from $\mathcal{B}(\boldsymbol{x}, K(\gamma, T)t)$, which is a compact set, to itself. Furthermore, $g_{\boldsymbol{x}}(t, \boldsymbol{v})$ is continuous in $\boldsymbol{v}$, since $\boldsymbol{w}_1^*(t, \boldsymbol{v})$ is continuous in $\boldsymbol{v}$ by (122). Thus, by the Brouwer fixed point theorem, we have that there exist a fixed point $\boldsymbol{v} \in \mathcal{B}(\boldsymbol{x}, K(\gamma, T)t)$, which finishes the argument. $\square$

Finally, we are ready to prove the main theorem. Our proof follows similar steps as that of (Pham & Nguyen, 2021a, Proof of Theorem 8).

*Proof of Theorem 7.2.* By Assumption 7.1, we have that

$$\lim_{t \to \infty} \operatorname*{ess\,sup}_{C_1} \mathbb{E}_{C_2}[|\mathbb{E}_{\boldsymbol{z}}\Delta_2^W(\boldsymbol{z}, C_1, C_2; \boldsymbol{W}(t))|] = 0.$$

By the definition of $\Delta_2^W(t, \boldsymbol{z}, C_1, C_2)$, we have

$$\lim_{t \to \infty} \operatorname*{ess\,sup}_{C_1} \mathbb{E}_{C_2}[|\mathbb{E}_{\boldsymbol{z}}\Delta_2^H(\boldsymbol{z}, C_2; \boldsymbol{W}(t))\sigma_1(\boldsymbol{w}_1(t, C_1)^T\boldsymbol{x})|] = 0.$$

Recall from Lemma E.3 that, for all finite $t$, $\boldsymbol{w}_1(t, C_1)$ has full support. Hence, we have that, for $\boldsymbol{u}_1$ in a dense subset of $\mathbb{R}^D$,

$$\lim_{t \to \infty} \mathbb{E}_{C_2}[|\mathbb{E}_{\boldsymbol{z}}\Delta_2^H(\boldsymbol{z}, C_2; \boldsymbol{W}(t))\sigma_1(\boldsymbol{u}_1^T\boldsymbol{x})|] = 0.$$

Our aim is to conclude that, for almost all $\boldsymbol{x}$, we have that $\mathbb{E}_{\boldsymbol{z}}\left[\partial_2 R(y, f(\boldsymbol{x}; \boldsymbol{W}(\infty)))|\boldsymbol{x}\right] = 0$. By definition of the backward path, we have that

$$\mathbb{E}_{C_2}[|\mathbb{E}_{\boldsymbol{z}}\Delta_2^H(\boldsymbol{z}, C_2; \boldsymbol{W}(t))\sigma_1(\boldsymbol{u}_1^T\boldsymbol{x})| - |\mathbb{E}_{\boldsymbol{z}}\Delta_2^H(\boldsymbol{z}, C_2; \boldsymbol{W}(\infty))\sigma_1(\boldsymbol{u}_1^T\boldsymbol{x})|]$$
$$\leq \mathbb{E}_{C_2}[|(\mathbb{E}_{\boldsymbol{z}}\Delta_2^H(\boldsymbol{z}, C_2; \boldsymbol{W}(t)) - \mathbb{E}_{\boldsymbol{z}}\Delta_2^H(\boldsymbol{z}, C_2; \boldsymbol{W}(\infty)))\sigma_1(\boldsymbol{u}_1^T\boldsymbol{x})|]$$
$$\leq K\mathbb{E}_{C_2}[\mathbb{E}_{\boldsymbol{z}}\left[|\Delta_2^H(\boldsymbol{z}, C_2; \boldsymbol{W}(t)) - \Delta_2^H(\boldsymbol{z}, C_2; \boldsymbol{W}(\infty))|\right]]$$
$$\leq K\mathbb{E}_{C_1, C_2}[(1 + |w_3(\infty, C_2)|) \cdot (|w_3(\infty, C_2) - w_3(t, C_2)| + |w_3(\infty, C_2)| \cdot |w_2(\infty, C_1, C_2) - w_2(t, C_1, C_2)|$$
$$+ |w_3(\infty, C_2)| \cdot |w_2(\infty, C_1, C_2)| \cdot \|\boldsymbol{w}_1(\infty, C_1) - \boldsymbol{w}_1(t, C_1)\|_2)].$$

By Assumption 7.1, the RHS of (123) converges to 0 as $t \to \infty$. Hence, by taking the limit on both sides, we have that, for $\boldsymbol{u}_1$ in a dense subset of $\mathbb{R}^D$,

$$\mathbb{E}_{C_2}[|\mathbb{E}_{\boldsymbol{z}}\Delta_2^H(\boldsymbol{z}, C_2; \boldsymbol{W}(\infty))\sigma_1(\boldsymbol{u}_1^T\boldsymbol{x})|] = \lim_{t \to \infty} \mathbb{E}_{C_2}[|\mathbb{E}_{\boldsymbol{z}}\Delta_2^H(\boldsymbol{z}, C_2; \boldsymbol{W}(t))\sigma_1(\boldsymbol{u}_1^T\boldsymbol{x})|] = 0,$$

which implies that, for almost all $c_2$,

$$\left|\mathbb{E}_{\boldsymbol{z}}\Delta_2^H(\boldsymbol{z}, C_2; \boldsymbol{W}(\infty))\sigma_1(\boldsymbol{u}_1^T\boldsymbol{x})\right| = 0.$$

By definition of $\Delta_2^H(\boldsymbol{z}, C_2; \boldsymbol{W}(\infty))$ we have that, for almost all $c_2$,

$$\mathbb{E}_{\boldsymbol{z}}\left[\partial_2 R(y, f(\boldsymbol{x}; \boldsymbol{W}(\infty)))w_3(\infty, c_2)\sigma_2'(H_2(\boldsymbol{x}, c_2; \boldsymbol{W}(\infty)))\sigma_1(\boldsymbol{u}_1^T\boldsymbol{x})\right] = 0. \tag{123}$$

Note that Assumption (B1) gives that $\sigma_2' \neq 0$, and Assumption 7.1 that $w_3(\infty, c_2) \neq 0$ with probability $> 0$ (where the probability is intended over $c_2$). Hence, we have that, with probability $> 0$ (over $c_2$),

$$\mathbb{E}_{\boldsymbol{z}}\left[\partial_2 R(y, f(\boldsymbol{x}; \boldsymbol{W}(\infty)))\sigma_2'(H_2(\boldsymbol{x}, c_2; \boldsymbol{W}(\infty)))\sigma_1(\boldsymbol{u}_1^T\boldsymbol{x})\right] = 0. \tag{124}$$

Recall that $\sigma_1(\boldsymbol{u}_1^T\boldsymbol{x})$ is a function of $\boldsymbol{x}$, but $\partial_2 R(y, f(\boldsymbol{x}; \boldsymbol{W}(\infty)))$ depends on both $y$ and $\boldsymbol{x}$. Thus, we can re-write (124) as

$$\mathbb{E}_{\boldsymbol{x}}\left[\mathbb{E}_y\left[\partial_2 R(y, f(\boldsymbol{x}; \boldsymbol{W}(\infty)))|\boldsymbol{x}\right]\sigma_2'(H_2(\boldsymbol{x}, c_2; \boldsymbol{W}(\infty)))\sigma_1(\boldsymbol{u}_1^T\boldsymbol{x})\right] = 0. \tag{125}$$

Now, we want to use the universal approximation property of $\sigma_1$ to conclude that, for almost every $\boldsymbol{x}$,

$$\mathbb{E}_y\left[\partial_2 R(y, f(\boldsymbol{x}; \boldsymbol{W}(\infty)))|\boldsymbol{x}\right]\sigma_2'(H_2(\boldsymbol{x}, c_2; \boldsymbol{W}(\infty))) = 0. \tag{126}$$

The idea is that linear combinations of $\sigma_1(\boldsymbol{u}_1^T\boldsymbol{x})$ can approximate any function in $\mathcal{L}_2(\mathcal{D}_{\boldsymbol{x}})$. Thus, if $\mathbb{E}_y[\partial_2 R(y, f(\boldsymbol{x}; \boldsymbol{W}(\infty)))|\boldsymbol{x}]\sigma_2'(H_2(\boldsymbol{x}, c_2; \boldsymbol{W}(\infty)))$ is in $\mathcal{L}_2(\mathcal{D}_{\boldsymbol{x}})$, we have that there exist a sequence of index sets $\{I_k\}_{k\in\mathbb{N}}$, such that:

$$\lim_{k\to\infty} \mathbb{E}_{\boldsymbol{x}}\left[\left|\mathbb{E}_y[\partial_2 R(y, f(\boldsymbol{x}; \boldsymbol{W}(\infty)))|\boldsymbol{x}]\sigma_2'(H_2(\boldsymbol{x}, c_2; \boldsymbol{W}(\infty))) - \sum_{i_k\in I_k} a_{i_k}\sigma_1(\boldsymbol{u}_{i_k}^T\boldsymbol{x})\right|^2\right] = 0.$$

To simplify the notation, we define:

$$g(\boldsymbol{x}) = \mathbb{E}_y[\partial_2 R(y, f(\boldsymbol{x}; \boldsymbol{W}(\infty)))|\boldsymbol{x}]\sigma_2'(H_2(\boldsymbol{x}, c_2; \boldsymbol{W}(\infty))),$$
$$h_k(\boldsymbol{x}) = \sum_{i_k\in I_k} a_{i_k}\sigma_1(\boldsymbol{u}_{i_k}^T\boldsymbol{x}).$$

From (125) and by linearity of expectation, we have that, for all $k$,

$$\mathbb{E}_{\boldsymbol{x}}[g(\boldsymbol{x})h_k(\boldsymbol{x})] = 0.$$

Thus we have

$$\begin{aligned}
0 &= \lim_{k\to\infty} \mathbb{E}_{\boldsymbol{x}}\left[|g(\boldsymbol{x}) - h_k(\boldsymbol{x})|^2\right] \\
&= \lim_{k\to\infty} \mathbb{E}_{\boldsymbol{x}}\left[|g(\boldsymbol{x})|^2 + |h_k(\boldsymbol{x})|^2 - 2g(\boldsymbol{x})h_k(\boldsymbol{x})\right] \\
&= \lim_{k\to\infty} \mathbb{E}_{\boldsymbol{x}}\left[|g(\boldsymbol{x})|^2 + |h_k(\boldsymbol{x})|^2\right],
\end{aligned}$$

which implies that

$$\mathbb{E}_{\boldsymbol{x}}\left[|g(\boldsymbol{x})|^2\right] = 0.$$

Hence, we have that

$$\mathbb{E}_{\boldsymbol{x}}\left[\left|\mathbb{E}_y[\partial_2 R(y, f(\boldsymbol{x}; \boldsymbol{W}(\infty)))|\boldsymbol{x}]\sigma_2'(H_2(\boldsymbol{x}, c_2; \boldsymbol{W}(\infty)))\right|^2\right] = 0,$$

which implies that (126) holds. Furthermore, to see that $\mathbb{E}_y[\partial_2 R(y, f(\boldsymbol{x}; \boldsymbol{W}(\infty)))|\boldsymbol{x}]\sigma_2'(H_2(\boldsymbol{x}, c_2; \boldsymbol{W}(\infty)))$ is indeed in $\mathcal{L}_2(\mathcal{D}_{\boldsymbol{x}})$, it suffices to note that, by Assumption 3.3,

$$\mathbb{E}_y[\partial_2 R(y, f(\boldsymbol{x}; \boldsymbol{W}(\infty)))|\boldsymbol{x}]\sigma_2'(H_2(\boldsymbol{x}, c_2; \boldsymbol{W}(\infty))) \le K^2.$$

By Assumption 3.3, we also have that $\sigma_2'(x) \ne 0$ for all $x$. Hence, (126) implies that, for almost every $\boldsymbol{x}$,

$$\mathbb{E}_y[\partial_2 R(y, f(\boldsymbol{x}; \boldsymbol{W}(\infty)))|\boldsymbol{x}] = 0. \tag{127}$$

Since the loss is convex in $f(\boldsymbol{x}; \boldsymbol{W}(\infty))$, we have

$$\mathbb{E}_{\boldsymbol{z}}[R(y, \widetilde{f}(\boldsymbol{x})) - R(y, f(\boldsymbol{x}; \boldsymbol{W}(\infty)))] \ge \mathbb{E}_{\boldsymbol{x}}[\mathbb{E}_y[\partial_2 R(y, f(\boldsymbol{x}; \boldsymbol{W}(\infty)))|\boldsymbol{x}](\widetilde{f}(\boldsymbol{x}) - f(\boldsymbol{x}; \boldsymbol{W}(\infty)))] = 0,$$

where the last passage follows from (127). Thus, we conclude that

$$\mathbb{E}_{\boldsymbol{z}}R(y, f(\boldsymbol{x}; \boldsymbol{W}(\infty))) = \inf_{\widetilde{f}} \mathbb{E}_{\boldsymbol{z}}[R(y, \widetilde{f}(\boldsymbol{x}))]. \tag{128}$$

Finally, we want to show that

$$\lim_{t\to\infty} \mathbb{E}_{\boldsymbol{z}}R(y, f(\boldsymbol{x}; \boldsymbol{W}(t))) = \mathbb{E}_{\boldsymbol{z}}R(y, f(\boldsymbol{x}; \boldsymbol{W}(\infty))). \tag{129}$$

To see this, we write

$$
\begin{aligned}
&|\mathbb{E}_{\boldsymbol{z}} R(y, f(\boldsymbol{x}; \boldsymbol{W}(t))) - \mathbb{E}_{\boldsymbol{z}} R(y, f(\boldsymbol{x}; \boldsymbol{W}(\infty)))| \\
&\leq K \mathbb{E}_{\boldsymbol{z}} |f(\boldsymbol{x}; \boldsymbol{W}(t)) - f(\boldsymbol{x}; \boldsymbol{W}(\infty))| \\
&\leq K \mathbb{E}_{C_1, C_2} \big[ |w_3(\infty, C_2) - w_3(t, C_2)| + |w_3(\infty, C_2)| \cdot |w_2(\infty, C_1, C_2) - w_2(t, C_1, C_2)| \\
&\quad + |w_3(\infty, C_2)| \cdot |w_2(\infty, C_1, C_2)| \cdot \|\boldsymbol{w}_1(\infty, C_1) - \boldsymbol{w}_1(t, C_1)\|_2 \big],
\end{aligned}
$$

and use again Assumption 7.1. By combining (128) and (129), we obtain the desired result.

$\square$

# F   Technical lemmas

**Lemma F.1** (Corollary of McDiarmid inequality). *(Mei et al., 2019, Lemma 30)*

*Let $\{X_i\}_{i \in [n]} \in \mathbb{R}^d$ be a sequence of i.i.d random variables, with $\|X_i\|_2 \leq K$ and $\mathbb{E}[X_i] = 0$, then we have:*

$$
\Pr \left( \left\| \frac{1}{n} \sum_{i=1}^n X_i \right\|_2 \geq K(\sqrt{1/n} + z) \right) \leq \exp\left(-nz^2\right).
$$

**Lemma F.2** (Azuma-Hoeffding bound). *(Mei et al., 2019, Lemma 31) Let $(X_k)_{k \geq 0}$ be a martingale taking values in $\mathbb{R}^D$ with respect to the filtration $(\mathcal{F}_k)$, with $X_0 = 0$. Assume that the martingale difference at time $k$ is $L_k$-subgaussian, which means the following holds almost surely for all $\lambda \in \mathbb{R}^D$:*

$$
\mathbb{E}\left[\exp\{\langle \lambda, X_k - X_{k-1}\rangle\}|\mathcal{F}_{k-1}\right] \leq \exp\left\{\frac{L_k^2 \|\lambda\|^2}{2}\right\}.
$$

*Then, we have*

$$
\Pr\left[\max_{k \in [n]} \|X_k\|_2 \geq 2\sqrt{\sum_{k=1}^n L_k^2}\left(\sqrt{D} + \delta\right)\right] \leq \exp\{-\delta^2\}.
$$

*Note that, if $L_k \leq L$ for all $k$, then*

$$
\Pr\left[\max_{k \in [n]} \|X_k\|_2 \geq 2\sqrt{nL^2}\left(\sqrt{D} + \delta\right)\right] \leq \exp\{-\delta^2\}.
$$

**Lemma F.3** (Pachpatte's inequality). *(Ames & Pachpatte, 1997, Chapter 1, Theorem 1.7.1)*

*Let $u$, $f$ and $g$ be non-negative continuous functions defined on $[0, T]$, for which the inequality*

$$
u(t) \leq u_0 + \int_0^t f(s) u(s) \, ds + \int_0^t f(s) \left( \int_0^s g(r) u(r) \, dr \right) ds
$$

*holds, where $u_0$ is a non-negative constant. Then we have:*

$$
u(t) \leq u_0 \left[ 1 + \int_0^t f(s) \exp\left( \int_0^s (g(r) + f(r)) \, dr \right) ds \right].
$$

**Corollary F.4** (Pachpatte's inequality for constants). *Let $u$ be a non-negative continuous function defined on $[0, T]$, and $\gamma, K$ be positive real numbers. Assume the following inequality holds:*

$$
u(t) \leq u_0 + \gamma \int_0^t u(s) \, ds + K \int_0^t \int_0^s u(r) \, dr \, ds.
$$

*Then, we have*

$$
u(t) \leq u_0 \left( 1 + \frac{\gamma^2}{\gamma^2 + K} \exp\left( \frac{\gamma^2 + K}{\gamma} t \right) \right) \leq u_0 \left( 1 + \exp\left( \frac{\gamma^2 + K}{\gamma} t \right) \right).
$$

