# OpenReview forum: "Mean-field analysis for heavy ball methods: Dropout-stability, connectivity, and global convergence"
_TMLR — Accepted by TMLR_

### Review · Reviewer_Vu68 · 2022-11-17

**Summary Of Contributions:**

The authors analyse the stochastic heavy ball (SHB) method on 2-layer and 3-layer fully connected neural networks with non-linear activation. They consider the mean-field limit where the number of neurons grow to infinity. They show existence + uniqueness of the solutions of the considered mean-field ODEs and provide non-asymptotic bounds on the error between the discrete finite-width SHB dynamics and the continuous mean-field limit.  They then leverage this result to show dropout stability and connectivity of the iterates after k steps of  SHB.



**Audience:**

Yes

**Claims And Evidence:**

Yes

**Requested Changes:**

As explained in the previous section, my major concerns are on the lack of rigor in definitions and notations.

Also I am skeptical about the conclusions which can be drawn from the bounds from theorem 5.1 and 5.2 given the fact the dependency on $T$ is prohibitive to drawing any meaningful conclusion on finite NNs.

**Strengths And Weaknesses:**

Let me first start by pointing out that I am unfortunately not a specialist of the mean-field regime setting as I have not personally worked on it. It is highly probable that some of my following comments are irrelevant for an "mean-field expert". However I believe that the paper should still be easily readable for someone (such as myself) which is familiar with the general NNs + momentum setting, but less with the mean-field setting.

Strengths:
- trying to understand the heavy ball method in the context of training NNs is very relevant to the community
- the proposed results are non-trivial and seem to be technically involved
- Theorem 5.1 and 5.2: obtaining non asymptotic results on finite-width networks will interest the community

Weaknesses / questions:
Overall when reading the paper I had a lot of difficulty to understand the setting, the notations and some equations as many objects are not properly introduced and there are many missing assumptions. To cite a few:
- Beginning of section 3.2: the loss $R(y, \hat{y})$ is never properly defined, and what are the assumptions on it ? is it convex as in (Krichene et al. 2020) ? is it differentiable ?
- just before equation(4): parameter $\gamma$ is never introduced, what is it ?
- equation (5): $\hat{\nabla}R(W(t))$ does not seem to be defined, why is there a "hat" here ?
- section 4: $\mathbb{E}_{\theta \sim \hat{\rho}_\theta}$ is an odd notation. The abuse of notation over $\theta$ (vector vs random variable) is a bit misleading and should be explained or changed.
- equation (11): all the definitions would be easier to understand if it was clearly stated what are the domain definitions of the introduced quantities.
- before equation (13): what do you mean by "(stochastic) ODE" ?
- equation (13): what does $W(t)$ correspond to once in the mean field regime ?
- equation (16): what is $z$ ? should it be $z_k$ ?

I have no doubt that nothing is wrong in all this, but it would be much easier for the reader to follow if all the objects are properly and rigorously defined. Also, the sloppy notations gives the impression that the tools are not properly handled.

Concerning the results:
- eq (12): are there any results showing the consistency of this ODE ( Theorem 1 in Krichene et al (2020) but in the noiseless case) ? if not then what is the validity of considering this regime ?
-  Theorem 4.1 for the case of 2 layers: in Krichene et al (2020), to show the existence and uniqueness of solutions of their dynamics, they need an additional assumption on the initial distribution ((A5) in their paper), which you don't seem to need. Can you comment on this ?
- Section 6: given the fact that $K(\gamma, T)$ is exceedingly big, it is hard to deduce / believe from (25) a "provable justification to two remarkable properties exhibited by solutions obtained via gradient-based methods, namely, dropout-stability and connectivity" as stated.
- some numerical simulations explaining / highlighting your results would have been much appreciated

Minor comments:
- in the intro: "This perspective has facilitated the study of architectures with multiple layers (Araújo et al., 2019; Lu et al., 2020; Nguyen & Pham, 2020; Fang et al., 2021), and it has given a rigorous justification to a number of properties displayed by SGD solutions, including convergence towards a global optimum": I believe saying "a rigorous justification" is in my optinion an overclaim: there are still many things which are not at all understood concerning the global convergence of discrete GD and even more discrete of SGD !
- top of page 4: should be N instead of D in $W \in \mathbb{R}^{n D + D}$
- it is not clear to me why the mean-field ODE for 3 layers (equation 3) is written in a complete different way than in the case of 2-layers (eq 12)

---

> ### Author Response · Authors · 2022-11-25
> **Response to Reviewer Vu68 -- Part 1**
>
> We thank the reviewer for the careful review and helpful suggestions. We reply below to all the comments. We will post an updated version of the paper after receiving all the reviews, as suggested in the TMLR guidelines.
>
> First of all, we realize that the paper is rather technical and quite dense in the notation used. Thus, in order to improve the readability, we will repeat some of the explanations, definitions and assumptions. We now address all the points raised by the reviewer concerning the definitions and notations. In case there are other notations or assumptions which remain unclear, we would be happy to address those as well.
>
> **(1) “Beginning of section 3.2: the loss $R(y,\hat{y})$ is never properly defined, and what are the assumptions on it? is it convex as in (Krichene et al., 2020)? is it differentiable?”**
>
> The assumptions on the loss function $R(y, \hat{y})$ are contained in (A1) (see Assumption 3.1) and (B1) (see Assumption 3.3) for networks with two and three layers, respectively. In particular, we require $R(\cdot, \cdot)$ to be differentiable with respect to its second argument and to have a bounded derivative. This is satisfied e.g. by the logistic or Huber loss. While the assumptions are not satisfied for the square loss, we expect our results to still hold provided that the assumptions are modified as in Mei et al. (2019) (for two-layer networks) and in Nguyen & Pham (2020) (for three layer networks). We also remark that, for the convergence to the mean field limit (Theorem 5.1 and 5.2), the convexity of the loss is not required. In contrast, to obtain the global convergence result of Theorem 7.2, we need to additionally assume that $R(\cdot, \cdot)$ is convex in the second argument, as mentioned in the statement of the theorem. We will review all these assumptions when the loss is first introduced, namely, at the beginning of Section 3.2.
>
> **(2) “just before equation(4): parameter $\gamma$ is never introduced, what is it?”**
>
> The parameter $\gamma$ is a constant, in the sense that it does not depend on the network width or on the step size of gradient descent. In particular, $\gamma$ allows us to pass from the formulation of the SHB dynamics in (3) (where the effect of the momentum is captured by $\beta$) to the formulation in (5) (which, instead, contains $\gamma$). As mentioned in the line before equation (4), the mapping is done by taking $ \beta = 1- \gamma \varepsilon$. We remark that the formulation with the parameter $\gamma$ is the same as in Krichene et al. (2020).
>
> **(3) “equation (5): $\hat{\nabla} R(W(t))$ does not seem to be defined, why is there a "hat" here?”**
>
> The notation $\hat{\nabla}R$ denotes the scaled gradient and it is defined in the line after equation (3). To improve readability and avoid confusion, we will recall this definition after equation (5).
>
> **(4) “section 4: $\mathbb E_{\theta \sim \rho_{\theta}}$ is an odd notation. The abuse of notation over $\theta$ (vector vs random variable) is a bit misleading and should be explained or changed.”**
>
> While we agree that the notation is somewhat overloaded, in consideration of the amount of notation already introduced, we would prefer to avoid defining yet another symbol at this stage. For this reason, our suggestion is to keep the notation and clarify right after the equation that $\boldsymbol{W} =(\boldsymbol{ \theta}(j))_{j \in [n]}$ and $\boldsymbol{ \theta}(j) = (\boldsymbol{w}_1(j), w_2(j))$. We hope that this addresses the comment of the reviewer, and we remain at disposal for additional clarifications.
>
> **(5) “equation (11): all the definitions would be easier to understand if it was clearly stated what are the domain definitions of the introduced quantities.”**
>
> We agree with the reviewer and we will add the following domain and co-domain definitions right after equation (11): $f( \cdot; \rho) : \mathbb{R}^{D} \rightarrow \mathbb{R}$; $R(\cdot ; \rho) : \mathbb{R}^{D+1} \rightarrow \mathbb{R}$, $R( \cdot) : \mathcal{P}_2(\mathbb{R}^{D+1}) \rightarrow \mathbb{R}$, where $\mathcal{P}_2(\mathbb{R}^{D+1}) $ denotes the space of probability measures on $\mathbb{R}^{D+1}$ with finite second moment; $\hat{\Psi}(\cdot,\cdot;\rho):  \mathbb{R}^{D+1} \times \mathbb{R}^{D+1} \rightarrow \mathbb{R}$; $\Psi(\cdot;\rho):  \mathbb{R}^{D+1} \rightarrow \mathbb{R}$.
>
> **(6) “before equation (13): what do you mean by "(stochastic) ODE" ?”**
>
> The mean field dynamics is defined as the evolution of the functions $\{\boldsymbol{w}_1(t,c_1), w_2(t,c_1,c_2), w_3(t,c_1,c_2)\}$. Thus, if we consider the dynamics as the evolution of each function value, then it is an ODE; if we replace $c_1, c_2$ with the random variables $C_1,C_2$, then the dynamics is a stochastic process and, hence, we have referred to it as a "(stochastic) ODE". Since we realize that this wording may create confusion, in the revision we will simply refer to it as an ODE, avoiding the clarification that it is stochastic.

---

> ### Author Response · Authors · 2022-11-25
> **Response to Reviewer Vu68 -- Part 2**
>
> **(7) “equation (13): what does $W(t)$ correspond to once in the mean field regime?”**
>
> We will clarify right after equation (13) that $\boldsymbol{W}(t)$ refers to the collection of weights $\{\boldsymbol{w}_1(t), w_2(t), w_3(t)\}$.
>
> **(8) “equation (16): what is $z$ ? should it be $z_k$?”**
>
> We thank the reviewer for pointing out this typo. $\boldsymbol{z}$ should indeed be $\boldsymbol{z}_k$. We will also recall that $\boldsymbol{z}_k=(\boldsymbol{x}_k, y_k)$, as defined in the paragraph before equation (3).
>
> --------------------------------------------------------------------------
> --------------------------------------------------------------------------
> --------------------------------------------------------------------------
>
> Let us now address the comments of the reviewer concerning the results.
>
> **(1) “eq (12): are there any results showing the consistency of this ODE (Theorem 1 in Krichene et al. (2020) but in the noiseless case)? if not then what is the validity of considering this regime?”**
>
> To the best of our knowledge, we are not aware of any existing results on the consistency of the noiseless dynamics for heavy ball methods, and our Theorem 5.1 provides the first such guarantee. We remark that the injection of noise in the ‘noisy’ dynamics often simplifies the analysis and it allows to prove stronger results (e.g., existence and uniqueness of the limit, see the next point in the response). However, the noiseless dynamics is particularly interesting, since Brownian noise is typically *not* added in practice while training.
>
> **(2) “Theorem 4.1 for the case of 2 layers: in Krichene et al. (2020), to show the existence and uniqueness of solutions of their dynamics, they need an additional assumption on the initial distribution ((A5) in their paper), which you don't seem to need. Can you comment on this?”**
>
> We thank the reviewer for the question, this is indeed a very good point. The short answer is that in Krichene et al. (2020), the authors prove the existence and uniqueness of the solution for any $t \in [0, \infty]$, which includes the existence and uniqueness of the limiting point (for $t=\infty$). In our case, we only prove the existence and uniqueness of the solution for any finite $t$, and we do not have guarantees on the limiting point. This difference is the source of the discrepancy in the assumptions.
>
> We now discuss in detail each assumption in (A5) of (Krichene et al., 2020), and how it relates to our work. We will incorporate this discussion after Theorem 4.1.
>
> (i) “$\mu_0$ is absolutely continuous”. This guarantees that, for any finite $t$,  $\mu_t$ is also absolutely continuous and, hence, the limiting point has a Gibbs form. In our proof, this Gibbs characterization is not needed and, hence, the assumption is not required.
>
> (ii) “$\rho_0$ satisfies $\langle g(\theta) + |r|^2/2 , \rho_0 \rangle < \infty$”. This guarantees that the second moment of $\rho_t$ is uniformly bounded (i.e., the upper bound is independent of $t$). The uniform bound then implies the weak compactness of the sequence $\rho_t$, and it guarantees a weak limit. In our proof, we need the second moment of $\rho_t$  to be bounded (but not necessarily to be uniformly bounded), so that the Wasserstein-2 distance is well-defined. In fact, in our assumptions 3.1 and 3.3, we assume that $\rho_0^{\theta}$ is $K$-subgaussian and $\rho_0^{r} = \delta_{0}$, which guarantees that the second moment is finite.
>
> (iii) “$\langle \log^+ \rho_0 , \rho_0 \rangle < \infty$” is also necessary for the weak compactness of $\{\rho_t\}$. In contrast, in our paper, we only show the existence of the solution for any finite time $t$, and we do not give guarantees on the limit. Hence, this assumption is not needed in our work.
>
>  (iv) “$\int |\nabla F_0'(\rho_0)(\theta)|^2 d\theta<\infty$”. This assumption is used in the proof of Theorem 5 to show that the limiting point is the minimizer of the free energy and it has a Gibbs form. Since we are not trying to characterize the limiting point, this assumption is not needed in our work.
>
> In conclusion, the fundamental difference between the noiseless case (considered here) and the noisy case (considered by Krichene et al. (2020)) lies in their limiting behavior. In fact, in the noisy case, the free energy has a second moment regularizer and an entropy regularizer, which implies that the mean-field dynamics is guaranteed to converge to the unique global minimizer of the free energy. In contrast, in the noiseless case, we do not add a regularizer and the convergence is not guaranteed. Indeed, note that the result of Theorem 7.2 has the flavor ``if the dynamics converges, then it does so to the global optimum’’. This difference is also reflected in the (different) assumptions required for the analysis to go through.

---

> ### Author Response · Authors · 2022-11-25
> **Response to Reviewer Vu68 -- Part 3**
>
> **(3) “Section 6: given the fact that $K(\gamma,T)$  is exceedingly big, it is hard to deduce / believe from (25) a "provable justification to two remarkable properties exhibited by solutions obtained via gradient-based methods, namely, dropout-stability and connectivity" as stated.”**
>
> This is a very good point and the dependence of the bounds on the time of the dynamics $T$ is a common shortcoming of the mean-field analysis. That being said, (i) the dependence on $T$ is overly pessimistic in the theoretical analysis, and (ii) the actual value of $T$ needed by the dynamics to converge is typically small, so the constant is actually not exceedingly big. This is well demonstrated by the numerical results that we will add in the revision, as detailed in the point below.
>
> **(4) “some numerical simulations explaining / highlighting your results would have been much appreciated”**
>
> Thank you for the excellent suggestion. We have performed a numerical experiment to illustrate the phenomenon of dropout stability. Our empirical results agree with the conclusions of equations (25)-(26), which follow from our bounds in Theorem 5.1 and 5.2. Below, we summarize the experimental details and the results. In the revision, we will incorporate this discussion together with a plot (rather than the tabular form presented here for convenience). We hope that our numerical illustration addresses this comment of the reviewer and also the skepticism deriving from the overly pessimistic dependency of our bound on $T$.
>
> We train a two-layer and a three-layer fully connected neural network in the mean field regime on the MNIST dataset. The training algorithm is stochastic gradient descent with momentum, and we evaluate the dropout stability of the learnt models. We find that:
>
> (i) The dropout error does not significantly increase as the training time $T$ grows. This implies that the constant $K(\gamma,T)$, with its problematic dependence on $T$, is actually not that tight.
>
> (ii) The dropout error scales as an inverse polynomial in the width, in agreement with (25)-(26).
>
> For the two-layer network, we take the width $n \in$ {$100,200,400,800,1600,3200,6400$}; for the three-layer network, we take $n_1,n_2=n$ and use the same grid for $n$. We pick the learning rate to be 0.05, and the momentum to be 0.9 (which implies that $\gamma = 2$). We rescale the learning rate so that the scaling of the gradient does not depend on $n$, as required by our theory. The batch size is 100 and we train for 25 epochs in total. For each model, we do 10 i.i.d. experiments and report their average. In particular, we compute the population loss for both the original network and the dropout network, obtained by randomly dropping out half of the neurons. For each experiment, we take 10 random dropout networks and report the average population loss.
>
> *Results for two-layer networks:*
>
> |  width            | 100   | 200   | 400  | 800 | 1600 | 3200 | 6400 |
> |---|---|---|---|---|---|---|---|
> |    step 0      |  0.0000   | 0.0000   |   0.0000 | 0.0000 |  0.0000 | 0.0000|  0.0000|
>  |   step 2500 |  0.0551    | 0.0272 |  0.0129| 0.0065 | 0.0036       | 0.0017|  0.0008 |
>  |  step 5000  | 0.0669    | 0.0323  | 0.153 | 0.0076|0.0037  | 0.0021| 0.0010  |
>  |  step 7500  | 0.0616    |0.0294 | 0.0140  | 0.0068  | 0.0034  | 0.0019 | 0.0009   |
> |  step 10000 |   0.0608| 0.0286 |  0.0136|  0.0065  |  0.0033  |0.0018 |0.0009    |
> | step 12500  | 0.0603 | 0.0283   | 0.0134 | 0.0065  | 0.0032 | 0.0018 |0.0008    |
>  |  step 15000 |  0.0602   |  0.0282| 0.0134|  0.0064 | 0.0032  | 0.0018 |0.0008    |
>
> *Results for three-layer networks:*
>
> |  width            | 100   | 200   | 400  | 800 | 1600 | 3200 | 6400 |
> |---|---|---|---|---|---|---|---|
> |    step 0      |  0.000   | 0.000   |   0.000 | 0.000 |  0.000 | 0.000| 0.000         |
>  |   step 2500 |  0.563    | 0.438 | 0.310 |  0.178| 0.104 | 0.055| 0.025         |
>  |  step 5000  | 0.670    | 0.611  | 0.611 | 0.507|0.395  | 0.180|   0.100       |
>  |  step 7500  | 0.639     |0.551 | 0.562  | 0.449  | 0.387  | 0.208 | 0.092         |
> |  step 10000 |   0.642 | 0.543 |  0.535|  0.414  |  0.340  |0.181 | 0.079         |
> | step 12500  | 0.650  | 0.545   | 0.527  | 0.399  | 0.324 | 0.165 | 0.071        |
>  |  step 15000 |  0.649   |  0.544| 0.525|  0.395 | 0.320  | 0.162    | 0.069       |

---

> ### Author Response · Authors · 2022-11-25
> **Response to Reviewer Vu68 -- Part 4**
>
> We conclude by addressing the minor points.
>
> **(1) “I believe saying "a rigorous justification" is in my opinion an overclaim”**
>
> We will change “it has given a rigorous justification to a number of properties” into “it has provided a path to rigorously understand a number of properties”.
>
> **(2) “top of page 4: should be $N$ instead of $D$”**
>
> Yes, indeed. Thank you for spotting this.
>
> **(3) “it is not clear to me why the mean-field ODE for 3 layers (equation 3) is written in a complete different way than in the case of 2-layers (eq 12)”**
>
> Let us clarify that it is indeed possible to define the mean field limit for three-layer neural networks in the same way as the two-layer case. This is done e.g. in (Araújo et al., 2019). However, this framework requires additional assumptions (namely, first and last layer not trained). Hence, in our work we have decided to follow the neuronal embedding framework, which we regard as a more general alternative. We also note that the neuronal embedding framework can recover the distributional dynamics for two-layer networks as a special case (see Corollary 22 of (Nguyen & Pham, 2020)). We will add this clarification in the revision.

---

### Review · Reviewer_LtDr · 2022-11-19

**Summary Of Contributions:**

This paper studies training 2-layer and 3-layer neural networks with SGD with momentum. The neural networks are parametrized in the mean-field regime. The 2-layer parametrization of the network follows Mei et al'18, Chizat-Bach'18, Rotskoff-Vanden-Eijnden'18, and Sirignano-Spiriopoulous'20. The 3-layer parametrization follows Nguyen-Pham'20.

(1) This paper proves a non-asymptotic error bound between the limiting mean-field dynamics and the dynamics of SGD+momentum as the number of neurons tends to infinity and step size tends to 0. The novelty relative to previous works is incorporating the additional momentum term in SGD.

(2) It is claimed that the solutions of SGD+momentum have dropout-stability and connectivity. This extends Shevchenko-Mondelli'20 to dynamics with the additional momentum term. (The proof of these claims is omitted, but follows in a straightforward way from Shevchenko-Mondelli'20 and the new quantitative convergence bounds.)

(3) It is shown that if the 3-layer network mean-field dynamics converge, then they converge to a global minimizer of the risk. The novelty relative to Pham-Nguyen is to incorporate momentum.





**Audience:**

Yes

**Claims And Evidence:**

Yes

**Requested Changes:**

The paper was well-written, so I do not think anything needs to be changed. However, it could be interesting to also mention in the literature review how the three-layer mean-field parametrization relates to the "maximal-update" parametrization of the Yang-Hu'21 paper which was also cited. Is the Yang-Hu parametrization incomparable to the one in this paper?

Typo:
-- In page 11, the second equation from the top should read $w_2*(t, w_1(0,C_1), w_2(0,C_1,C_2), w_3(0,C_2))$ instead? Where $w_2$ has been replaced by $w_3$.

**Strengths And Weaknesses:**

Strengths: The writing is precise and easy to follow. I checked the math, and it seemed correct.

The submission is certainly of interest to some readers of TMLR, since stochastic heavy ball methods are more commonly used in practice than vanilla SGD without momentum. Furthermore, analyses of the 2-layer SGD+momentum mean-field limit have been performed in the past, but without quantitative bounds on convergence -- so these quantitative bounds on convergence are new and will be appreciated by some members of the community.

This work seems like a good reference that other papers will be able to use whenever they want to analyze mean-field dynamics with momentum.

Weaknesses: The results in this paper mainly seem like modifications of existing analyses of mean-field dynamics. I am unsure what is qualitatively new about the momentum term. I suppose that a take-away of this paper is that one can do the same analyses as before, but adding a momentum term, and not much changes.

The proofs of the claims in Section 6 are missing, although it is clear that they follow in a straightforward way from previous work + the new quantitative analyses of convergence. (So this is not so important.)

---

> ### Author Response · Authors · 2022-11-25
> **Response to Reviewer LtDr**
>
> We thank the reviewer for the careful review and the positive comments. We address the questions and concerns raised by the reviewer below. We will post an updated version of the paper after receiving all the reviews, as suggested in the TMLR guidelines.
>
> **(1) how the three-layer mean-field parametrization relates to the "maximal-update" parametrization of Yang-Hu'21**
>
> We thank the reviewer for bringing up an interesting point. The short answer is that our mean-field analysis does not directly apply due to the different scaling, but we believe it is applicable after slightly changing the definition of the mean-field dynamics.
>
> First of all, the $\mu$P parameterization for a three-layer neural network is the following: take $n_1 = n_2 = n$, let the scaling of the weights $\boldsymbol{w}_1,w_2,w_3$  be  $1 ,  \frac{1}{\sqrt{n}}, \frac{1}{n}$, respectively, and the scaling of the learning rate be $n$ . In contrast, in our work, we take $n_1 = n_2 = n$ , the scaling of the weights $\boldsymbol{w}_1,w_2,w_3$  is  $1 ,  \frac{1}{n}, \frac{1}{n}$, respectively, and the scaling factors for the learning rates are $ n, n^2$ and $n$, respectively. Thus, our analysis is not directly applicable due to the different scaling. Nevertheless, we believe that one can still prove the convergence to the mean-field limit by following the approach in the recent paper by Chen et al. (2022). The key observation by Chen et al. (2022) is that, although the scaling of the weights is different, the scaling of $\partial_t H_2(t, C_2(j_2))$ remains the same, which leads to the same limit when taking $n \rightarrow \infty$. Thus, by using arguments similar to those carried out in our work, results along the lines of Theorems 5.1, 5.2 and 7.2 can potentially be proved. We remark that Chen et al. (2022) focus on the case in which the weights of the first layer are fixed and provide an alternative definition of the mean-field limit as a distributional dynamics, which tracks the evolution of the joint measure of $w_3$ and $H_2$. This perspective allows the authors to prove a convergence to the global optimum with a quantitative rate (which is stronger than what we show in Theorem 7.2). However, it remains unclear whether a similarly strong result holds when $\boldsymbol{w}_1$ is also trained, due to the non-convexity of the objective.
>
> Finally, we remark that the different scaling is simply a methodological choice based on the theoretical limiting behaviour.
>
> We will add this discussion concerning the “maximal-update” parameterization in the related work part of the revision, and we will add a reference to the recent work by Chen et al. (2022).
>
> **(2) Typo in page 11**
>
> We thank the reviewer for pointing out the typo, which we will correct in the revision.
>
> (Chen et al., 2022) Zhengdao Chen, Eric Vanden-Eijnden, and Joan Bruna. A Functional-Space Mean-Field Theory of Partially-Trained Three-Layer Neural Networks. *arXiv preprint arXiv:2210.16286*, 2022.

---

### Review · Reviewer_ESTq · 2022-11-29

**Summary Of Contributions:**

This paper provides a mean-field analysis of stochastic  heavy ball (SHB) method for training two-layer and three-layer neural networks. The authors proved that the SHB method converges to a PDE in the large width limit. Quantitative bounds between the dynamics of SHB and the mean-field dynamics were derived. These result extend the earlier mean field results of SGD, which is a first order dynamics, to the setting of second order dynamics. Dropout stability and connectivity were discussed as consequences of the mean field analysis.  The authors also obtained the global convergence of the mean-field ODE for three layer networks.

**Audience:**

Yes

**Claims And Evidence:**

Yes

**Requested Changes:**

I would strongly suggest the authors making a thorough discussion on the  values and implications of the mean field results by addressing the concerns raised above.

**Strengths And Weaknesses:**

Strengths: The results on the heavy ball method presented in the paper are technical, interesting and seem to be new in the literature.  The earlier result on this method by Krichene et al. is only for noisy dynamics, while the present paper considers the noiseless dynamics with the only randomness from the stochastic gradient. Among all the results, the most interesting and novel one to me is the global-in-time convergence of the mean field ODE for three-layer networks.

Weaknesses: Despite the originality of the results, I do have some concerns and comments. My first concern is about the values and implications of the mean field limit of heavy ball method. The authors discussed the use of mean field results in dropout stability and connectivity. But I was not fully convinced by the discussions. I would imagine one can prove similar mean field results for many other training algorithms. Do they all enjoy dropout stability and connectivity?  Do these mean field results provide any insights on the pros or cons of heavy ball method compared to GD or SGD?

At a technical level, the proof strategies used in the paper are standard and not original. The proofs of mean field limit follow from the standard coupling strategy with minor twists from dealing with the stochastic gradient and multiple layers. The proof of global convergence seems also to be an adaption of the paper by Pham & Nguyen. Part (C3) of Assumption 7.1 needs a proof rather than being made as an assumption. With this assumption, the result looks pretty trivial to me.

---

> ### Author Response · Authors · 2022-12-04
> **Response to Reviewer ESTq -- Part 1**
>
> We thank the reviewer for the careful review and insightful comments. We reply below to all the comments, and we have just posted an updated version of the paper incorporating these discussions.
>
> **”My first concern is about the values and implications of the mean field limit of heavy ball method. [...] I would imagine one can prove similar mean field results for many other training algorithms. Do they all enjoy dropout stability and connectivity?”**
>
> Let us clarify that a **quantitative** convergence to the mean-field limit of the form proved in Theorem 5.1-5.2 implies dropout stability and connectivity. Proving only the consistency of the mean-field limit, as done in Theorem 1 of Krichene et al. (2020), is not sufficient.
>
> Thus, in order to obtain guarantees on dropout stability and connectivity, the key question is whether one can bound the distance between the weights resulting from the training algorithm and i.i.d. particles evolving according to the mean-field dynamics. After a  non-trivial technical effort, such bounds have been derived in (Mei et al., 2019) and (Pham & Nguyen, 2021) for neural networks trained via SGD with two and three layers, respectively. Our paper is the **first** to tackle an algorithm with momentum, namely the stochastic heavy ball method (SHB). This resolves an open question posed by Krichene et al. (2020).
>
> We highlight that, in general, the quantitative convergence to the mean-field limit is far from obvious. It could certainly be the case that other training algorithms do enjoy similar guarantees, but it remains unclear *(i)* what class of algorithms does so, and *(ii)* how to carry out the analysis. As for optimization with momentum, our strategy is very much tailored to the stochastic heavy ball method. We remark that a similar approach could be successful also for Nesterov’s accelerated method, due to the similarity in the continuous limit. However, the study of other popular training algorithms (e.g., Adam) most likely requires an entirely different technical analysis, whose investigation is left as an interesting future direction.
>
>
>
> **”Do these mean field results provide any insights on the pros or cons of heavy ball method compared to GD or SGD?”**
>
> Our bounds in Theorem 5.1-5.2 are dimension free and capture the optimal dependence with respect to the number of neurons of the network. Thus, it is not possible to improve the corresponding dropout stability and connectivity guarantees, in terms of the number of neurons/input dimension. This means that the behavior of SGD and SHB is similar, as both algorithms are optimal in this regard.
>
> We remark that mean-field theory does not only lead to the properties described in Section 6, but it also gives an exact (limit) description of the whole dynamics of the algorithm, which is *different* for SGD and SHB. For example, this could lead to different convergence rates and to a different implicit bias in the algorithm. Our mean-field analysis constitutes the necessary prerequisite to answer these fundamental questions.
>
> **“At a technical level, the proof strategies used in the paper are standard and not original. The proofs of mean field limit follow from the standard coupling strategy with minor twists from dealing with the stochastic gradient and multiple layers.”**
>
> As mentioned in the paper, our proof strategies build on the work by Mei et al. (2019) and Pham & Nguyen (2021) for neural networks with two and three layers, respectively. However, these papers focus on vanilla SGD, and dealing with SHB requires a number of delicate technical results, which are detailed in the appendices. In particular  *(i)* we establish several boundedness and smoothness properties of the mean-field dynamics, *(ii)* we track various new quantities and exploit a second-order Gronwall lemma (Pachpatte’s inequality) to bound them, *(iii)* we perform a discretization of the particle dynamics which is different from the SGD case, and *(iv)* we prove a key measurability property for the second-order dynamics, which characterizes the dependency between layers during training and significantly differs from the two-layer case. In short, even if the overall strategy has similarities with existing work, the technical bulk of the arguments has to be re-worked and ends up being rather different.
>
> Given the practical success of optimization with momentum, we believe that these technical contributions would be of interest to TMLR readers.

---

> ### Author Response · Authors · 2022-12-04
> **Response to Reviewer ESTq -- Part 2**
>
>
> **“Part (C3) of Assumption 7.1 needs a proof rather than being made as an assumption. With this assumption, the result looks pretty trivial to me.”**
>
> The need for part (C3) of Assumption 7.1 comes from the lack of entropic and moment regularization, which makes it difficult to characterize the limiting points of the noiseless mean-field dynamics. We would like to point out that, even with this assumption, the proof of the global convergence for SGD is far from trivial, and it requires several other ideas that were developed in the related work discussed in Section 2. Most notably, the argument crucially exploits a universal approximation property which is preserved during training (for any finite time of the dynamics).
>
> Finally, we note that the uniform convergence of the gradient (the fourth assumption in (C3)) could be replaced by Morse-Sard type of regularity assumptions (see Section 8 in Nguyen & Pham (2020)). However, removing completely assumption (C3) is a challenging problem which remains unsolved also for SGD and is likely to require a significantly different approach.

---

### Decision · Action_Editors · 2023-01-27

**Recommendation:** Accept as is

**Comment:**

The paper is well-written, after a round of revision three expert reviewers think that the paper has solid contributions. The authors have addressed a few minor concerns, added additional discussions, and several claims have been properly worded, which improved the quality of the paper even further. I agree with the reviewers and suggest accepting this paper.


**Audience:**

The paper is very relevant and timely, and I believe many TMLR readers would find interesting.

**Claims And Evidence:**

The claims are well supported with rigorous statements.